# Adaptive translational reprogramming of metabolism limits the response to targeted therapy in BRAF$^{V600}$ melanoma

Lorey K. Smith [1,2✉], Tiffany Parmenter[1], Margarete Kleinschmidt[1], Eric P. Kusnadi [1], Jian Kang[1], Claire A. Martin[1], Peter Lau [1,2], Riyaben Patel[1,2], Julie Lorent[3], David Papadopoli[4], Anna Trigos [1], Teresa Ward[1], Aparna D. Rao[1,2], Emily J. Lelliott[1,2], Karen E. Sheppard [1,2,5], David Goode[1,2], Rodney J. Hicks [1,2], Tony Tiganis[1,2,6], Kaylene J. Simpson[1,2], Ola Larsson [4], Benjamin Blythe[1], Carleen Cullinane [1,2], Vihandha O. Wickramasinghe [1,2], Richard B. Pearson [1,2,5] & Grant A. McArthur[1,2,7✉]

Despite the success of therapies targeting oncogenes in cancer, clinical outcomes are limited by residual disease that ultimately results in relapse. This residual disease is often characterized by non-genetic adaptive resistance, that in melanoma is characterised by altered metabolism. Here, we examine how targeted therapy reprograms metabolism in BRAF-mutant melanoma cells using a genome-wide RNA interference (RNAi) screen and global gene expression profiling. Using this systematic approach we demonstrate post-transcriptional regulation of metabolism following BRAF inhibition, involving selective mRNA transport and translation. As proof of concept we demonstrate the RNA processing kinase U2AF homology motif kinase 1 (UHMK1) associates with mRNAs encoding metabolism proteins and selectively controls their transport and translation during adaptation to BRAF-targeted therapy. UHMK1 inactivation induces cell death by disrupting therapy induced metabolic reprogramming, and importantly, delays resistance to BRAF and MEK combination therapy in multiple in vivo models. We propose selective mRNA processing and translation by UHMK1 constitutes a mechanism of non-genetic resistance to targeted therapy in melanoma by controlling metabolic plasticity induced by therapy.

[1] Cancer Research Division, Peter MacCallum Cancer Centre, Melbourne, Australia. [2] Sir Peter MacCallum Department of Oncology, University of Melbourne, Melbourne, Australia. [3] Department of Oncology-Pathology, Karolinska Institutet, Stockholm, Sweden. [4] Lady Davis Institute for Medical Research and Gerald Bronfman Department of Oncology, McGill University, Montreal, Canada. [5] Department of Biochemistry and Molecular Biology, University of Melbourne, Melbourne, Australia. [6] Department of Biochemistry and Molecular Biology, Monash University, Melbourne, Australia. [7] Department of Medicine, St. Vincent's Hospital, University of Melbourne, Melbourne, Australia. ✉email: lorey.smith@petermac.org; grant.mcarthur@petermac.org

Clinical outcomes for cancer patients treated with oncogene targeted therapy are limited by residual disease that ultimately results in relapse. This residual disease is often characterized by drug-induced cellular adaptation that precedes the development of resistance. Maximum inhibition of oncogenic signalling has been the prevailing paradigm for improving anti-tumor responses to targeted therapies. For example, maximal suppression of BRAF-MEK signalling using combination therapy is the current standard of care for BRAF mutant melanoma patients. Although this approach extended median survival to over 24 months from a historical base of less than 12 months[1,2], the majority of patients still develop resistance and succumb to the disease. Targeting genetic features of drug-resistant, relapsed disease has emerged as another paradigm to achieve more durable responses, however over 20 mechanisms of resistance have been identified in melanoma patients progressing on targeted therapy[3], revealing limitations in this approach. Prior to relapse, BRAF-targeted therapy induces cellular adaptation that underlies residual disease[4–7], and it has been demonstrated that non-genetic mechanisms underpin this adaptability and may provide new targets to improve clinical outcomes for patients[8,9].

Altered metabolism is a hallmark of cancer that has been intensely investigated over the last decade. How therapy reprograms metabolism and the role this plays during the adaptive response and development of resistance has received much less attention. In the setting of melanoma, we have previously shown that BRAF$^{V600}$ inhibitor sensitivity correlates with glycolytic response in pre-clinical[10] and clinical studies[11]. BRAF inhibition (BRAFi) also renders BRAF$^{V600}$ melanoma cells addicted to oxidative phosphorylation (OXPHOS) by releasing BRAF mediated inhibition of a MITF-PGC1α-OXPHOS pathway[12]. This unleashes adaptive mitochondrial reprogramming, ultimately facilitating drug tolerance likely by compensating for suppressed glycolysis. Consistent with these observations, a "nutrient-starved" cell state emerges during the early drug adaptation phase following combined BRAF and MEK inhibition in vivo, and critically, cells appear to transition through this adaptive state as they acquire resistance[7]. Clinically, PGC1α is induced in BRAF$^{V600}$ melanoma patients treated with BRAFi, either alone[12] or in combination with MEK inhibitors[13], whilst tumours that relapse following MAPK inhibitor treatment display an elevated mitochondrial biogenesis signature[14]. Together, these data suggest that maximal suppression of glycolysis and concurrent inhibition of adaptive mitochondrial metabolism may lead to improved outcomes to MAPK pathway targeted therapy by interfering with metabolic reprogramming underpinning drug-induced cellular adaptation. Notably, however, early results emerging from clinical trials of mitochondrial inhibitors such as biguanides have been largely disappointing[15], and recent preclinical analyses support the concept that mechanisms underlying metabolic plasticity and adaptation may represent a more attractive therapeutic target[16].

Here, we examined metabolic reprogramming in the drug tolerance phase prior to acquired resistance using a genome-wide RNAi screen and global transcriptomic profiling. This systematic approach uncovered mRNA transport and translation pathways as regulators of metabolic response to BRAFi in BRAF$^{V600}$ melanoma cells. Mechanistically, we demonstrate that metabolic response and adaptation are associated with selective mRNA transport and translation of metabolic proteins critical to BRAF inhibitor sensitivity and resistance, including glucose transporters and OXPHOS enzymes. This translational reprograming requires the RNA processing kinase UHMK1 (also known as Kinase Interacting with Stathmin, KIS) that regulates mitochondrial flexibility to control BRAFi sensitivity and controls the abundance of metabolic proteins through the export and translation of the mRNA that encodes them. Importantly, the genetic inactivation of UHMK1 increases sensitivity to BRAF and MEK combination therapy and delays resistance in multiple in vivo models. Together, our data support a model wherein selective mRNA transport and translation contributes to metabolic adaptation underpinning therapy-induced cancer cell plasticity and suggests inhibition of this pathway may delay resistance to MAPK pathway targeted therapies.

## Results

**RNA binding, transport and translation pathways regulate metabolic response to inhibition of oncogenic BRAF signalling.** To identify regulators of metabolic response following treatment with oncogene targeted therapy, we performed a genome-wide RNAi screen using BRAF$^{V600}$ melanoma cells treated with the BRAF inhibitor (BRAFi) vemurafenib (Vem) as a paradigm (Fig. 1a)[17]. We assessed glycolysis in our primary screen based on the observation that glycolytic response confers BRAFi sensitivity in pre-clinical[10] and clinical studies[11]. Lactate is routinely used to measure glycolysis and can be readily detected in growth medium using a lactate dehydrogenase (LDH) enzyme-based reaction. Cell number was determined from nuclear DAPI staining using automated image analysis and change in cell number throughout the drug treatment was used as a proxy of viability, whereby negative values indicate cell death (see "Methods"). For the screen, cells were first transfected with the human siGENOME SMARTpool library and subsequently treated with DMSO or a sub maximal dose of Vem (~IC25; Fig. 1a). We chose a 48 h treatment that is within the window of metabolic adaptation following BRAFi, whereby maximal suppression of glycolysis[10] and adaptive mitochondrial reprogramming (refs. [12,14]; Supplementary Fig. 1A) is observed. Notably, increased expression of *SLC7A8* (LAT2), a biomarker of a drug-tolerant "starved" melanoma state following BRAFi+MEKi in vivo[7], was also observed (Supplementary Fig. 1B). Transfection of WM266.4 BRAF$^{V600}$ cells with siRNA targeting polo-like kinase 1 (PLK1; death control) and pyruvate dehydrogenase kinase 1 (PDK1; glycolysis control) were used to define the dynamic range of the screening assays (Fig. 1b), and notably, glycolysis was significantly more attenuated in Vem+siPDK1 cells compared to either Vem or siPDK1 alone, providing proof of principle for the major aim of the screen. All technical aspects of the screen are described in detail in an accompanying data descriptor[17] and the complete screening dataset is provided as a resource on PubChem[18].

In the absence of drug, viability was impaired by depletion of 622 genes (Supplementary Data 1) that formed a robust network (Supplementary Fig. 2A) enriched for regulators of cell cycle, translation and the ribosome (Supplementary Fig. 2B), processes previously shown to be critical for melanoma survival[19–21]. Glycolysis was reduced by depletion of 164 genes (Supplementary Data 2), and as expected these genes were enriched with annotations associated with metabolism (Supplementary Fig. 2C and Supplementary Data 2). To identify genes that regulate viability and glycolytic response to BRAFi, genes were grouped based on fold change data for each parameter in DMSO versus Vem treatment conditions (see supplementary information). This analysis identified 717 genes (Supplementary Data 3) that were enriched for MAPK and GPCR signalling, and histone methylation, consistent with previous studies investigating BRAFi resistance (Fig. 1c and Supplementary Fig. 2D)[22]. However, the most striking feature of the gene set was RNA binding and transport, which was associated with 4 of the top 20 annotations ranked by *P*-value (Fig. 1c and Supplementary Data 3), with a total of 12 annotations associated with these pathways enriched in the dataset (Supplementary Data 3). The identification of RNA binding and transport genes in our screen was particularly

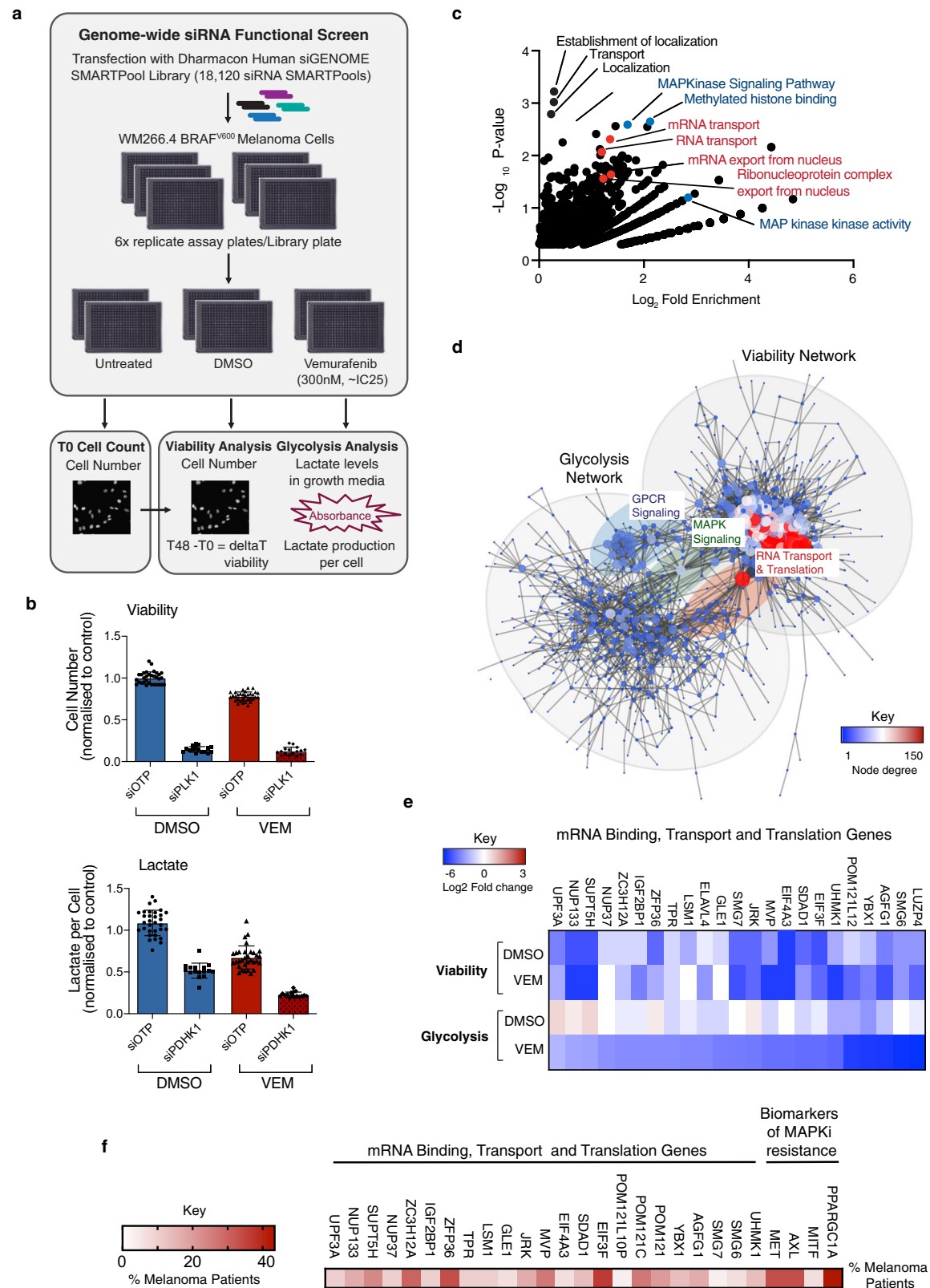

intriguing given these proteins are emerging as key determinants of gene expression programs activated in response to micro-environmental stress, including nutrient deprivation[23]. This group also included components of the EIF3 translation initiation complex, and genes that regulate selective mRNA translation, thus also implicating mRNA translation in metabolic response to

BRAFi. Comparative network analysis revealed three major hubs connect the viability and glycolysis networks; (1) GPCR signalling, (2) MAPK signalling, and (3) RNA transport and translation (Fig. 1d), suggesting these pathways may coordinately regulate metabolic and viability responses to BRAFi. Consistently, seven of the RNA transport and translation genes also enhanced the effects

**Fig. 1 RNA binding, transport and translation pathways regulate metabolic response to BRAF inhibition. a** Schematic summarizing screen workflow (see methods). **b** WM266.4 cells were transfected with the indicated siRNA and treated with DMSO or 300 nM Vem for 48 h. Cell number was calculated using high content image analysis of DAPI stained cells (top panel) and growth media was collected for determination of lactate levels. Lactate absorbance values were normalized to cell number to determine lactate production per cell (bottom panel). Data are presented as mean fold change relative to DMSO siOTP control, ±StDev; Data points are individual replicate wells from two technical replicate screening plates. **c** Functional annotation enrichment analysis was performed on 717 genes that enhanced the effects of Vem on lactate production (DMSO lactate per cell ratio <0.5-fold change and Vem lactate per cell ratio >0.5-fold change; see Supplementary Data 1) using DAVID. Data is displayed as Log2 fold change versus −Log10 p-value. Annotations previously linked with BRAFi response and/or resistance are shown in blue, and the top 4 annotations linked with RNA binding and transport are shown in red. **d** Network analysis was performed on 622 viability screen hits and 717 hits that enhanced the effects of Vem on lactate production using String (see Supplementary Fig. 1). Comparative network analysis was performed using Cytoscape, and hubs connecting the two networks are highlighted. **e** Heat map displaying viability and lactate screening data for the indicated genes. **f** Heatmap displaying percentage of melanoma patients with upregulation of the indicated mRNA transport and translation genes on progression following treatment with MAPK pathway inhibitors (data sourced from https://www.ncbi.nlm.nih.gov/geo/query/acc.cgi?acc=GSE65186[24]; see Supplementary Data 4). See also Supplementary Figs. 1–2. Source data for **b–f** are provided as a Source Data file.

of Vem on viability (Fig. 1e). The major findings of the screen were confirmed using a secondary de-convolution screen, whereby four individual siRNA duplexes were assessed to determine the reproducibility of gene knockdown phenotypes. Confirmed hits were defined as those with ≥2 siRNA duplexes reproducing the primary screen phenotype. Overall, validation rates for the screen exceeded those previously reported for comparable RNAi screens[17], whereby 60% and 53.25% of genes were confirmed as enhancers of the BRAFi response in the context of viability and glycolysis, respectively. Notably, 33% of the RNA transport and translation genes were validated by two or more duplexes (Supplementary Fig. 2E). We next assessed changes in expression of the RNA binding, transport and translation gene set using a published transcriptomic analysis of melanoma patients progressing after treatment with BRAF ± MEK inhibitor treatment[24]. Strikingly, this analysis revealed that 18 out of 23 (78%) RNA transport and translation genes were upregulated in 10–36% of patients progressing on BRAF ± MEK inhibitor treatment (Fig. 1f and Supplementary Data 4). By way of comparison, *PPARGC1A* (PGC1α) was upregulated in 43% of patients in this dataset, whilst other previously documented biomarkers of acquired resistance to MAPK pathway inhibition in patients, *c-MET* and *AXL*, were upregulated in 33% of patients (Fig. 1f). Viewed together, these large-scale systematic analyses support a role for RNA binding, transport and translation pathways in the regulation of metabolic response and viability following BRAFi.

**BRAFi induces transcriptional and translational reprogramming of metabolism in BRAF^V600 melanoma cells.** Given our functional screen suggested a role for post-transcriptional gene regulation pathways in metabolic reprogramming following BRAFi, we next assessed changes in mRNA abundance and translation efficiency by isolating total mRNA and mRNA bound to ribosomes using poly-ribosome (polysome) profiling (Fig. 2a). Cell lysates were fractionated on a sucrose density gradient to isolate mRNA in sub-polysome (RNA-protein (mRNP) complexes and 40S, 60S, and 80S monomer peaks) or actively translating polysome (four or more ribosomes) fractions[25], and were analysed using RNA sequencing (RNA-seq). Of note, the number of ribosomes bound to mRNA is proportional to translation efficiency under most conditions[26]. Global polysome profiles generated from DMSO treated A375 cells revealed a high basal rate of translation, and strikingly, this was potently suppressed by BRAFi at both 24 and 40 h (Fig. 2b). Notably, this global inhibition of mRNA translation coincides with overt cellular adaptation (see above) that presumably requires the synthesis of new proteins, thus supporting the idea that selective mRNA processing and translation pathways play a role during the adaptive response to BRAFi.

In order to identify transcriptome-wide changes in mRNA abundance and translation we used anota2seq (Fig. 2a and Supplementary Data 5)[27]. Consistent with our previous studies[10], GSEA of changes in total mRNA levels revealed downregulation of multiple gene sets associated with the cell cycle and MYC transcription following 24 h of BRAFi, and these gene sets were further downregulated following 40 h treatment (Supplementary Fig. 3A and Supplementary Data 6). In contrast, amongst the most significantly upregulated transcripts following 40 h BRAFi were biomarkers of the adaptive starved melanoma cell state identified in vivo (*SLC7A8*, *CD36* and *DLX5*; Supplementary Fig. 3B)[7]. We next explored the global relationship between mRNA levels and mRNA translation efficiency during the drug treatment time course. Notably, although changes in total mRNA levels correlated strongly with changes in polysome association after 24 and 40 h BRAFi compared to DMSO ($R^2 = 0.94$, and 0.91 respectively), this relationship was less apparent when the 24 and 40 h timepoints were compared ($R^2 = 0.57$) (Fig. 2c), indicating that changes in polysome-associated mRNA cannot be solely explained by corresponding changes in mRNA abundance. These data indicate that mRNA transcription and processing is tightly coupled with mRNA translation efficiency within the early BRAFi response, however, interestingly, this relationship appears to be uncoupled later during drug-induced adaptation (from 24 to 40 h) indicating post-transcriptional modes of gene expression regulation. Analysis at the pathway level using GSEA also revealed differences between mRNA levels and mRNA translation 24 and 40 h post treatment, whereby the cell cycle and MYC pathways were the only significantly downregulated pathways in both datasets, and notably, decreases in translation efficiency of these pathways occurred later in the drug treatment at the 40 hr time point (see above). Comparative analysis of total mRNA and polysome-associated mRNA levels identified genes with changes in total mRNA that were not reflected by a similar change in polysome-associated mRNA. These genes are termed translationally buffered (see above)[27], and indicate post-transcriptional mechanisms of gene regulation. GSEA of the translationally buffered gene set identified enrichment of multiple metabolic pathways, including pyrimidine metabolism and multiple OXPHOS gene sets (Figs. 2d and S3C). Furthermore, functional annotation enrichment analysis of significant buffered genes (FDR < 0.1; see above) also revealed enrichment of OXPHOS and aerobic respiration ($p < 0.05$; Fig. 2e and Supplementary Data 7), further supporting post-transcriptional regulation of aerobic mitochondrial metabolism following BRAFi. Of note, these OXPHOS gene sets showed an overall decrease in total mRNA levels and no change in polysome-associated mRNA levels, as observed in the single-sample GSEA pathway activity plot (Fig. 2f). We next assessed individual components of the OXPHOS gene sets using

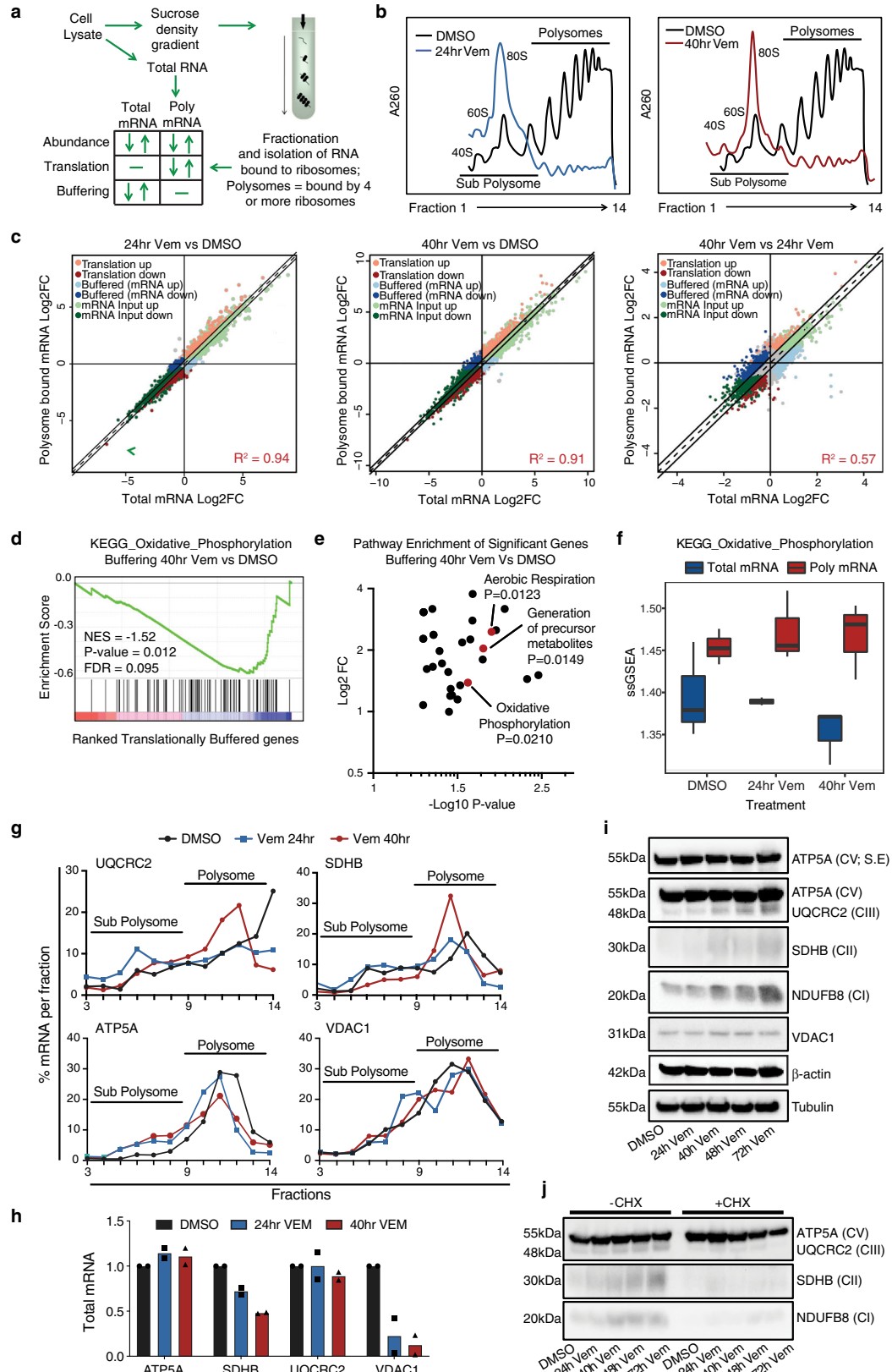

RT-qPCR analysis of independently generated samples (Fig. 2g). Analysis of polysome-bound UQCRC2 (OXPHOS complex III) and SDHB (OXPHOS complex II) mRNA revealed an initial decrease in translation efficiency 24 h post Vem treatment, followed by a pronounced redistribution of these mRNA to heavy polysome fractions after 40 h treatment (Fig. 2g), indicating an

increase in mRNA translation efficiency following 40 h BRAFi. Total UQCRC2 mRNA remained unchanged, while a decrease in SDHB was observed (Fig. 2h). Consistent with elevated translation efficiency, UQCRC2 and SDHB protein levels increased after 40 hr Vem treatment and continued to increase throughout a 72 h treatment time course (Fig. 2i). VDAC1 (voltage-dependent anion

**Fig. 2 BRAFi induces transcriptional and translational reprogramming of metabolism in BRAF<sup>V600</sup> melanoma cells. a** Schematic summarising the polysome profiling assay and different modes of gene expression identified using anota2seq applied to RNA sequencing samples. **b** Polysome profiles of A375 cells treated with either DMSO or 1 μM Vem for the indicated time on a 10–40% sucrose gradient (representative of $n = 3$ biologically independent experiments). **c** Scatterplots of Log2 fold change (Log2FC) total mRNA vs polysome-bound (translated) mRNA in cells treated with DMSO or 1 μM Vem for the indicated time. Different modes of gene expression identified by anota2seq are shown. See Supplementary Data 5 for source data ($n = 3$ biologically independent experiments). **d** Significantly enriched pathways for the different modes of gene expression were identified using gene set enrichment analysis (GSEA; https://www.gsea-msigdb.org/gsea/index.jsp; NES normalised enrichment score, corrected for multiple comparisons using false discovery rate (FDR); FDR < 0.1; see also Supplementary Fig. 3 and Supplementary Data 6). GSEA plot of the KEGG oxidative phosphorylation (OXPHOS) pathway is shown. **e** Functional annotation enrichment analysis was performed on 579 significantly buffered genes (FDR < 0.1; Supplementary Data 5) using DAVID (https://david.ncifcrf.gov/; GO Biological Process and KEGG; *P*-value < 0.05; Source data and *P*-values are shown in Supplementary Data 7). **f** Single sample GSEA (ssGSEA) pathway activity plot demonstrating translational buffering of the OXPHOS pathway. Box plot indicates median (middle line), 25th and 75th percentile (box), and range (whiskers). **g** Distribution of mRNA encoding the indicated genes on a 10–40% sucrose gradient was determined using RT-qPCR following 1 μM Vem treatment for the indicated time (representative of $n = 2$ biologically independent experiments). **h** Total mRNA levels of the indicated genes was determined using RT-qPCR. **I** Whole-cell lysates were analysed by western blot for the indicated proteins following treatment with 1 μM Vem for the indicated time (representative of $n = 3$ biologically independent experiments; tubulin is loading control for ATP5A, UQCRC2, SDHB, NDUFB8, and β-actin is loading control for VDAC1). See also Supplementary Fig. 3. **j** Whole-cell lysates were analysed by western blot for the indicated proteins following treatment with 1 μM Vem for the indicated time, with or without cycloheximide (CHX; 100 μg/mL) treatment (representative of $n = 3$ biologically independent experiments); ATP5A is loading control for UQCRC2, SDHB and NDUFB8. See also Supplementary Fig. 3. Source data for **b**, **e** and **g**–**j** are provided as a Source Data file.

channel 1) was translationally buffered, whereby no change in polysome-bound mRNA or protein levels were observed, despite a significant reduction in total mRNA levels (Fig. 2g–i). These data, therefore, indicate multiple modes of post-transcriptional regulation for OXPHOS associated proteins in response to BRAFi. Demonstrating specificity in the analysis and regulation of pathway components, analysis of ATP5A (OXPHOS complex V) revealed no significant change in translation efficiency, total mRNA levels or protein levels. Notably, treatment with the mRNA translation inhibitor cycloheximide (CHX) obliterated the BRAFi-induced increase in UQCRC2, SDHB and NDUFB8 OXPHOS proteins (Fig. 2j), directly confirming a role for mRNA translation in OXPHOS protein accumulation following BRAFi. In contrast, CHX did not affect ATP5A protein levels thus suggesting that ATP5A is regulated at the level of protein stability during the acute response to BRAFi in melanoma cells. We also noted that MYC targets were enriched in the translational buffering dataset, potentially indicating that transcriptional downregulation of MYC targets may be uncoupled from changes in corresponding protein levels. Because MYC-dependent regulation of glycolysis is a critical factor determining BRAFi sensitivity[10], we next explored adaptive translational buffering of MYC targets (Supplementary Fig. 3D) that relate to glycolysis, glucose transporter 1 (GLUT1) and hexokinase 2 (HK2). Analysis of GLUT1 and HK2 revealed decreased total mRNA levels throughout Vem treatment (Supplementary Fig. 3E), however no change in polysome-bound mRNA was observed (Supplementary Fig. 3F). Analysis of GLUT1 and HK2 protein levels revealed a decrease following Vem treatment (Supplementary Fig. 3G), however this occurred at later time-points, particularly for GLUT1. Although this does not fit the classical definition of translational buffering (characterized by alterations in mRNA levels that are not accompanied by changes in polysome occupancy nor protein levels), our data suggest that translational mechanisms may blunt rapid transcriptional inactivation of glycolysis pathway components in an attempt to preserve normal rates of glycolysis. We also assessed HIF1α that acts as a central factor in BRAF<sup>V600</sup>- driven glycolysis[10], and here we observed congruent downregulation of total mRNA, polysome-bound mRNA and protein levels (Supplementary Fig. 3E–G). Together these data raise the hypothesis that inactivation of adaptive reprogramming of mRNA translation may achieve more rapid and complete inactivation of the glycolysis pathway following BRAFi, which is consistent with reduced lactate production in the original RNAi screen when expression of

genes encoding regulators of mRNA processing and translation were reduced.

Viewed collectively, these findings are consistent with our genome-wide functional screen and support a role for selective post-transcriptional mRNA processing pathways in the regulation of the proteome during early adaptive responses to BRAFi. This includes key pathways implicated in metabolic reprogramming by BRAF and BRAFi sensitivity, MYC-driven glycolysis and oxidative mitochondrial metabolism.

**Depletion of the RNA binding kinase UHMK1 sensitizes BRAF<sup>V600</sup> melanoma cells to BRAFi.** Our systematic functional and transcriptomic approaches supported a role for selective RNA processing and translation pathways in metabolic response to BRAFi. Among the RNA processing proteins identified in our screen, U2AF homology motif (UHM) kinase 1 (UHMK1, also known as Kinase interacting with Stathmin, KIS) was of most interest given it validated strongly in the deconvolution screen (see above), and was also part of the RNA transport and translation hub connecting both the glycolysis and viability networks (see above). UHMK1 is the only known kinase to contain an RNA recognition motif (the UHM domain)[28] raising the hypothesis that it may function as a hub linking cell signalling and RNA processing. Moreover, UHMK1 regulates neuronal plasticity and adaptation via selective RNA transport and translation[29,30] thus we hypothesized it may facilitate adaptive cellular reprogramming in the context of adaptation following BRAFi. We next investigated the role of UHMK1 in the regulation of proliferative and metabolic responses to BRAFi in a panel of BRAF mutant melanoma cell lines (Figs. 3 and S4). First, UHMK1 knockdown was confirmed using RT-qPCR and western blotting (Supplementary Fig. 4A). Because the available UHMK1 antibodies do not specifically detect the endogenous human protein in our melanoma cells, we also confirmed increased levels of its key target p27, which is degraded following phosphorylation by UHMK1[31]. siUHMK1 + Vem treated cells showed more attenuated lactate production (Fig. 3a), glucose utilization (Fig. 3b), and extracellular acidification rates (ECAR; Fig. 3c), when compared to BRAFi alone, indicating a reduction in glycolysis. A more marked reduction in cell number (Supplementary Fig. 4B) and cell proliferation (Fig. 3d, e) was also observed in siUHMK1 + Vem cells compared to Vem alone, and conversely, an increase in cell death was observed in three out of four cell lines (Fig. 3f). Together

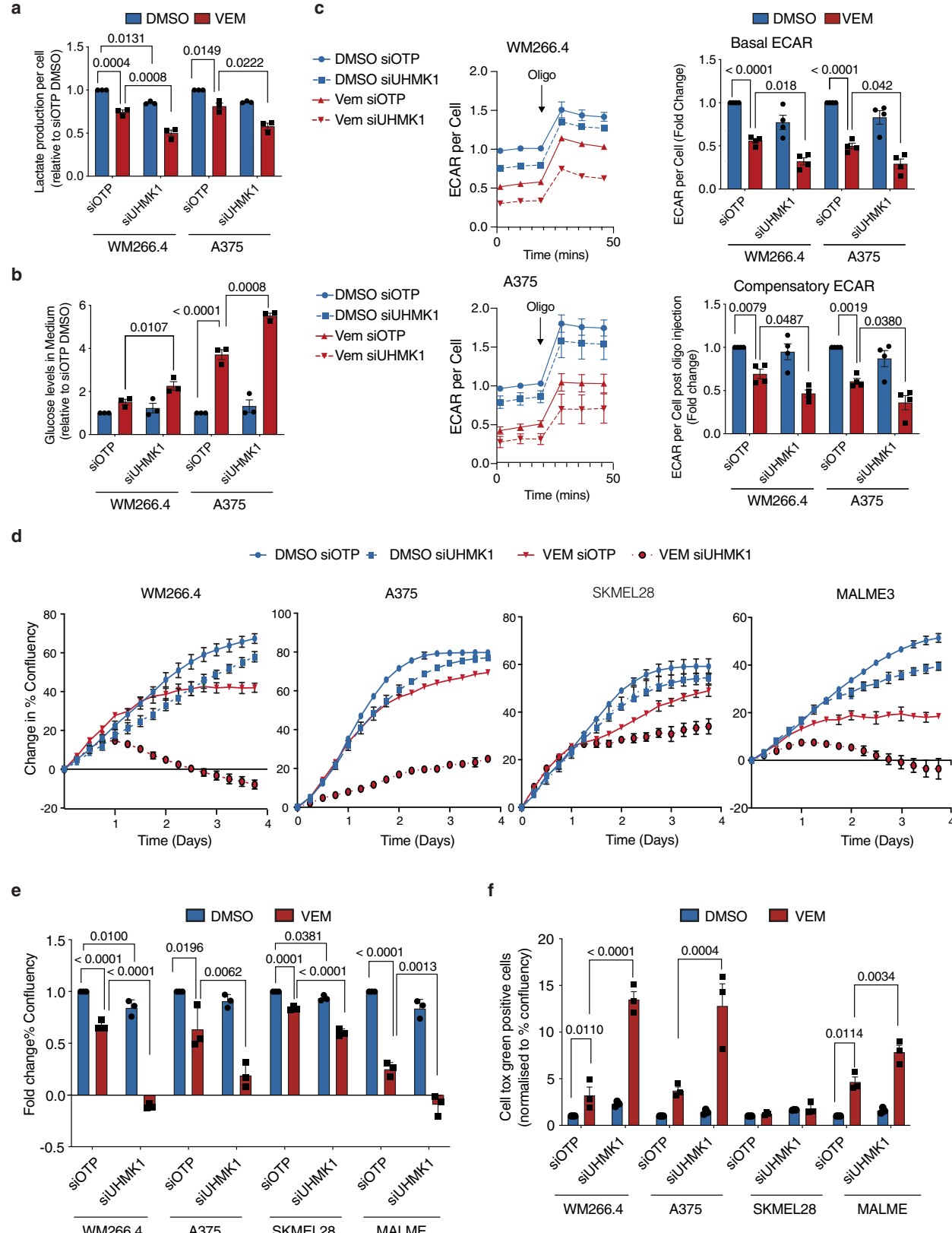

these data confirm a role for UHMK1 in glycolytic, proliferative, and viability responses to BRAFi in BRAF$^{V600}$ melanoma cells.

**UHMK1 reprograms mitochondrial metabolism in response to BRAFi in BRAF$^{V600}$ melanoma cells.** We next investigated whether UHMK1 can also promote adaptive reprogramming of mitochondrial metabolism in response to BRAFi in melanoma cells. Due to cell death after 72 h treatment with Vem + siUHMK1 (Fig. 3f), we assessed cells after 48 hr which immediately precedes overt mitochondrial reprogramming (see above). Analysis of oxygen consumption rates (OCR) using Seahorse

**Fig. 3 Depletion of the RNA processing kinase UHMK1 sensitizes BRAF$^{V600}$ melanoma cells to BRAFi.** WM266.4 and A375 cells were transfected with the indicated siRNA and treated with DMSO or 300 nM Vem for 48 hr. Media was collected and lactate production (**a**) and glucose utilization (**b**) was determined. Data are presented as mean values ± SEM ($n = 3$ biologically independent experiments). **c** Extracellular acidification rate (ECAR) was determined using Seahorse Extracellular Flux Analysis and normalized to cell confluency (left panels). Basal ECAR was calculated from the third ECAR reading, and compensatory ECAR was calculated after treatment with the mitochondrial inhibitor oligomycin (fourth ECAR reading; right panels). Data are presented as mean values ± SEM ($n = 4$ biologically independent experiments). **d** Cell proliferation was assessed in melanoma cells transfected with the indicated siRNA and treated with DMSO or 300 nM Vem by monitoring confluency over time using an Incucyte automated microscope. Representative proliferation curves from $n = 3$ biologically independent experiments are shown. Data are presented as mean % confluency ± StDev. **e** Average % confluency (normalized to T0) was calculated from proliferation data shown in (**d**) following 96 h treatment. Data are presented as mean values ± SEM ($n = 3$ biologically independent experiments). **f** Cell death was assessed in melanoma cells treated as in (**e**) using a Cell tox green cell death assay. Data are normalized to % confluency and presented as mean values ± SEM ($n = 3$ biologically independent experiments). Statistical significance was determined using a one-way ANOVA adjusted for multiple comparisons. See also Supplementary Fig. 4. Source data are provided as a Source Data file.

extracellular flux analysis (Fig. 4a) revealed only modest effects on basal OCR (Fig. 4b) in Vem + siUHMK1 treated cells. However, significant reductions in maximal OCR (Fig. 4c), spare respiratory capacity (Fig. 4d) and ATP production (Fig. 4e) were observed, indicating a reduced ability to respond to changes in energy demand and suggesting that UHMK1 can promote mitochondrial flexibility in response to BRAFi. Impaired mitochondrial metabolism in Vem + siUHMK1 treated cells was not associated with a reduced mitochondrial number (Fig. 4f), moreover, only modest effects on *PPARGC1A* mRNA expression were observed (Fig. 4g). We also assessed the expression of mitochondrial transcription factor A (*TFAM*), another key regulator of mitochondrial biogenesis, and again saw no evidence of a role for UHMK1 in its expression. Instead, analysis of OXPHOS protein levels following Vem treatment revealed that increased expression of NDUFB8, SDHB, and UQCRC2 was UHMK1 dependent (Fig. 4h), indicating UHMK1 regulates mitochondrial function via regulation of protein levels rather than changes in mitochondria number. In order to establish whether these metabolic defects underpin the enhanced anti-proliferative and cell death responses to BRAFi in UHMK1 depleted cells, we supplemented growth media with the electron acceptors pyruvate and α-ketobutyrate, which have been shown to rescue proliferation in respiration deficient cells[32]. Although pyruvate and α-ketobutyrate only partially rescue the anti-proliferative effects of the siUHMK1 + BRAFi combination (Fig. 4i), a complete rescue of cell death was observed (Fig. 4j), demonstrating that defects in metabolism in siUHMK1 + Vem treated cells underpin enhanced BRAFi sensitivity. Together these data suggest that UHMK1 mediates adaptive reprogramming of mitochondrial metabolism to limit response to BRAFi.

**UHMK1 associates with mRNA encoding metabolic proteins and regulates their nuclear-cytoplasmic transport in BRAF$^{V600}$ melanoma cells adapting to BRAFi.** In order to establish how UHMK1 regulates metabolic response to BRAFi, we next assessed its role in the mRNA expression pathway from transport to translation. The effect of Vem and UHMK1 knockdown on nuclear-cytoplasmic mRNA export was first assessed using RNA fluorescence in situ hybridization (FISH) with an oligo(dT) probe which specifically binds to poly(A)$^+$ pools of RNA (Fig. 5a). In control conditions, the poly(A)$^+$ signal was predominantly equal between the nucleus and cytoplasm (Fig. 5a), however in contrast, depletion of the principal mRNA export factor NXF1 caused accumulation of the poly(A)$^+$ signal in the nucleus (Fig. 5a, b). Notably, nuclear accumulation of poly(A)$^+$ mRNA was also observed in UHMK1 depleted cells, confirming a role for UHMK1 in mRNA export in the context of melanoma cells. BRAFi also gave rise to a significant increase in the poly(A)$^+$ nuclear to cytoplasm ratio (Fig. 5b), however, no further change was observed in the siUHMK1 + Vem and siNXF1 + Vem treated cells. These data identify UHMK1 as a regulator of global

mRNA export in melanoma cells, however, this role is unlikely to contribute to the effects of UHMK1 depletion in the context of BRAFi. These data also establish a prominent role for mRNA export in the BRAFi response, consistent with the findings of our genome-wide screen.

The more modest phenotype of UHMK1 compared to NXF1 depletion indicated a selective role for UHMK1 in mRNA export. UHMK1 directly regulates localization and translation of specific mRNA transcripts by complexing with mRNA[29,30]. Therefore, to extend our observations, we next assessed individual mRNA transcripts encoding GLUT1 and UQCRC2 that showed evidence of post-transcriptional regulation from our polysome profiling analysis and are critical components of the glycolysis and oxidative metabolism pathways, respectively. To assess whether UHMK1 associates with UQCRC2 and GLUT1 mRNA we performed RNA immunoprecipitation (RNA-IP) assays using UHMK1-V5 expressing A375 cells following DMSO or Vem treatment (Supplementary Fig. 5A). Strikingly, GLUT1 and UQCRC2 mRNA were not found in association with UHMK1-V5 in treatment naïve cells, however, their association with UHMK1 was induced by Vem treatment (Fig. 5c). Indicating specificity of the analysis and the pathway, no ATP5A mRNA could be detected in association with UHMK1 above the Immunoglobulin G (IgG) control in any condition (Fig. 5c). These data demonstrate that UHMK1 can associate selectively with GLUT1 and UQCRC2 mRNA, and this association is induced by BRAFi.

We were next interested in whether UHMK1 can regulate localization of these transcripts. We assessed nuclear-cytoplasmic export of UQCRC2 and GLUT1 mRNA using RT-qPCR analysis of nuclear and cytoplasmic mRNA pools generated from subcellular fractionation. The fractionation was verified by monitoring levels of mRNA known to be enriched within the nucleus (metastasis-associated lung adenocarcinoma transcript 1; MALAT1) and cytoplasm (ribosomal protein S14; RPS14) (Supplementary Fig. 5B; top panel), and western blot analysis of cytoplasmic (tubulin) and nuclear (Histone H3) specific proteins (Supplementary Fig. 5B; bottom panel). Notably, reduced cytoplasmic mRNA (UQCRC2) and increased nuclear mRNA (GLUT1) was observed in the Vem + siUHMK1 treated cells when compared to Vem alone (Supplementary Fig. 5C), culminating in a significant increase in the nuclear/cytoplasm mRNA ratio (Fig. 5d). These data indicate UHMK1 depletion modifies localization of GLUT1 and UQCRC2 mRNA following BRAFi. In contrast, analysis of ATP5A transcripts revealed no significant change in mRNA distribution (Figs. 5d and S5C), consistent with no evidence of a role for post-transcriptional mechanisms or UHMK1 in ATP5A regulation from previous analyses (see above). Together, these observations demonstrate that UHMK1 selectively associates with GLUT1 and UQCRC2 mRNA specifically in the context of BRAFi, and this is associated with changes in nuclear-cytoplasmic mRNA localization.

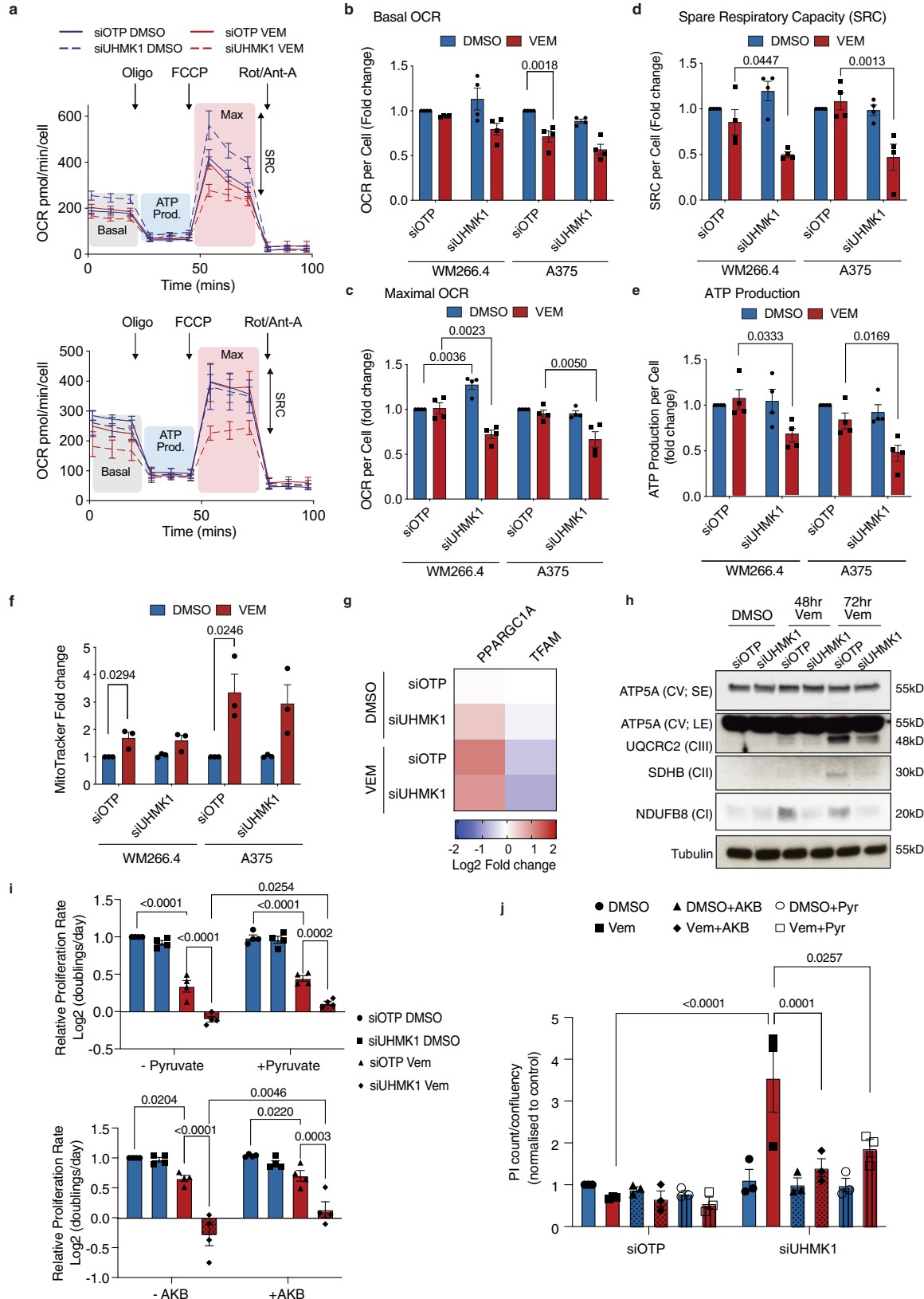

**UHMK1 associates with polysomes and is required for selective translation of mRNA encoding metabolic proteins following BRAFi**. We next tested the hypothesis that UHMK1 can selectively regulate translation of UQCRC2 and GLUT1 mRNA following BRAFi. To achieve this, we analysed de novo synthesis of GLUT1 and OXPHOS proteins by giving a pulse with the

methionine analogue L-azidohomoalanine (AHA), which is incorporated into all newly synthesized proteins (Fig. 6a, top panel). This is followed by biotin labelling, streptavidin pulldown, and western blot analysis. Consistent with our polysome profile analysis, we observed a striking decrease in total AHA-labelled protein confirming global inhibition of protein synthesis

**Fig. 4 UHMK1 reprograms mitochondrial metabolism in response to BRAFi.** WM266.4 and A375 cells were transfected with the indicated siRNA and treated with DMSO or 300 nM Vem for 48 h. **a** Oxygen consumption rate (OCR) was determined using Seahorse Extracellular Flux Analysis and representative profiles for WM266.4 (top panel) and A375 (bottom panel) cells are shown (Oligo = oligomycin; FCCP = Carbonyl cyanide-4-(trifluoromethoxy) phenylhydrazone; Rot/Ant-A = rotenone + antimycin-A; representative of $n = 4$ biologically independent experiments). Effect of gene knockdown and Vem treatment on basal OCR (**b**), max OCR (**c**), spare respiratory capacity (SRC) (**d**), and ATP production (**e**) was determined following treatment with mitochondrial inhibitors as indicated in (**a**). Data are presented as mean values ± SEM ($n = 4$ biologically independent experiments). Statistical significance was determined using a one-way ANOVA adjusted for multiple comparisons. **f** Mitochondrial number was determined using high content image analysis of Mitotracker stained melanoma cells treated as indicated. Data are presented as mean values ± SEM ($n = 3$ biologically independent experiments). Statistical significance was determined using a one-way ANOVA adjusted for multiple comparisons. **g** Effect of gene knockdown and Vem treatment on expression of the indicated genes was determined using q-RT-PCR. Data are expressed as mean Log2 fold change ($n = 3$ biologically independent experiments). **h** Whole-cell lysates were analysed by western blot analysis for the indicated proteins. Data are representative of n = 3 biologically independent experiments (SE = short exposure; LE = long exposure). **i** A375 cells were transfected with the indicated siRNA and treated with DMSO or 300 nM Vem, in the presence or absence of electron acceptors pyruvate (1 mM) or α-ketobutyrate (AKB; 1 mM). Cells were fixed and stained with DAPI 5 days post treatment and proliferation rate was calculated [(Log2(Day 5 count/Day 0 count)/4 days]. Data are presented as mean values ± SEM ($n = 4$ biologically independent experiments). Statistical significance was determined using a two-way ANOVA adjusted for multiple comparisons. **j** Cell death was assessed in A375 cells treated as in (**i**) using a propidium iodide (PI) cell death assay. Data are normalized to % confluency and are presented as mean values ± SEM ($n = 4$ biologically independent experiments). Statistical significance was determined using a two-way ANOVA adjusted for multiple comparisons. Source data are provided as a Source Data file.

following 72 h Vem treatment (Fig. 6a, bottom panel). In contrast, analysis of OXPHOS proteins following Vem treatment revealed an increase in de novo synthesis of UQCRC2 (Fig. 6b, c), and significantly, increased synthesis of this OXPHOS protein was UHMK1 dependent. Again, supporting the specificity of this pathway, no significant change in synthesis of ATP5A protein was observed (Fig. 6b), consistent with polysome profiling of ATP5A mRNA (see above). We also observed that although GLUT1 protein synthesis was decreased following Vem treatment, this reduction was significantly more pronounced following UHMK1 depletion (Fig. 6b, c). Notably, these data are consistent with polysome profiling analysis of GLUT1 mRNA (see above) which indicated that cells may attempt to preserve critical components of the glycolysis pathway via a translational mechanism, and suggest that UHMK1 depletion can overcome this process and thereby achieve more rapid and complete inhibition of GLUT1 protein synthesis. These observations are consistent with enhanced suppression of glycolytic function observed in our siUHMK1 + Vem treated cells (see above). Linking these observations to UHMK1's role in cellular responses to BRAFi, depletion of UQCRC2 and GLUT1 phenocopies UHMK1 knockdown whereby enhanced sensitivity to BRAFi was observed in cell proliferation assays (Supplementary Fig. 6A, B). However, in contrast, no effect on Vem sensitivity was observed in the context of Vem+siATP5A treated cells (Supplementary Fig. 6A, B). We do note that we did not achieve a strong knockdown of ATP5A which may be due to the large role of protein stability in the regulation of this protein in our cells (see above). Notably, a significant decrease in both SRC (Supplementary Fig. 6C) and ATP production (Supplementary Fig. 6D) were also observed in Vem+siUQCRC2 treated cells, but not in ATP5A depleted cells. With regard to glycolysis, whilst depletion of UQCRC2 or ATP5A had no significant effect on ECAR either alone or in combination with Vem, depletion of GLUT1 significantly enhanced the effects of Vem on glycolysis (Supplementary Fig. 6e). Together, these data support a model whereby UHMK1 regulates glycolysis and mitochondrial metabolism following BRAFi via translational regulation of key pathway components including UQCRC2 and GLUT1.

Differential association of mRNA processing and transport proteins with polysomes, and selective delivery of the transcripts they associate with, is an attractive hypothesis to explain translation of selective transcripts. To further explore the role of UHMK1 in adaptive programs following BRAF therapy, we precipitated proteins associated with polysomes using UHMK1-V5 expressing cells treated with DMSO or Vem (Fig. 6d). As expected, small ribosomal protein RPS6 (a 40S ribosome component) was distributed in all fractions in control conditions, whilst large ribosomal protein RPL11 (an 80S ribosome component) was absent from early mRNP and 40S fractions. A significant reduction in the polysome to sub-polysome ratio was observed after Vem treatment (Fig. 6d), consistent with global inhibition of translation (Fig. 6d). Moreover, tubulin was restricted to sub-polysome fractions in both DMSO and Vem treated samples, further confirming specificity of the analysis (Fig. 6d). In contrast, UHMK1-V5 protein was predominantly associated with sub polysome fractions in control conditions, however, a redistribution of the protein to actively translating polysome fractions was observed following Vem treatment (Fig. 6d). These data suggest that not only is UHMK1 recruited to polysomes in melanoma cells, but this association increases in response to BRAF therapy. Consistent with these observations, immunofluorescence analysis revealed a dramatic re-localization of UHMK1 from the nucleus to cytoplasm in cells treated with Vem (Fig. 6e). To further expand these observations, we next investigated whether UHMK1 delivers mRNA to actively translating polysomes. To achieve this, we immunoprecipitated (IP) UHMK1 from heavy polysome fractions using a modified RNA-IP protocol (see methods) and analysed key mRNA cargo identified in our previous analyses. We first verified successful IP of UHMK1 from pooled polysome fractions (fractions 9–14; Fig. 6f). Strikingly, UQCRC2 mRNA co-precipitated with UHMK1 isolated from polysomes in Vem treated cells (Fig. 6g), thus supporting the model that UHMK1 delivers mRNA transcripts to polysomes to facilitate their translation.

Together these data support a model whereby UHMK1 binds to mRNA and is translocated from the nucleus to the cytoplasm in response to BRAFi, where a proportion of the protein (~13%) associates with polysomes and delivers mRNA to facilitate their translation.

**UHMK1 requires a functional kinase and UHM domain to regulate the BRAFi response.** We were next interested in establishing the role of UHMK1's kinase activity and RNA processing function mediated via the UHM domain in the response to BRAFi. The kinase domain of UHMK1 shows limited homology to known kinases, however, a K54R mutation in the putative active site extinguishes kinase activity[33]. The UHM domain of UHMK1 has not been extensively characterised, however, there are multiple features conserved across UHM domain-containing proteins[34]. The UHM domain is classified as

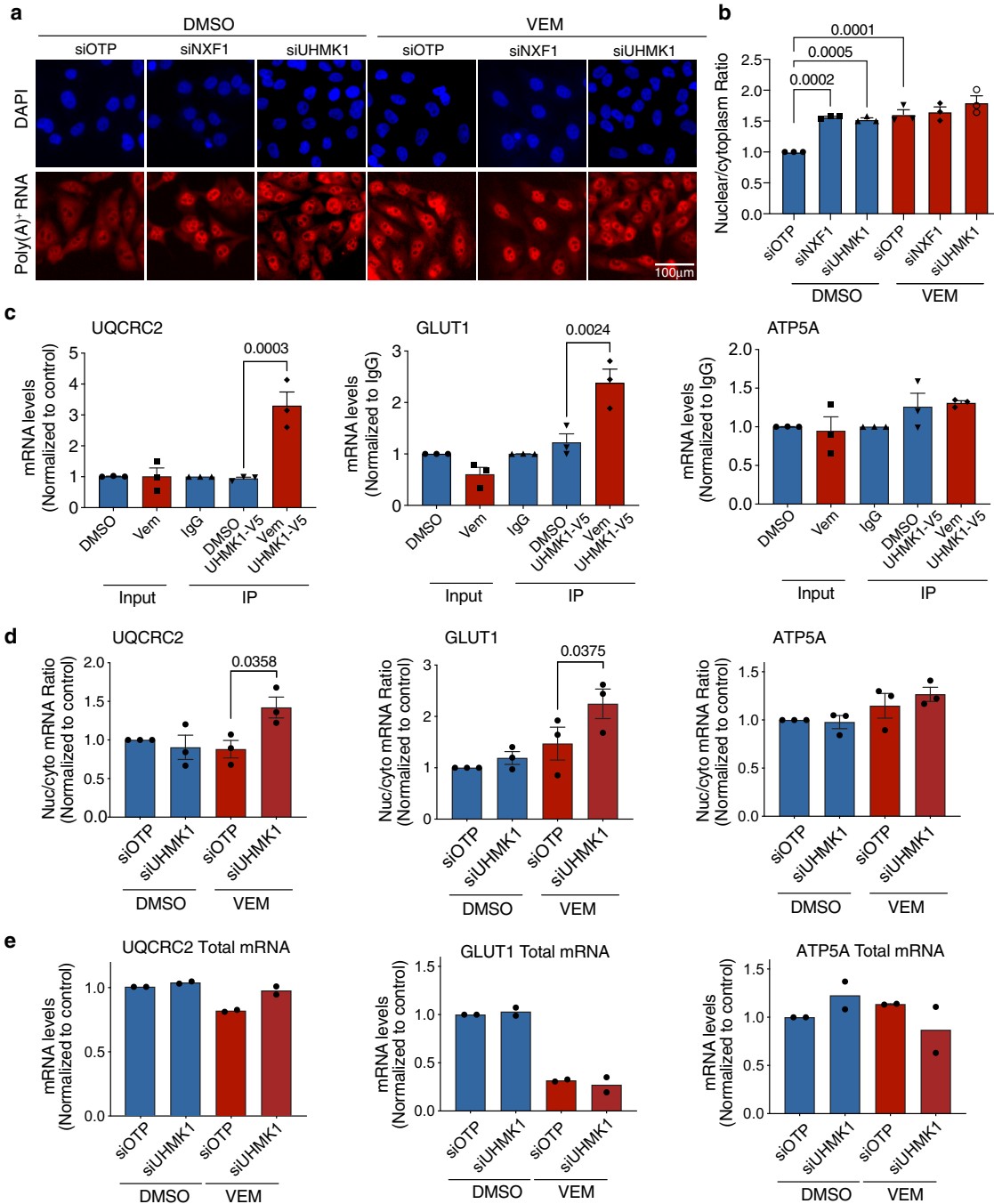

**Fig. 5 UHMK1 associates with mRNA encoding metabolic proteins and promotes selective mRNA transport in BRAF$^{V600}$ melanoma cells adapting to BRAFi.** A375 cells were transfected with the indicated siRNA and treated with DMSO or 1 μM Vem for 48 h. **a** RNA fluorescence in situ hybridization (FISH) using a poly(A)$^+$ RNA specific probe in A375 cells treated as indicated (representative of $n = 3$ biologically independent experiments). **b** The nuclear to cytoplasm ratio of poly(A)$^+$ RNA was calculated using high content image analysis. Data are expressed as mean fold change ± SEM ($n = 3$ biologically independent experiments). Statistical significance was determined using a one-way ANOVA adjusted for multiple comparisons. **c** RNA immunoprecipitation (RNA-IP) assays were performed in UHMK1-V5 expressing A375 cells following treatment with DMSO or 1 μM Vem for 48 h. The indicated mRNA transcripts were then analysed using RT-qPCR. Data are expressed as mean ± SEM ($n = 3$ biologically independent experiments). Statistical significance was determined using a one-way ANOVA adjusted for multiple comparisons. **d** Cell lysates were fractionated into nuclear and cytoplasmic pools of RNA and analysed for the indicated genes using RT-qPCR. The nuclear/cytoplasm (Nuc/Cyto) ratio was calculated from analysis of individual nuclear and cytoplasmic compartments. Data are expressed as mean ± SEM ($n = 3$ biologically independent experiments). See also Supplementary Fig. 5C. Statistical significance was determined using a one-way ANOVA adjusted for multiple comparisons. **e** Whole-cell lysates were used to assess total mRNA levels. Data are expressed as mean ± SEM ($n = 2$ biologically independent experiments). See also Supplementary Fig. 5. Source data are provided as a Source Data file.

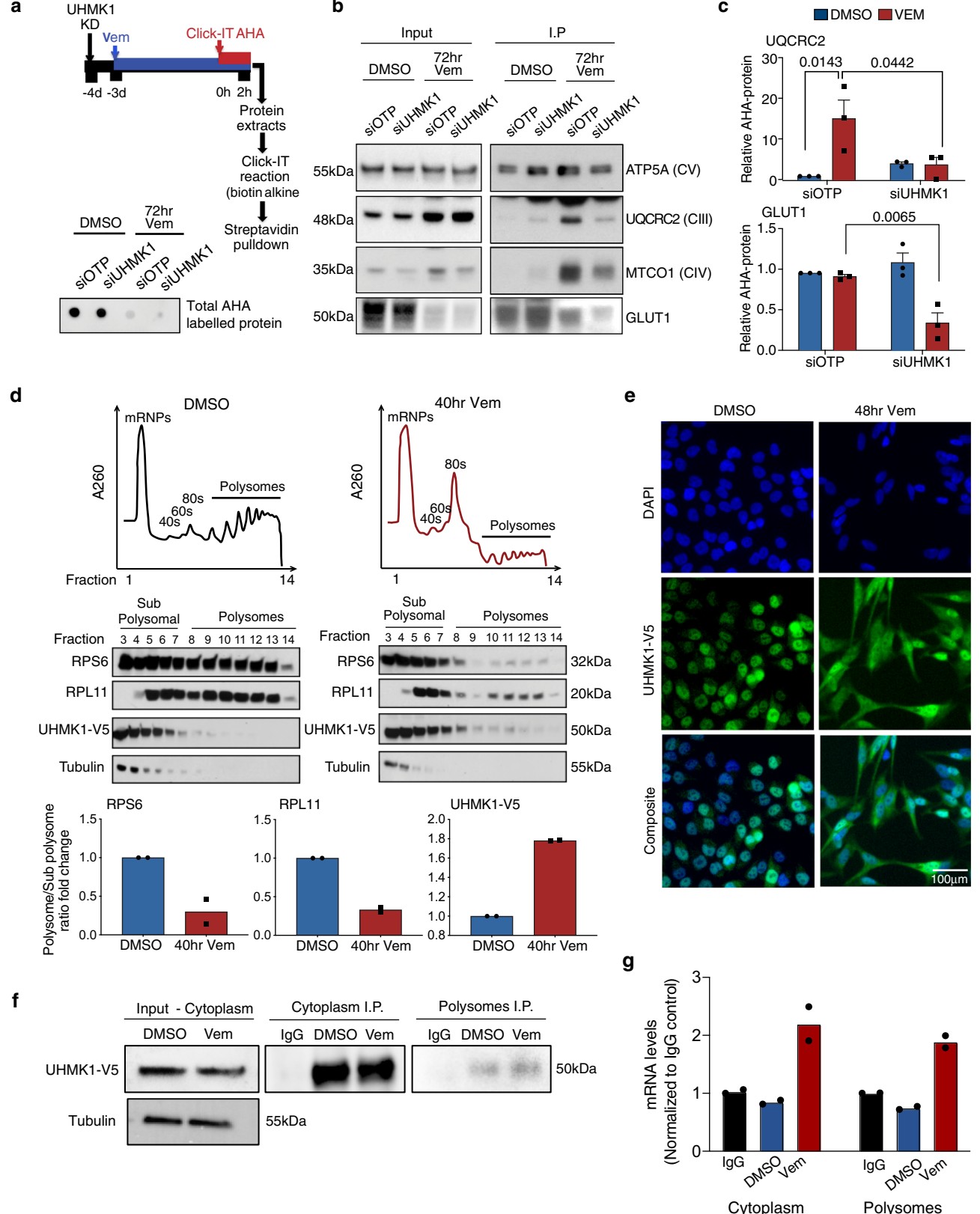

an RNA binding domain based on the presence of ribonucleo-protein (RNP) 1 and RNP2 RNA recognition motifs, however, these motifs are atypical, which is consistent with the previously documented ability of UHM domains to interact with RNA processing proteins[34]. The UHMK1 RNP1 motif shows conserved residues with the consensus RNP1 sequence at position 1,2 and 5

(Supplementary Fig. 7A)[35], and structural modelling using AlphaFold (https://alphafold.ebi.ac.uk/)[36] supports an important function for these residues based on their predicted involvement in hydrogen bond formation and presence in the juxtaposed RNP2/RNP1 core (Supplementary Fig. 7A). The UHM domain also contains a conserved RXF motif that is required for

**Fig. 6 UHMK1 associates with polysomes and regulates selective translation of mRNA encoding metabolic proteins following BRAFi. a** Schematic depicting the AHA-based de novo protein synthesis assay (top panel) and dot blot (bottom panel) showing total AHA labelled protein obtained from siOTP or siUHMK1 transfected cells following treatment with DMSO or 1 μM Vem for 72 h. Data are representative of $n = 3$ biologically independent experiments. **b** Protein lysates from input samples (left panel) and following streptavidin IP (right panel) were assessed using western blot analysis of the indicated proteins. ATP5A does not change with Vem treatment (see Fig. 2) and was used as a loading control. Data are representative of $n = 3$ biologically independent experiments. **c** Quantitation of AHA labelled protein shown in (**b**). Data represent mean ± SEM from $n = 3$ biologically independent experiments. Statistical significance was determined using a one-way ANOVA. **d** UHMK1-V5 expressing A375 cells were treated with DMSO or Vem for the indicated time, prior to polysome profiling. Representative profiles of $n = 2$ biologically independent experiments are shown (top panel). Proteins were precipitated from the sucrose fractions and the indicated proteins were analysed using western blotting (middle panel). Protein levels in sub polysome (fractions 3–8) vs polysome (fractions 9–14) fractions were calculated using densitometry, and sub polysome to polysome ratios were calculated (bottom panel). Data are representative of $n = 2$ biologically independent experiments. **e** UHMK1 localization was assessed using high content image analysis of UHMK1-V5 expressing A375 cells treated with DMSO or 1 μM Vem for the indicated time. Data are representative of $n = 3$ biologically independent experiments. **f–g** UHMK1-V5 expressing A375 cells were treated with DMSO or Vem for the indicated time, prior to polysome profiling as in (**d**). UHMK1-V5 protein was immunoprecipitated (IP) from the cytoplasm fraction (input) and polysome fractions (fractions 9–14 fractions) and samples were analysed using western blotting for the indicated proteins (**f**). Data are representative of $n = 2$ biologically independent experiments. The indicated mRNA transcripts were then analysed using RT-qPCR (**g**). Data represent mean of $n = 2$ biologically independent experiments. Source data are provided as a Source Data file.

interaction with UHM ligand motif (ULM) containing proteins[34]. Mutation of the RXF motif to AAA in the UHM domain-containing protein SPF45 is sufficient to disrupt interactions with RNA processing proteins and inactivate its RNA processing function[37]. Based on these observations, we introduced the K54R mutation in the kinase domain to assess the role of UHMK1 kinase activity in BRAFi responses, and point mutations in conserved residues of the UHMK1 RNP1 (R369A-G370A-Q-V-F372A) and RXF (R392A-M393A-F394A) motifs (Supplementary Fig. 7B). Inactivation of the kinase domain in the K54R mutant was verified by increased accumulation of p27 protein levels, an established biomarker of UHMK1 kinase activity (Supplementary Fig. 7C)[33]. Notably, we did not observe increased p27 levels in the RXF or RNP1 mutant expressing cells. RNA association was examined by analysing a previously established mRNA target of UHMK1, β-actin[29], in UHMK1 RNA-immunoprecipitation (RNA-IP) experiments (Supplementary Fig. 7C, D). The RNP1 but not RXF motif in the UHMK1 UHM domain is critical for complexing with β-actin RNA, whilst the K54R mutant also showed reduced association with β-actin mRNA (Supplementary Fig. 7D). Because the UHM-RNP1 mutant protein did not associate with mRNA or alter UHMK1 kinase activity, we used this mutant protein to specifically assess the requirement of UHMK1's RNA processing function in the response to BRAFi. First, we established the UHM-RNP1 mutant does not efficiently bind to UQCRC2 or GLUT1 mRNA (Fig. 7a). To assess the contribution of these domains in BRAFi responses, we first genetically inactivated UHMK1 using CRISPR-Cas9 (sgUHMK1) (Supplementary Fig. 7E) and confirmed increased sensitivity of sgUHMK1 A375 cells to BRAFi (Fig. 7b). Notably, the increased sensitivity in sgUHMK1 cells was rescued by expression of UHMK1-V5, but not the kinase-dead K54R-V5 nor the UHM-RNP1-V5 mutant proteins (Fig. 7b). Together, these data demonstrate that UHMK1 regulates response to BRAFi via both its kinase and UHM domain, and thus confirm an essential role for both the kinase and RNA processing function of UHMK1 in mediating adaptive responses to BRAFi.

**Depletion of UHMK1 sensitizes melanoma cells to MAPK pathway targeted therapies in vitro and in vivo.** We were next interested in testing the hypothesis that UHMK1 depletion would improve response and delay resistance following treatment with the current standard of care for BRAF$^{V600}$ melanoma patients, a BRAF + MEK inhibitor combination. First, we performed cell proliferation assays and observed more attenuated proliferation in

cells treated with the siUHMK1 + BRAFi + MEKi triple combination compared to the BRAFi + MEKi combination alone (Fig. 8a, b). To assess the role of UHMK1 in therapeutic response to BRAFi + MEKi in vivo, we implanted A375 cells expressing CAS9 or two independent UHMK1 gRNA into NOD scid interleukin 2 gamma chain null (NSG) mice (Fig. 8c, d). Importantly, increased sensitivity to BRAFi+MEKi combination therapy was observed in mice implanted with both UHMK1 knock out cell lines compared with mice implanted with the control cell line (Fig. 8e), culminating in an overall increase in survival (Fig. 8f). Importantly, we also observed a significant increase in overall survival in mice implanted with WM266.4 sgUHMK1 cells treated with BRAFi +MEKi compared to mice implanted with the control cell line (Supplementary Fig. 8), indicating these observations are not selective to one in vivo melanoma model. Viewed together, these data confirm a role for the UHMK1 RNA processing pathway in MAPK pathway inhibitor responses in BRAF$^{V600}$ melanoma cells both in vitro and in vivo, and demonstrate UHMK1 inactivation is sufficient to delay targeted therapy resistance. Finally, we assessed the effectiveness of UHMK1 depletion in combination with the MEK inhibitor trametinib (tram) in the setting of NRAS mutant melanoma. The siUHMK1 + Tram combination resulted in more robust growth inhibition in multiple NRAS mutant melanoma cell lines (Fig. 8g) providing evidence that UHMK1 depletion can also play a role in MAPK targeted therapy response in the setting of a different oncogenic driver.

Altogether, our findings support a model wherein selective post-transcriptional gene expression pathways regulate metabolic adaptation underpinning targeted therapy response in melanoma. As proof of concept, we demonstrate a role for UHMK1 in the regulation of metabolic response and adaptation following BRAFi by controlling the abundance of metabolic proteins through selective transport and translation of the mRNA that encodes them. Importantly, inactivation of this pathway delays resistance and significantly improves survival following combined BRAF and MEK inhibition in multiple in vivo melanoma models, suggesting this pathway may provide therapeutic opportunities to disrupt non-genetic mechanisms of resistance and delay disease relapse in melanoma.

## Discussion

Despite the success of therapies targeting oncogenes in cancer, clinical outcomes are limited by drug-induced adaptation and acquired resistance[8,24]. An emerging phenomenon observed following inhibition of oncogenic signalling in a range

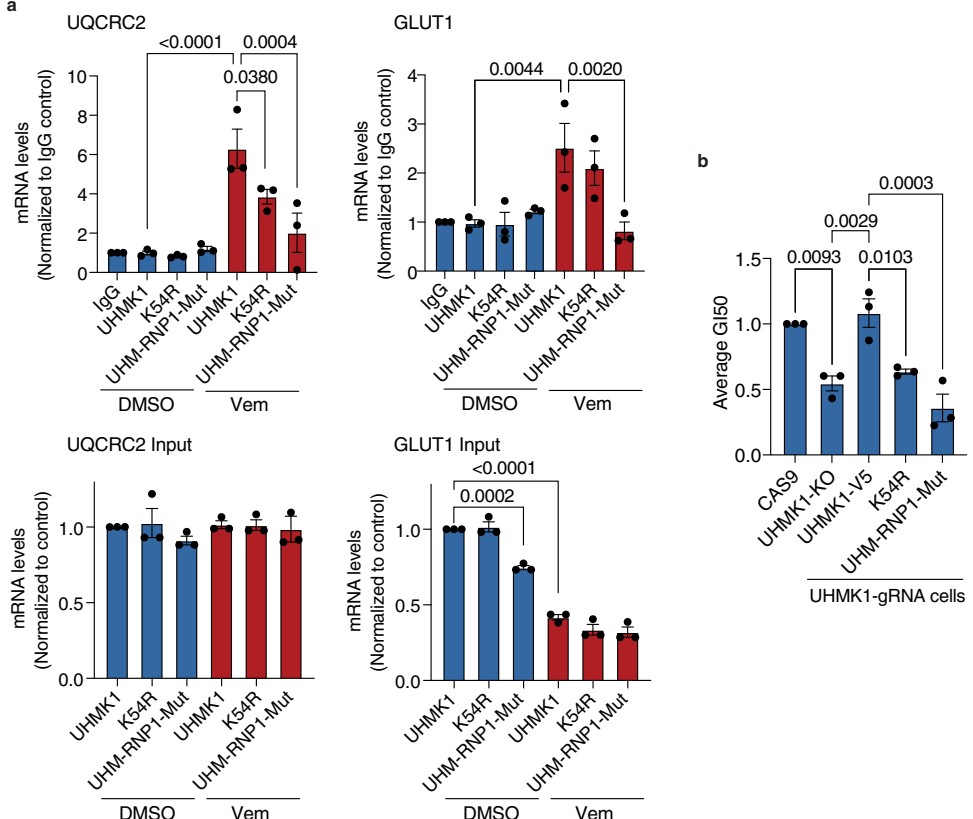

**Fig. 7 UHMK1 requires a functional kinase and UHM domain to regulate the BRAFi response.** UHMK1 was genetically inactivated in A375 cells using CRISPR-Cas9, and the MSCV-GFP vector was stably expressed in A375-Cas9 and A375-UHMK1-gRNA cells. UHMK1-V5, UHMK1-K54R-V5, and UHMK1-RNP1-mut-V5 (R369A-G370A-Q-V-F372A) were ectopically expressed in A375-UHMK1-gRNA cells. **a** RNA immunoprecipitation (RNA-IP) assays were performed in the indicated cell lines following treatment with DMSO or 1 μM Vem for 48 hr. The indicated mRNA transcripts were then analysed using RT-qPCR. Data represent mean ± SEM of $n = 3$ biologically independent experiments. **b** Sensitivity to Vem was assessed using dose-response assays and 50% growth inhibition (GI50) drug concentrations were calculated. Data represent mean ± SEM of $n = 3$ biologically independent experiments. Statistical significance was determined using a one-way ANOVA adjusted for multiple comparisons. Source data are provided as a Source Data file.

of cancers is suppression of glycolysis, and adaptive mitochondrial reprogramming and enhanced reliance on oxidative metabolism[10,12,38–43]. Inhibitors of oxidative metabolism, or the processes controlling adaptive mitochondrial reprogramming, are therefore attractive targets for combination therapy to circumvent acquired resistance before it can develop in a broad range of cancers. Here, we define a mechanism of non-genetic drug adaptation in melanoma whereby adaptive mitochondrial metabolism is regulated at the level of mRNA export and translation, and we identify the RNA processing kinase UHMK1 as a central factor in this process. We propose inactivation of this pathway may provide therapeutic opportunities to interfere with adaptive metabolic reprogramming following oncogene targeted therapy, and delay resistance in melanoma patients.

mRNA translation has been implicated in responses to MAPK pathway inhibition and development of resistance in melanoma[20,44], and a growing body of evidence now supports translational reprogramming as a mechanism that mediates adaptation to metabolic stress[16,45,46]. Here, our systematic functional genomic analysis of metabolic response to BRAFi identified mRNA binding, transport and translation pathways as key regulators of the adaptive BRAFi response, and our analysis of the global translatome directly supports these observations. Despite global suppression of translation during the early drug response phase, extensive reprogramming of specific pathways, including OXPHOS, occurs via changes in mRNA translation efficiency and translational buffering, revealing an underappreciated and prominent role for translational regulation of selective transcripts in the metabolic response to BRAFi. These observations are in line with a recent report describing the translational regulation of selective transcripts in drug-tolerant melanoma cells involving EIF4A[47]. The extensive translational buffering we identified throughout the BRAFi response is particularly intriguing and may represent an adaptive response to preserve the activity of critical pathways. Interestingly, analysis of the translatome following ERα inactivation in prostate cancer cells also revealed extensive translational buffering that appeared to sustain an adaptive proteome[48]. Of note, a recent study described a mechanism whereby mRNA bound to polysomes are protected from degradation following exposure to stress, such as glucose deprivation[49]. It is tempting to speculate that this mechanism may protect specific transcripts to allow rapid protein production during the adaptive stress response, and it is possible this phenomenon may contribute to the buffering phenotype identified in our polysome profiling analysis. Our data also implicates a role for other RNA binding proteins in mechanisms underlying translational buffering, and further investigation of BRAFi induced translational buffering is warranted to more completely understand the post-transcriptional mechanisms that underpin the BRAFi response.

Our analysis of individual OXPHOS related transcripts and proteins revealed regulation at the level of elevated translational efficiency (UQCRC2, SDHB, NDUFB8), translational buffering (VDAC1) and protein stability (ATP5A). Analysis of de novo protein synthesis directly confirmed elevated translation of

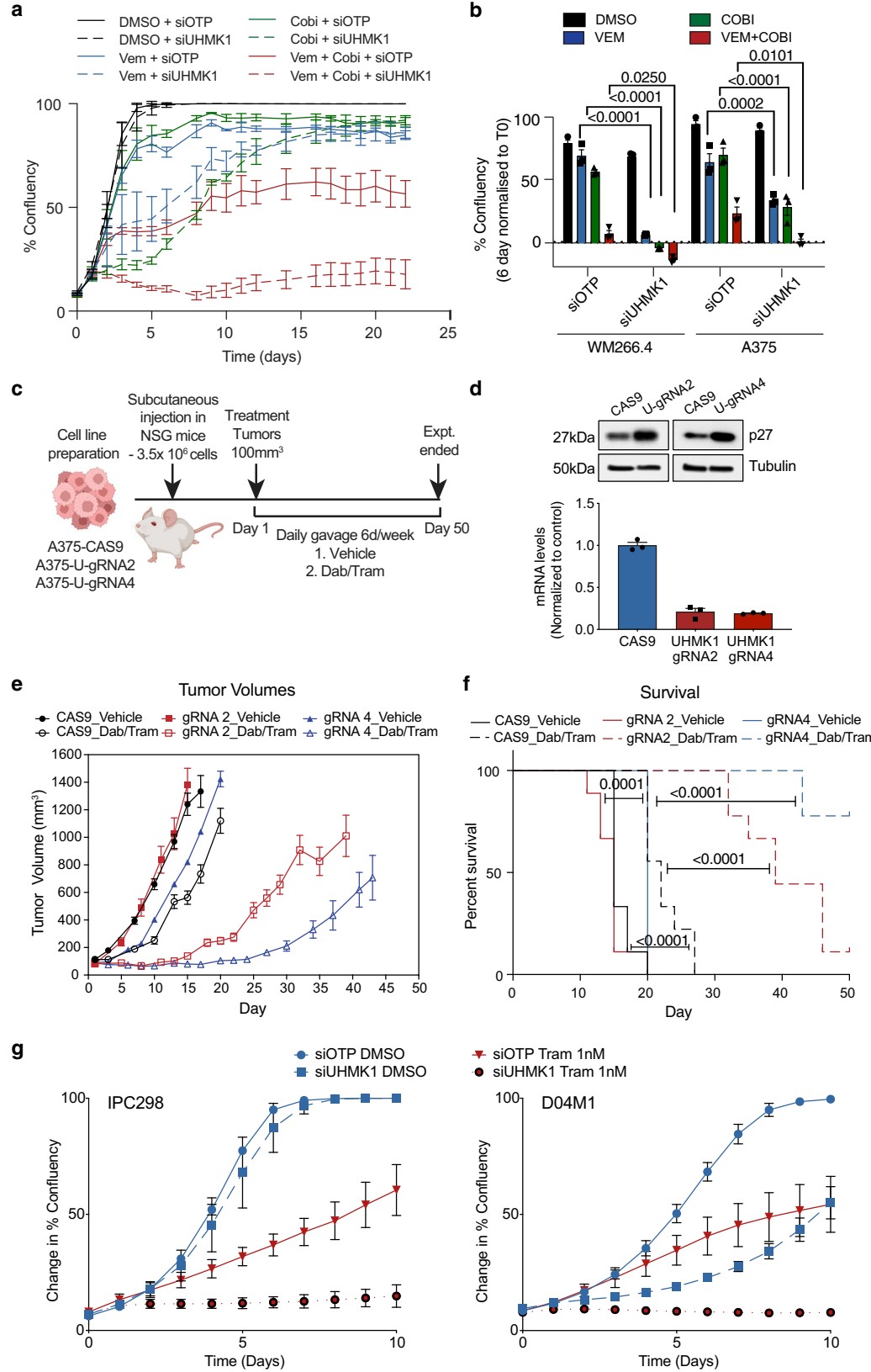

OXPHOS transcripts following BRAFi, and importantly, this was dependent on the RNA processing kinase UHMK1. Translational buffering of glycolysis genes (GLUT1 and HK2) also emerged from our polysome profiling analysis, however, although these genes do not fit the classical definition of buffering due to a reduction in protein levels, these data support a model whereby translational mechanisms may blunt rapid transcriptional inactivation of glycolysis pathway components in an attempt to preserve normal rates of glycolysis and facilitate cell survival. Supporting this model, de novo protein synthesis assays revealed GLUT1 translation was maximally suppressed following UHMK1 depletion in combination with BRAFi, reflective of disrupted

**Fig. 8 Genetic inactivation of UHMK1 sensitizes BRAF$^{V600}$ melanoma cells to BRAF and MEK combination therapy in vitro and in vivo. a** Cell proliferation was assessed by monitoring confluency over time using an Incucyte automated microscope in melanoma cells transfected with the indicated siRNA and treated with DMSO, 300 nM Vem, 10 nM Cobi or Vem + Cobi. Proliferation curves representative of $n = 3$ biologically independent experiments are shown. Data represent mean confluency ± StDev. **b** Average % confluency normalized to T0 following 96 hr treatment as described in (**a**). Data represent mean ± SEM of $n = 3$ biologically independent experiments. **c** Schematic of the in vivo drug sensitivity study. **d** UHMK1 was genetically inactivated in A375 cells using CRISPR-Cas9 and UHMK1 KO was confirmed using RT-qPCR (top panel) and western blot analysis of UHMK1 target p27 (bottom panel). **e** Growth of A375-CAS9, A375-UHMK1-gRNA2 and A375-UHMK1-gRNA4 tumours treated with vehicle or dabrafenib and trametinib (Dab/Tram). Data represent mean tumour growth ± SEM of $n = 9$ individual mice per group. **f** Kaplan–Meier curve of data in (**e**) shows survival advantage where survival is defined as time to a tumor exceeding a volume of 1200 mm$^3$. Statistical significance was determined by Log-rank (Mantel-Cox) test. **g** Cell proliferation was assessed in NRAS mutant melanoma cells (IPC298 and D04M1) transfected with the indicated siRNA and treated with DMSO or 1 nM trametinib (tram) by monitoring confluency over time using an Incucyte automated microscope. Proliferation curves representative of $n = 3$ biologically independent experiments are shown. Data represent mean ± StDev. Source data are provided as a Source Data file.

translational buffering, and accordingly stronger glycolytic suppression was observed in the siUHMK1 + BRAFi cells. Although GLUT1 is a key transcriptional target of MYC and HIF1α, recent studies have also shown regulation of GLUT1 translation by RBPs during adaptive responses to hypoxia[50], and codon-specific translational reprogramming of glycolytic metabolism occurs in melanoma, in this case, mediated by translational regulation of HIF1α by uridine 34 (U$_{34}$) tRNA enzymes[44]. Interestingly, these tRNA enzymes have been linked with translational buffering or offsetting in prostate cancer cells depleted of ERα[48]. Mechanistically, the reduction in metabolic protein synthesis in BRAFi + siUHMK1 treated cells likely reduces the capacity of these cells to cope with glucose deprivation associated with BRAFi, a model supported by a reduction in spare respiratory and glycolytic capacity, and the ability of the electron acceptors pyruvate and AKB to rescue cell death in the siUHMK1 + BRAFi treated cells. Because the UHM RNA processing domain is essential for UHMK1's role in the BRAFi response, we suggest these translational mechanisms contribute to the metabolic plasticity observed in melanoma cells following BRAFi in order to facilitate the survival. Notably, upregulation of OXPHOS proteins occurs in melanoma patients progressing on BRAF and MEK targeted therapy[14], and patient response to BRAFi correlates with glycolytic response as assessed by FDG-PET imaging[11], suggesting that inactivation of adaptive translational reprogramming may mitigate therapy-induced metabolic plasticity and improve targeted therapy response in melanoma patients. Indeed, we observe a significant delay in resistance to MAPK targeted therapy in our preclinical mouse model implanted with multiple melanoma cell lines depleted of UHMK1. Interestingly, UHMK1 has recently been reported to promote gastric cancer progression by promoting de novo purine synthesis[51], revealing a potentially broader role for this kinase in metabolic reprogramming in non-oncogene driven cancers, however in this case, it was UHMK1's kinase activity that mediated this effect. Because UHMK1 knock-out mice remain viable with no severe defects[52], and both the kinase activity and RNA processing UHM domain are required for UHMK1-mediated regulation of BRAFi sensitivity, this makes UHMK1 an attractive therapeutic target and the development of specific inhibitors is a priority.

In order for mRNA to be translated into protein, it must first be exported from the nucleus and transported into the cytoplasm. This process is not always constitutive, as transcript selective RNA export pathways can regulate a range of adaptive biological processes including DNA repair, proliferation and cell survival[53]. Interestingly, RNA binding proteins have recently been shown to regulate pro-oncogenic networks to control melanoma development[54], however, their role in therapeutic response and oncogenic BRAF function has not been reported. Our work now implicates mRNA binding, export and transport in BRAFi response in melanoma cells whereby analysis of poly(A)$^+$-mRNA

localization confirms a prominent defect in mRNA export following BRAFi, which is consistent with the major findings of our functional screen. We also show that UHMK1 selectively associates with mRNA encoding proteins relevant to metabolic response to MAPK pathway inhibitors, modifies their export and delivers them to actively translating polysomes. UHMK1 requires a functional UHM RNA processing domain to modulate sensitivity to BRAFi indicating its RNA processing function is essential for its role in the BRAFi response. However, more studies are required to establish the specific contribution of mRNA export and transport to this phenotype. Nevertheless, we speculate mechanisms of selective mRNA export and transport allow cells to rapidly respond to cellular stimuli and stress such as nutrient deprivation associated with targeted therapy, and likely contribute to mechanisms of adaptive translational reprogramming described above. Indeed, differential association of mRNA binding proteins with polysomes is one mechanism cells employ to rapidly regulate transcript selective translation[55], and association of UHMK1 protein with polysomes following BRAFi is consistent with this concept. Moreover, a recent proteomic analysis of polysomes revealed 45% of all proteins identified were annotated as RNA binding, and a significant proportion of these were regulators of RNA transport and processing[55]. Further analyses are now required to better define the precise role of mRNA export and transport in the BRAFi response.

Viewed collectively, our work supports a model wherein selective mRNA transport and translation is activated in response to therapeutic stress and contribute to metabolic reprogramming underpinning the adaptive therapeutic response in melanoma. Our data identify a key role for UHMK1 in this process, and importantly, the inactivation of UHMK1 delays resistance and improves survival following combined BRAF and MEK inhibition in vivo. We propose that selective RNA transport and translation serve as a non-genetic mechanism of resistance by facilitating cancer cell adaptation and may provide therapeutic opportunities to improve the efficacy of targeted therapies by preventing acquired resistance. We speculate this mechanism may also be relevant in broader oncogene-driven cancer settings where responses to targeted therapies are blunted by phenotypic adaptation involving reprogrammed glycolysis and mitochondrial networks.

## Methods

**Cell lines and reagents**. Vem and its analog PLX4720 were provided by Plexxikon Inc. (Berkeley, CA, USA). Cobimetinib, dabrafenib and trametinib were purchased from Selleck Chemicals (Houston, Texas, USA). All cell lines (WM266.4, A375, MALME3, SKMEL28, IPC298, D04M1, HEK-293T) were purchased from the American Tissue Culture Collection (ATCC), and identity confirmed using STR profiling. All melanoma cell lines were maintained in RPMI 1640 containing 10% FBS, 2 mM L-alanyl-L-glutamine in a 37 °C humidified, 5% CO$_2$ incubator. The BRAF and NRAS mutation status of all cell lines has been reported previously[56]. HEK-293T cells were cultured in DMEM containing 10% FBS, 2 mM L-alanyl-L-

glutamine, in a 37 °C humidified, 5% $CO_2$ incubator. All cell lines were routinely tested for mycoplasma.

**Genome-wide RNAi screen for regulators of glycolysis and viability**. The Dharmacon human siGENOME SMARTpool library (RefSeq27; Dharmacon RNAi Technologies, Horizon Discovery) was used for the screen. This library contains 18,120 SMARTpool reagents (4× individual siRNA duplexes targeting each gene per SMARTpool) targeting each gene in the human genome. The library was arrayed across 58× library plates and screened in 384-well format within the Victorian Centre for Functional Genomics (VCFG, Peter MacCallum Cancer Centre, Australia). Each library plate was assayed in duplicate. All liquid handling steps were performed using a robotic BioTek 406 liquid handling platform, unless otherwise stated. All fixation and staining solutions were filtered (0.45 μm filter) prior to use and plates were briefly centrifuged (500 × *g* for 30 s) prior to all incubations.

**Screen method**. To perform the screen, fresh vials of low passage WM266.4 cells (P8) were recovered and used for each individual batch of screening assay plates (58x library plates; 10–16 library plates screened each batch). For each library plate, 6× assay plates were required (2× no treatment cell number plates (T0), 2× 48 h control treated (0.1% DMSO), 2× 48 h drug treated (300 nM Vem)). Cells were robotically seeded into columns 1–23 of black-walled 384-well assay plates (450 cells/well;Corning) in 25 μL growth media, and 25 μL of media alone was added to column 24 for the lactate assay background control. Plates were pulse centrifuged (500 × *g*) and incubated for 10 min on a level bench at room temperature (RT), then incubated overnight at 37 °C in a Liconic STX200 automated microplate humidified incubator (37 °C with 5% $CO_2$). The transfection was performed 24 h post cell seeding using a Caliper Sciclone ALH3000 liquid handling robot (Perkin Elmer, USA), RNAi MAX transfection lipid (Invitrogen, 0.03 μL per well in 37.5 μL) and siGENOME SMARTpool siRNA at a final concentration of 40 nM. siOTP (D-001810-10-10) was used as the non-targeting control (16× wells per plate), siPLK1 (M-003290-01-0005) was used as a cell viability positive control (8× wells per plate), and siPDK1 (M-005019-00-0005) was used as a lactate assay positive control (8× wells per plate). Plates were pulse centrifuged (500 × *g*) and returned to the automated microplate incubator. 24 h post transfection, transfection media was aspirated (z-height of 36) and replaced with 25 μL of fresh media. Plates were pulse centrifuged to 500 × *g* and returned to the incubator. 48 h post transfection, media was aspirated (z-height of 36) from 4× assay plates and replaced with 25 μL of fresh phenol-free RPMI media with 10% FBS and 2 mM glutamine containing either vehicle (0.1% DMSO) or drug (300 nM Vemurafenib). To generate "T0" cell number plates, 2× assay plates were fixed with 4% paraformaldehyde (PFA; Electron Microscopy Sciences, USA) and stained with DAPI DNA dye (1 μg/mL) in PBS containing 10% triton X-100 (40 μL per well) for 20 min. Plates were imaged on a Cellomics ArrayScan VTi automated microscope (Cellomics, Thermo Fisher Scientific, USA) using a 10× objective and 25 fields were captured per well. Image analysis and cell number calculation were performed using the Cellomics "Cell cycle" bioapplication. Optimal exposure time and object identification thresholds were identified for each individual batch of screening plates. To quantify lactate production per cell, media was collected from each assay plate 48 h post drug treatment. Briefly, plates were centrifuged for 3 min (500 × *g*) and 10 μL media was collected and transferred to a fresh plate using the Sciclone robot. Media was diluted 1:3 with PBS, mixed and stored at −80 °C. Lactate concentration was determined using an L-lactate assay kit (Eton Biosciences) according to the manufacturer's protocol. Briefly, 15 μL of lactate reagent was added to 15 μL media, mixed and incubated at 37 °C in a $CO_2$-free incubator for 45 min. The reaction was stopped through the addition of acetic acid (0.5 M) and absorbance (490 nm) was read using a Cytation 3 Imaging Multi-Mode plate reader (Biotek). In parallel, cells were fixed and stained with DAPI and analysed as described above. Background media absorbance was subtracted from experimental lactate absorbance values, converted to nM concentrations based on a lactate standard curve, and normalised to cell number to generate the parameter lactate production per cell.

**Screen analysis**. Data were expressed as fold change (FC) relative to the average of all siOTP non-targeting control wells included on each plate. Normalised sample values were averaged between replicated plates and robust z-score thresholds calculated. Cell number hits were identified based on T48 cell count (Z-score < −1.5; FC < 0.3; 723 hits) and viability hits were identified based on change in cell number during drug treatment (ΔT48 = T48−T0 cell count; positive values indicate change in proliferation and negative values indicate cell death; Z-score < −1.5; FC < 0.08; 622 hits) in vehicle-treated plates (0.1% DMSO)(Supplementary Data 1). Glycolysis hits were identified based on lactate production per cell (lactate absorbance/cell number) in vehicle-treated plates (0.1% DMSO; Z-score < −1.66; FC < 0.5; 164 hits)(Supplementary Data 2). Due to inaccuracies in lactate quantitation at low cell numbers, lactate data were filtered based on T48 cell count to remove genes with a FC < 0.2. A binning strategy was developed for each of the screen output parameters in order to identify genes with selective activity in the context of drug. To identify drug enhancers in the context of viability and lactate, genes were binned based on fold change data in control versus drug-treated arms of the screen

(DMSO and Vem ΔT48 cell count >0.3; DMSO lactate per cell ratio >0.4-fold change; and Vem lactate per cell ratio <0.5-fold change; Supplementary Data 3). Enrichment analysis for gene ontology (GO) terms (molecular function (MF) and biological process (BP)) and pathways (KEGG and Biocarta databases), was performed using DAVID (v6.8; https://david.ncifcrf.gov/; Supplementary Data 1–3). Protein interaction networks were identified using STRING (https://string-db.org/), and network data were visualised and analysed using Cytoscape (v3.8.2; https://cytoscape.org/).

**Secondary deconvolution validation screen**. To confirm the findings of the screen, we performed a secondary deconvolution validation screen, whereby each of the four individual siRNA duplexes was arrayed into individual wells to confirm the reproducibility of phenotypes. The duplexes were screened at 25 nM using the protocol described above. Duplexes were confirmed as hits if fold-change values for specific phenotypes were ±2 standard deviations of the median of non-targeting controls on each screening assay plate. DMSO/VEM ratios for viability and lactate were also calculated and used to define validated drug enhancement hits.

For a more detailed description of the screen method, technical performance and analysis please refer to our accompanying data descriptor[17]. The complete datasets for the genome-wide primary screen and secondary deconvolution validation screen have been deposited on PubChem[18,57].

**Polysome profiling**. For polysome profiling, cells were pre-treated with 100 μg/mL cycloheximide for 5 min, washed with ice-cold PBS containing 100 μg/mL cycloheximide and lysed in a hypotonic lysis buffer (5 mM Tris-HCl (pH 7.5), 2.5 mM $MgCl_2$, 1.5 mM KCl, 100 μg/mL cycloheximide, 2 mM DTT, 0.5% Triton X-100, and 0.5% sodium deoxycholate). Lysates were pre-cleared by centrifugation to remove nuclei, and the cytoplasm was collected and 20% volume was collected to control for input. The remaining sample volume was loaded onto a 10–40% linear sucrose density gradient (containing 20 mM Tris-HCL (pH 7.6), 100 mM KCl, 5 mM $MgCl_2$) and centrifuged at 95,000 × *g* [SW40 Ti rotor (Beckman Coulter, Inc)] for 2.15 h at 4 °C. Gradients were fractionated and collected (14 fractions per sample), and optical density was continuously recorded at 260 nm using an ISCO Tris and UA-6 UV/VIS detector (Teledyne). Input RNA (20% lysate volume) was used to control for total amount of RNA per sample. RNA was isolated from sucrose fractions using phenol-chloroform extraction. For RNAseq, RNA pellets from fractions 9–14 (corresponding to polysome fractions) were pooled and further purified using RNeasy Mini Kits (QIAGEN), according to the manufacturer's directions for RNA clean up. For analysis of individual mRNA transcripts using q-RT-PCR, RNA was isolated from individual fractions. For analysis of proteins, 10% trichloroacetic acid (final concentration) was used to precipitate protein from each fraction, and protein pellets were subsequently dissolved in SDS sample buffer and analysed using western blot analysis. The ratio of mRNA or protein associated with sub-polysome and polysome fractions was then calculated.

**RNA sequencing (RNAseq) and data analysis**. RNA quality and quantity were confirmed using Agilent Tapestation (Agilent Technologies), and all samples had an RNA integrity number (RIN) of 8.8 or higher. Approximately 1 μg of RNA was used for library preparation using the TruSeq Stranded Total RNA Preparation Kit with Ribo-Zero Gold (Illumina). Briefly, ribosomal RNA (rRNA) was removed using biotinylated, target-specific oligos and magnetic beads. The RNA was then fragmented using divalent cations under elevated temperature and reverse scribed to cDNA with random primers. Indexed adaptors were then ligated and the library was amplified. Samples were then pooled and sequenced on a NextSeq500 (Illumina) high output flow cell to generate ~25 million single-end 75 bp reads per sample. Library preparation and sequencing procedures were performed by the Molecular Genomics core facility at Peter MacCallum Cancer Centre. The RNAseq data generated in this study have been deposited in the GEO database under accession code GSE190071.

Analysis of RNAseq data was performed using the anota2seq R package (v1.4.2)[27], with the following modifications. Genes with an average read count lower than 30 were removed from the analysis and data was normalized using the TMM-log2 approach and a batch effect (replicate number) was included in the models. Changes in polysome-associated mRNA (pool of efficiently translated mRNA, i.e., mRNAs associated with 4 or more ribosomes) can be influenced by changes in corresponding total mRNA levels and/or be the result of changes in translational efficiency. Anota2seq distinguishes changes in amounts of polysome-associated mRNA that are independent of changes in corresponding total mRNA levels (regulation by mRNA translation) from fluctuations at the total mRNA level (regulation by, e.g., transcription and/or mRNA stability). Furthermore, anota2seq can detect translational buffering which is another mode of regulation of gene expression where changes in polysome-associated mRNA and input total mRNA are also decoupled. In this case, polysome-associated mRNA levels (and protein levels) are preserved despite fluctuations in total mRNA levels. GSEA was performed on the gene lists generated by anota2seq using the preranked tool within the GSEA 3.0 software (Broad Institute). Genes were ranked based on Log2FC normalized for the adjusted *p*-value (Log2FCx(1/adj*p*-val)) and run against the Hallmark (v6.2) and KEGG (v6.2) gene sets. Gene sets with FDR < 0.1 were considered significant. Differentially expressed genes were filtered for fold

change ± 1.5 FC and adjusted *p*-values (*P*adj) < 0.1, and gene ontology enrichment analysis using the biological process and KEGG gene ontology sets was performed using DAVID. Gene ontologies with *p*-value < 0.05 were considered significant. Single-sample GSEA (ssGSEA)[58] was performed using the GSVA 53 package (v1.20.0) in R (v3.3.2)[59], which provides an enrichment score of the level of activity of gene sets in individual samples. The KEGG and Hallmark Oxidative Phosphorylation pathway gene sets used in the analyses were obtained from MSigDB c2 v6.2.

**Protein stability assays**. In order to assess protein stability, cells were treated with cycloheximide (CHX; 100 μg/mL) for 20 h prior to the completion of the indicated drug treatment with Vem. Whole-cell lysates were generated and analysed for proteins of interest using western blot analysis.

**siRNA-mediated gene knockdown**. Cells were forward-transfected with 40 nM siGENOME SMARTpool siRNAs (Dharmacon) using 0.08 μl of Lipofectamine™ RNAiMAX (Invitrogen) per 100 μl of transfection media per well, as per manufacturer's directions. Briefly, RNAiMAX transfection lipid was diluted in OPTI-MEM and equilibrated for 5 min, prior to complexing with siRNA for 20 min at RT. A non-targeting siOTP-NT siRNA was used as a control alongside siPLK1 as a technical control for cell viability. Media was changed 24 h after transfection and plates were incubated at 37 °C for indicated times and/or drug treated as described. Knockdown of selected genes were confirmed by RT-qPCR and western immunoblotting.

**Plasmids and establishment of stable cell lines**. pLX304-V5 and pDONR-UHMK1 were purchased from Addgene. The pLX304-UHMK1-V5 expression vector was generated using Gateway cloning (Invitrogen), following the manufacturer's directions. The K54R kinase-dead UHMK1 mutant was generated using a QuickChange site-directed mutagenesis kit (Stratagene), as per the manufacturer's directions. The UHMK1 RNP1 (R369A-G370A-Q-V-F372A) and RXF (R392A-M393A-F394A) mutant constructs were commercially generated (GenScript). https://www.genscript.com/). The panel of UHMK1 mutants were PCR amplified and a V5 tag engineered, and transcripts were then cloned into MSCV-GFP using XhoI and MluI restriction enzyme sites. FuCas9Cherry was a gift from Dr. Marco Herold. HEK-293T cells were transfected with each plasmid along with the appropriate packaging plasmids (pVSVG, pMDL and pRSV-rev for pLX304 and FuCas9Cherry lentiviral vectors, and pEQ and RD114 for the MSCV-GFP retroviral vector) by complexing with polyethylenimine (PEI). Virus generated by HEK-293T cells was supplemented with protamine sulfate (10 μg/ml), filtered, and transferred to melanoma cell lines 3–4 times for 12–16 h. Virus-infected cells were selected with the appropriate antibiotic or by fluorescent activated cell sorting (FACS). FuCas9Cherry cells were sorted for top 30% expressing cells, and MSCV-GFP cells were sorted for "medium" GFP expressing cells to achieve endogenous levels of UHMK1 expression. All cell lines were verified using STR profiling and regularly tested for mycoplasma.

**CRISPR-CAS9 genome editing**. A375-CAS9 and WM266.4-CAS9 stable cell lines were generated as described above and sorted for the top 30% expressing cells using FACS. Synthetic guide RNAs (gRNA) targeting UHMK1 (gRNA-2 AACTGCTT GAGGGCGCCGGG and gRNA-4 CTTGCCGCCAGGAACCACCG) were designed using the Benchling online platform (https://benchling.com/crispr) and were synthesized by Sigma. A375-CAS9 high expressing cells were transiently transfected with each gRNA (20 nM diluted in 10 mM TRIS-HCL pH7.5) and transactivator RNA (20 nM diluted in 10 mM TRIS-HCL pH7.5) using Dharmafect Duo transfection reagent. Cells were single-cell sorted 72 h post transfection into 96-well plates for single-cell cloning. Clones were verified using sequence analysis of gDNA, q-RT-PCR analysis of mRNA and analysis of UHMK1 target protein p27. Because the WM266.4 cells failed to single-cell clone, we used stable expression of UHMK1 gRNA using the transEDIT system where multiple gRNA are coexpressed from the pCLIP-dual-SFFV-ZsGreen plasmid [TEDH-1086688 CCGAAGGCCTCCAGAAAACG; CCAAAAGGAATGCTAAAGAA; TEDH-1086690 TCTTGCCGCCAGGAACCACC; GCACTCCACAATATGTTACG]. Stable cell lines were generated as described above and sorted for the top 30% GFP expressing cells to enrich for UHMK1 editing. UHMK1 depletion was verified using sequence analysis of gDNA, q-RT-PCR analysis of mRNA and analysis of UHMK1 target protein p27.

**Metabolic assays**. For lactate production and glucose utilization assays, 15 μL of phenol-free growth medium from treated cells was removed after pulse centrifugation (500 × *g*). Growth media was diluted 1:3 with PBS and snap frozen at −80 °C. Lactate levels were determined using an L-lactate assay kit (Eton Biosciences) as described above. Glucose levels were determined using a glucose fluorometric assay kit (BioVision) according to the manufacturer's protocol. Absorbance (lactate) and fluorescence (glucose) were determined using a Cytation 3 Imaging Multi-Mode plate reader (Biotek). After the assay, cells were fixed and stained with DAPI DNA dye and cells were imaged using a Cellomics Arrayscan automated microscope. Image analysis and cell number calculation were performed using the Cellomics "Cell cycle" bioapplication (10× magnification; 16× fields), as

described above. Lactate production and glucose utilization were normalised to cell number. To determine the mitochondrial number, cells were labelled with Mito-Tracker (400 nM for 30 min) according to the manufacturer's protocol. Cells were fixed and stained with DAPI DNA dye and cells were imaged using a Cellomics Arrayscan automated microscope or on a Nikon C2 confocal microscope. Image analysis was performed using the Cellomics "Spot Detection" bioapplication (20× magnification; 16× fields).

**Extracellular flux analysis**. Extracellular flux analyses were performed on a Seahorse XF$^e$24 or XF$^e$96 Analyzer (Agilent, USA). For all assays, Flux Packs that contained the cell culture microplates, sensor cartridges and XF calibrant were used (Agilent 102416-100, 102340-100). Assay medium was prepared using Seahorse XF Base Medium DMEM (containing 5.5 mM glucose, 2 mM glutamine and 1 mM sodium pyruvate, adjusted to pH 7.4 and kept at 37 °C; Agilent 102353-100). Prior to cell seeding, Seahorse cell culture plates were coated with Corning Cell-Tak (438512) as per manufacturer's directions. After the desired duration of gene knockdown and drug treatment, cell culture medium was removed and replaced with Seahorse XF medium and cells were equilibrated in a non-CO$_2$ incubator for 1 h prior to the assay. The XF Cell Mito Stress Test protocol was performed as per manufacturer's directions, using oligomycin (1 μM), FCCP (1 μM) and rotenone/antimycin A (0.5 μM). The assay was run with repeated cycles of 3 min mix and 3 min measurements following each drug injection with simultaneous measurement of OCR and extracellular acidification rate (ECAR). After the assay, cells were either fixed and stained with DAPI DNA dye or hoescht live cell nuclear stain was injected at completion of the assay. Cells were imaged using a Cellomics Arrayscan automated microscope (10× magnification; 4× fields) and image analysis and cell number calculation was performed using the Cellomics "Cell cycle" bioapplication as described above. OCR and ECAR values were subsequently normalised to cell numbers.

**Dose-response, proliferation and cell death assays**. Dose-response assays were conducted in 96-well plates following 72 h drug treatments. Cells were fixed and permeabilized with methanol (MetOH), stained with DAPI nuclear dye and imaged using the Cellomics Arrayscan automated microscope (10× magnification; 16× fields). Image analysis and cell number calculation were performed using the Cellomics "Cell cycle" bioapplication as described above. Log[inhibitor] vs. response curves were generated by non-linear regression/curve fitting and GI50 concentrations (the concentration of drug required to reduce growth by 50%) were obtained as a measure of drug sensitivity. GI50s are displayed as mean ± SEM and statistical significance was determined using a Students *t*-test or one-way ANOVA (*p* < 0.05). For proliferation assays, cells were plated at low density and transfected with siRNA 24 h post seeding, and 24 h post-transfection cells were treated with medium containing inhibitors. Phase-contrast images were acquired and analysed daily using the IncuCyte (Essen Bioscience) continuous live-cell imaging and analysis system. For cell death, the growth medium was supplemented with either CellTox Green cell impermeable fluorescent dye or propidium iodide (PI; final concentration 1 μg/mL), and either green or red object counts were determined using the IncuCyte. Green fluorescent or red object counts were normalised to cell confluency. For electron acceptor assays, growth media containing either DMSO or Vem was supplemented with pyruvate (1 mM) or α-ketobutyrate (1 mM), and media was replaced every 3 days. A T0 cell count was determined prior to treatment, and cells were fixed and stained with DAPI 5 days post treatment. Proliferation rate was calculated [(Log2(Day 5 count/Day 0 count)/4 days] and expressed as fold change relative to DMSO siOTP controls.

**RNA-FISH**. Cells were grown in black-walled 96-well plates and fixed with 4% PFA for 5 min at room temperature. Cells were washed twice in PBS prior to overnight incubation in 100% ethanol at −20 °C. The next day, cells were washed in PBS prior to permeabilization with PBS + 0.5% Triton-X for 10 min at room temperature. Cells were then washed 2× with saline sodium citrate (SSC) buffer (Sigma, S6639) prior to incubation for 30 min at 37 °C in pre-hybridization buffer (2× SSC buffer, 0.2% BSA, 20% formamide, 1 mg/mL yeast tRNA, prepared in ultrapure water). Cells were then stained with a Cy3 labelled oligo(dT) primer (Sigma) in hybridization buffer (10% dextran sulfate, oligo(dT) primer 1:500, prepared in pre-hybridization buffer) for 3 h at 37 °C. Cells were then washed 4× in 2× SSC buffer (pre-warmed to 42 °C) for 5 min, 2× in 1× SSC buffer at room temperature for 5 min, and 2× PBS at room temperature for 5 min. Cells were then stained with DAPI (1 μg/mL in PBS) for 5 min at room temperature. Cells were mounted in 100 μL PBS for imaging. Plates were imaged on a Cellomics ArrayScan VTi automated microscope (Cellomics, Thermo Fisher Scientific, USA) using a 20× objective and 25 fields were captured per well. Image analysis and quantification was performed using the Cellomics "Nuclear translocation" bioapplication. A nuclear mask was generated from the DAPI channel and applied to the Cy3 oligo(dT) channel to calculate the average nuclear pixel intensity. A cytoplasmic mask 5 pixels wide was generated 1 pixel from the nuclear boundary in order to quantify the average cytoplasmic pixel intensity. The nuclear to cytoplasm ratio was calculated from these intensities.

**RNA fractionation**. Nuclear and cytoplasmic fractions were obtained by digitonin permeabilization of whole cells and centrifugation. Briefly, cells were harvested in digitonin lysis buffer (50 µg/mL digitonin (ICN Biomedicals), 100 mM NaCl, 10 mM Tris pH 8.0, protease inhibitors (Roche) and RNase inhibitor (Invitrogen)) and incubated on ice for 15 min. Following centrifugation at $1000 \times g$ for 5 min, the supernatant or cytoplasm fraction was separated from the pelleted nuclei, and RNA was isolated using RNeasy Mini Kits (QIAGEN), according to the manufacturer's directions.

**Protein-RNA immunoprecipitation**. Cells were harvested using a non-denaturing hypotonic buffer (5 mM Tris-HCl (pH 7.5), 2.5 mM MgCl2, 1.5 mM KCl, 2 mM DTT, 0.5% Triton X-100, 0.5% sodium deoxycholate, and RNAse inhibitor), and pre-cleared by centrifugation. After protein determination, samples were adjusted to 2 mg and 10% volume was taken for protein and RNA input samples. Samples were incubated with antibody or IgG control at 4 °C overnight under agitation. A/G Sepharose beads were blocked for 30 min in lysis buffer containing 10 mg/mL BSA and 0.1 mg/mL yeast total RNA, prior to incubation with lysates for 4 h at 4 °C under agitation. To elute protein, 50% of the sample was collected by boiling in 3× SDS sample buffer (187.5 mM Tris-HCl (pH 6.8), 6% w/v SDS, 30% glycerol, 150 mM DTT, 0.03% w/v bromophenol blue) and analysed using SDS-Page. In order to avoid background from IgG heavy chains, UHMK1-V5 IP samples were detected with TidyBlot detection reagent (BioRad) that only detects native, non-denatured antibodies. To extract RNA, 1 mL Trizol reagent was added to the remaining 50% of sample and RNA isolated following the manufacturer's protocol. RNA was further purified using RNA clean up columns (Macherey-Nagel) following manufacturer's protocol.

**RNA extraction and analysis of mRNA expression using Quantitative Real-Time PCR (RT-qPCR)**. RNA was extracted using RNeasy Mini Kits (QIAGEN) and cDNA synthesis was performed using High Capacity cDNA Reverse Transcription Kits (Applied Biosystems), according to the manufacturer's directions. RT-qPCR was performed using Fast SYBR® Green PCR master mix using the primers listed in Supplementary Table 1, on a Step One Plus Real-time PCR system (Applied Biosystems). Data were processed using the comparative CT method, relative to the house keeping gene *NONO*. For RNA-IP experiments, data were analysed relative to the housekeeping genes *GAPDH* and *BACT* (β-actin). Changes in mRNA expression were expressed as fold change relative to assay controls and were analysed using a Students *t*-test or one-way ANOVA ($p < 0.05$).

**Western immunoblotting**. Protein was extracted from cells using western solubilization buffer (WSB; 0.5 mM EDTA, 20 mM HEPES, 2% SDS), unless otherwise stated. Protein samples were subjected to SDS-PAGE analysis followed by western immunoblotting using the following antibodies: β-actin-HRP, Sigma A3854 (1:10,000); ERK (p44/42-MAPK), Cell Signalling Technology (CST) 9102 (1:1000); phospho-ERK (p44/42-MAPK; Thr202/Tyr204), CST 9101 (1:1000); GAPDH-HRP, CST 3683 (1:3000); GLUT1, US Biologicals, G3900-0J (1:500); HIF1α, AB2185 (1:1000); HK2, CST 1206 (1:1000); OXPHOS antibody cocktail, AbCam AB110413 (1:1000); p27, BD transduction 610242 (1:1000); RPS6, CST 2217 (1:1000); RPL11, Invitrogen 373000 (1:1000); α-Tubulin, Sigma T5168 (1:10,000); V5, CST 13202 (1:1000); VDAC1, CST 4661 (1:1000). For analysis of OXPHOS proteins using the OXPHOS antibody cocktail, lysates were boiled at 50 °C for 10 min, as per manufacturer's directions, and SDS-PAGE analysis was performed using a 8–16% Mini-PROTEAN TGX Gel. Uncropped and unprocessed scans of all western blots displayed in figures are provided in the Source Data file.

**De novo protein synthesis assay**. De novo protein synthesis assays were performed using the methionine analogue L-azidohomoalanine (AHA)[60]. Briefly, cells were starved for 30 min in methionine-free RPMI media containing 10% dialysed FBS and 0.2 mM L-cysteine. Cells were given a pulse with AHA (100 µM) and incubated for 2 h at 37 °C in a humidified 5% $CO_2$ incubator. Cells were washed in PBS, harvested in lysis buffer (1% SDS in 50 mM Tris-HCl pH 8.0) and boiled at 95 °C or 50 °C for OXPHOS proteins. Biotin labelling was performed using the Click-iT Protein Reaction Buffer kit (Invitrogen, C10276) as per manufacturer's protocol, followed by streptavidin pull-down (Dynabeads M-280 streptavidin) and western blot analysis.

**In vivo mouse experiments**. All animal studies were performed according to protocols approved by the Animal Ethics Committee of Peter MacCallum Cancer Centre and in accordance with the National Health and Medical Research Council Australian code for the care and use of animals for scientific purposes, 8th Edition, 2013. The housing facility was kept at 21 °C, with a relative humidity of around 50%. The light/dark cycle was 14 h light/10 h dark. Xenograft experiments were performed using $3.5 \times 10^6$ A375-CAS9, A375-CAS9-UHMK1-gRNA2 or A375-CAS9-UHMK1-gRNA4 cells prepared in 50% Matrigel and subcutaneously injected into the right flank of 6–7 week old female NOD-Scid interleukin 2 receptor gamma chain null (NSG) mice. For WM266.4 cells, $5 \times 10^6$ WM266.4-CAS9, WM266.4-CAS9-UHMK1-gRNA88 or WM266.4-CAS9-UHMK1-gRNA90 cells were prepared in 50% Matrigel and subcutaneously injected into the right flank of 6–7 week old male NSG mice. Once tumours reached an average volume of 100 mm³, mice were randomized into groups of 8 (WM266.4 study) or 9 (A375 study) for therapy studies. Dabrafenib (30 mg/kg in 0.5% HPMC and 0.2% Tween 80 in $H_2O$) and trametinib (0.15 mg/kg in 0.5% HPMC and 0.2% Tween 80 in $H_2O$) were administered daily via oral gavage for 6 out of 7 days each week. Mice were euthanised when they met a tumour volume ≥1200 mm³.

**Statistical analysis**. All statistical analyses were performed using GraphPad PRISM. Comparisons between two groups were analysed using the student's *t*-test, and where more than two groups were compared, an analysis of variance (ANOVA) was performed, followed by the relevant multiple comparisons test. $P < 0.05$ was considered statistically significant. Un-clustered heatmaps were created in GraphPad PRISM.

**Reporting summary**. Further information on research design is available in the Nature Research Reporting Summary linked to this article.

## Data availability
The primary siRNA screen data generated in this study have been deposited in the PubChem database under accession number AID: 1508588. The processed primary screening data are available in the Supplementary Information/Source Data File. The secondary deconvolution screen data generated in this study have been deposited in the PubChem database under accession number AID: 1508587. The RNAseq data generated in this study have been deposited in the GEO database under accession code GSE190071. The patient data used in this study are available in the GEO database under accession code GSE65186. Source data are provided with this paper and are available in the Supplementary Information and the Source Data File. Source data are provided with this paper.

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

## Acknowledgements

We thank the following Peter MacCallum Cancer Centre core facilities: Victorian Centre for Functional Genomics (VCFG), Molecular Genomics, Flow Cytometry and the Centre for Advanced Microscopy and Histology. The VCFG (K.J.S.) is funded by the Australian Cancer Research Foundation (ACRF), the Australian Phenomics Network (APN) through funding from the Australian Government's National Collaborative Research Infrastructure Strategy (NCRIS) program and the Peter MacCallum Cancer Centre Foundation. We thank Daniel Thomas, Jennii Luu, Kate Gould and Piyush Madham-shettiwar from the VCFG for technical and analytical assistance with the genome-wide RNAi screen. We thank Dr Gisela Mir Arnau from Molecular Genomics for technical assistance with RNAseq. We also thank Rachael Walker and Susan Jackson from the Translational Research Lab and Alison Slater from the Molecular Oncology Lab for assisting in the in vivo studies. This work was supported by the Peter MacCallum Cancer Foundation and grants from the National Health and Medical Research Council of Australia (#1053792 and #1106576), the Cancer Council Victoria (APP1184894) and the CASS Foundation (#8539). A.R., P.L., R.P. and E.L. were supported by doctoral scholarships from the University of Melbourne and Cancer Therapeutics Cooperative Research Centre.

## Author contributions

L.K.S., T.P. and G.M. conceived and designed the project. L.K.S., T.P., K.J.S. and G.M. designed experiments. L.K.S., T.P., M.K., E.K., J.K., T.W., A.R. and D.P. conducted experiments. L.K.S., A.T., J.L., O.L. and D.G. performed data analysis. C.C., B.L. and L.K.S. designed and C.M., P.L., R.P. and E.L. performed the in vivo experiments. V.W., R.B.P., T.T., K.E.S., C.C. and R.J.H. provided critical scientific input, protocols and/or reagents. L.K.S., V.W., R.B.P. and G.M. were involved in writing the manuscript, with all authors providing feedback.

## Competing interests

The authors declare no competing interests.
