## [Peer Review File · Nature Communications]

Adaptive translational reprogramming of metabolism limits the response to targeted therapy in BRAFV600 melanomaReviewers' comments:

Reviewer #1 (Remarks to the Author);

In this manuscript, Smith and colleagues show that post-transcriptional regulation by UHMK1 promotes adaptation of melanoma cells to BRAF inhibitors. UHMK1 promotes expression of factors involved in glucose and mitochondrial metabolism under BRAF-inhibition, and depletion of UHMK1 decreases expression of these factors, sensitizes melanoma cells to BRAF/MEK inhibition and promotes survival in vivo. Regarding the mechanisms of UHMK1 action, the authors claim that UHMK1 promotes the transport and translation of transcripts encoding metabolic proteins (UQCRC2, MTCO1, GLUT1). While the role in translation is clear, that in mRNA transport is, in my opinion, not supported by the current data. Altogether, this is a nice piece of work that highlights the role of mRNA-specific translation in metabolic reprogramming and resistance to BRAF inhibitors.

- Main comments:

1) The reproducibility of the high-throughput data seems low, as only 33% of tested hits could be validated using independent assays (Fig S2E). Validation does not seem to improve for genes bypassing more stringent thresholds (z-score). What is the real value, then, of the conclusions reached by global analysis of the screen?

2) Page 8 (lanes 243-247) and Fig S3F-G: It is difficult to reconcile inactivation of glycolysis if the validated targets GLUT1 and HK2 decrease at the mRNA level but their translation remains unaffected. I would rather interpret that the cell tries to preserve normal rates of glycolysis upon Vem treatment by using translational buffering. How does this fit with the statement that “concordant inactivation of transcription and selective mRNA translation pathways may achieve more rapid and complete inactivation of glycolysis following BRAF targeted therapy”? Can the authors provide specific genes of the glycolytic pathway whose translation (measured by polysome association) and protein levels (measured by Western blot) decrease upon Vem treatment?

3) Figure 5 and S5: Assessing lack of cross-contamination after nucleo-cytoplasmic separation using only RNA read-outs is dangerous because RNAs are transcribed from the nucleus and, thus, there is always a fraction present in that compartment. Please, confirm correct nucleo-cytoplasmic fractionation using Western blots against exclusive cytoplasmic (e.g. tubulin) and nuclear (e.g. histone) proteins. This is essential to hold the claims of Figure 5.

Furthermore, observed changes in mRNA distribution do not always correlate, and some statements in the text do not seem to be supported by the results shown in Figure 5:

- Fig 5D: UQCRC2 total mRNA levels do not change upon siUHMK1, whether or not Vem is applied. For the total levels to remain unchanged, any change in one compartment should be compensated by the opposite change in the other compartment. However, in the DMSO control there is an increase in cytoplasmic levels without concomitant decrease in nuclear levels. A similar situation happens upon Vem treatment: there is a decrease in cytoplasmic levels that is not accompanied by a significant increase in nuclear levels.

- Fig S5B: The authors claim that increased HK2 nuclear mRNA is observed, specifically in Vem+siUHMK1 treated cells. According to the figure, these changes are not significant. Even if they were, when assessing differences between Vem and Vem+ siUHMK1 cells, there is also an increase in cytoplasmic levels, ruling out a function for UHMK1 in nucleo-cytoplasmic distribution of this mRNA. The authors were careful of using the term “transport” rather than “export” in their statements in the main text, but this does not change the fact that only a role of UHMK1 in promoting export of specific mRNAs upon Vem treatment would fit with the requirement of this factor for increased expression of such transcripts (Fig 4H). Thus, even though some changes in nucleo-cytoplasmic “transport” are observed, the contribution of these to the roles of UHMK1 in supporting resistance to Vem treatment are unclear.

4) Figure S4C: Why the signal of V5 in the middle and right panels of the western blot does not correlate with the signal of UHMK1? How does p27 behave after over-expression of wt and kinase-

mutant UHMK1 in Cas9 control cells? Right now, this information is difficult to infer as there are separate blots with differing tubulin amounts.

Similarly, for Figure S5C-II, why the signal of V5 does not correspond to that of UHMK1? The levels of UHMK1 decrease upon Vem treatment, but this is not reflected in the V5 western blot.

- Other comments:

5) Figure 5: If there is no association of UHMK1 to the reported transcripts in the absence of Vem (Fig 5F), changes in the distribution of these mRNAs in the absence of VEM (Fig 5D) are bound to be indirect. Perhaps it would be easier for the reader if the authors would show first the association of UHMK1 to mRNAs, and then the effect in nucleo-cytoplasmic distribution as siUHMK1/siOPT in DMSO and Vem conditions with two graphics: total mRNA, and nucleo-cytoplasmic ratio. Then the authors can show the partition (i.e. current Figure 5C-E) as supplementary data.

6) Please, define the SMARTPool library and screen design: how many libraries (from the Tables it seems that 3 different libraries were used), how many genes per library, how many different siRNAs per gene, reference n< of the libraries (if not customized); how many siRNAs per cell in the transfections, how many replicates of the controls in each plate, etc.

7) Page 6 (lane 148): The authors mention that they find components of the eIF4F translation initiation complex. In Table S3, however, I could only find eIF4A3, which is not a component of eIF4F and is not involved in translation.

8) Fig 2I: Can the authors show a Western blot against VDAC? This is the transcript that changes the most in levels and, although the polysome association does not change upon Vem treatment, it is possible that polysomes are stacked and not translating this mRNA.

9) Page 5 (lane 116): "Cell number and viability were determined...". Should it better read: "Cell number was used as a proxy of viability, and was determined from nuclear DAPI staining..."

10) Figure 1C: Please indicate the names of genes rather than function, and explain blue vs red.

11) Why Fig 2D is slightly different than Fig S3D, while including the same comparison?

12) Figure 3E: Please, indicate the cell lines at the bottom.

13) Figure 4H: MTOC1, which is mentioned in the main text, is not present in this Western blot. Similarly, in Figure S6, the data on GLUT1 mentioned in the main text is not shown in the figure.

14) Figure S5C-I: I don't understand that the protocol includes an RNA-IP for 16 h, and this is not explained in the Materials and Methods or the figure legend. The pull-down (which is the same thing as RNA-IP) is performed 16h after what?

15) Please, mention in the introduction that UHMK1 is KIS.

16) It is RT-qPCR (not qRT-PCR). It is "these data are..." (not this data is...).

Reviewer #2 (Remarks to the Author);

The manuscript "Adaptive post-transcriptional reprogramming of metabolism limits response to targeted therapy in BRAF(V600E) melanoma" by Smith et al details the identification and characterization of UHMK1 in modulating the response to BRAF-inhibition.

Briefly, the authors have using a genome-wide siRNA screen set out to identify genes that improve

the response of BRAF(V600E)-mutant melanoma cells to a sub-lethal dose of vemurafenib. Guided by the screen hits, organizing network analyses, and interpolating known tenets of response to BRAFi, the authors somehow arrive at UHMK1 as a subject for characterization. Genetic loss of function studies based on siRNA transfection, supported by CRISPR/Cas9 targeted genome-editing, validates UHMK1 as a modulator of BRAF-i sensitivity both in vitro and in vivo. An excursion into the metabolic effects of UHMK1 suppression and response to BRAF-is conducted provide some correlates, but its interaction with certain mRNAs (UQCRC2 and GLUT1) links it's RNA binding and reducing these RNAs polysome association with a potential metabolic regulatory role.

This is indeed an interesting study with a potentially important hitherto uncovered role of UHMK1 in modulating BRAF-inhibitor responses. However, there are some outstanding concerns that should be addressed before proceeding with this work.

MAJOR CONCERNS:

How was UHMK1 rationally selected for characterization from the 622 hits; Based on L2F/p-value rank? It is not clear from the text or the figures, which leaves the reader wondering.

While the rescue of Cas9/sgUHMK1 edited cells using WT UHMK1 is good, the kinase-dead allele K54A is not. Specifically, the commonly used kinase-dead allele for the lysine residue substitution is arginine, thus the K54R allele should be used. Moreover, it needs to be shown that the allele is indeed expressed to the same extent as the WT rescue to be able to draw conclusions from this experiment. Alternatively, the catalytic aspartate could be mutated to alanine, which may help to maintain the overall structure.

On the same issue, it would be appropriate to mutate the RNA binding domain and measure association with UQCRC2/GLUT1 RNAs for the V5-IPs (Figure 5F).

It would also be interesting to examine the effects of over-expressing UHMK1 on modulating BRAFi effects both in vitro, and possibly in vivo. If there is no effect, its inhibitory effects on translation (Figure 6E) is probably not a key effect, but rather the kinase activity (see above).

For the in vivo experiments, it would be important to demonstrate the effects of sgUHMK1 potentiating the effects of DABRA+TRAM in an additional melanoma cell line.

Finally, why would the effects be limited to BRAF(V600E)-mutant melanomas; Is the expression of shUHMK1 limited to melanomas? To this end, what happens with response to MEK-i after sgUHMK1 genome editing in a NRAS-mutant melanoma cell line. Alternatively would a non-melanoma BRAF(V600E)-mutant cell line, i.e colon, NSCLC, or papillary thyroid cancer cell line show alternate response to BRAF-i?

Minor comments:

The concept of minimal residual disease (MRD) is derived based on outcome studies of patients with hematological cancers, but whether it indeed correlates (inversely) with treatment responses in solid cancers, and melanoma in particular, is largely unknown.

The sentence (page 9, lines 268-270) need to be rewritten to say that "there are no commercially available antibodies raised against UHMK1 that can detect the endogenous protein". This is if this indeed correct because there are perhaps a few; see SCBT sc-393605 and from other vendors.

MALME is not the name of a commonly known melanoma cell line...! Please, indicate whether MALME3M or MALME3 cells are used (figure 3D/F).

Figure 5F/PGC1A panel: use log10 on the Y-axis to display values across all IP samples.

Reviewer #3 (Remarks to the Author);

In this article Smith et al, employed a genome wide RNAi screen to identify gene expression and metabolic adaptations of BRAF-mutated melanoma to targeted therapy. This revealed that post-transcriptional regulation of expression of metabolic genes may play a major role in adaptation of BRAF-mutant melanoma cells to BRAFi. Specifically, the authors show that UHMK1 acts as a major regulator of nuclear export and translation of mRNAs encoding pivotal metabolic factors which appear to underpin development of resistance to BRAFi. Finally, Smith et al provide evidence that disrupting UHMK1 post-transcriptional network dramatically potentiates the effects of BRAFi. Collectively, I found that this is a strong study wherein a large body of data strongly support major conclusions of the manuscript. Moreover, considering the heightened interest in understanding the mechanisms of the development of drug resistance, I thought that this study is of a sufficient interest to the broad scientific audience. Notwithstanding the above mentioned strengths, some weaknesses were also noted, which if addressed, at least in my opinion, would further strengthen this already excellent study. My specific comments and concerns are provided below:

Major concerns:

-“ genome wide RNAi glycolysis screen” is somewhat of misnomer as it may suggest that the only genes that were screened are those involved in glycolysis. The authors are encouraged to consider rephrasing in this part of the manuscript. Moreover, the authors should provide a better rationale why glycolysis (lactate production) was used as a functional readout.

-In figures 2G, H and I it is hard to appreciate the modes of translational regulation as polysome profiling and total mRNA experiments were done at 24 and 40h whereas Western blots are done after 48h and 72h. Consolidation of time points appears to be warranted.

-In figure 2I VDAC1 Western blot is missing.

-The authors should perform experiments to exclude the contribution of changes in protein stability on the ETC component levels.

-Data in sup. figures 3F and G should be supported by monitoring the levels of GLUT1 and HK2 proteins.

-In figure 4H the authors should comment on different dynamics of induction of UQCRC2 and SDHB vs. NDUFB8 protein induction by BRAFi. Are these differential dynamics reflected in polysome profiles?

-Loading control used throughout the manuscript should also be added to figure 6B.

-Sup. figure S6C is missing control Western blots to confirm depletion of indicated proteins. In addition, how do the effects of UQCRC2 and ATP5A depletion on mitochondrial functions compare?

-Quantification appears to be warranted for the data presented in figures 6D-E.

-In figure 7D it seems that there is still some residual expression of UHMK1. Can authors comment on this?

-In many instances, controls which are normalized to e.g. 1 are missing SD or SEM values. These should be included.

Minor comments:

-The authors should indicate which mRNAs change translational efficiency, are congruently regulated

or buffered when referring to figure 2G in the text.

-In the discussion session, the authors should consider speculating regarding potential mechanisms of translational buffering in the context of adaptation to BRAFi and its functional consequences.

-“This data suggests that UHMK1 depletion may cooperate with BRAFi to elicit a double-hit on the glycolysis pathway, whereby both GLUT1 mRNA transcription and translation is concurrently switched off” This statement should perhaps be clarified, since the effects of UHMK1 have broad effects on metabolism (including OXPHOS).

-“Data” are plural. Consider changing “this data” to “these data”.

I hope that the authors find my assessment of their work constructive and with sufficient pathos.

Sincerely

I/Topisirovic

Reviewer #4 (Remarks to the Author):

The manuscript by Smith et al., described a potential mechanism of non-mutational adaptive reprogramming responsible for drug resistance in melanoma. The manuscript started from a genome-wide RNAi screen of potential target genes that may improve the efficiency of targeted therapy against BRAF in melanoma cells. The screening and gene expression profiling revealed that the mRNA of metabolic proteins, including glucose transporters and oxyphosphatases enzymes, were selectively transported and translated. Among those post-transcriptional regulators, UHMK1, a RNA binding kinase required for mitochondrial flexibility, was proposed to be responsible for selective mRNA translocation and translation, as well as metabolic remodeling in cell adaption. In addition, inactivation of UHMK1 improves the sensitivity of targeted therapy against BRAF, and delays drug resistance and disease recurrence. The authors have done tremendous amount of work including high-throughput screenings. Some parts in the manuscript are novel and worthy of publication. Here are some of my concerns.

Major points:

1. It is not clear how BRAF and/or MEK inhibition upregulates mRNA transport and translation genes' expression, including UHMK1's. Can authors find out the molecular mechanism?
2. To conclude that depletion of UHMK1 sensitizes BRAFi treatment. The combination treatment need to be further evaluated, in order to determine the exact effect (synergistic, additive or antagonistic). Otherwise, the effects of UHMK1 inhibition may be independent of BRAFi treatment. Actually, some of the data in Figure 3 seem to support the additive effect. Figure 7B has the same problem.
3. One of the most significant findings in this paper is that UHMK1 selectively regulates the mRNAs of certain metabolic or mitochondrial genes upon BRAFi, but it is unclear how UHMK1 obtains such preferences of mRNAs. Which motifs/domains of UHMK1 are responsible for the binding/interaction? Can authors comment?
4. It should be explained why there was no further change in VEM-treated melanoma after UHMK1 inhibition (Fig. 5AB). Does this mean the selective role for UHMK1 in mRNA transport does not play a role in BRAFi, and is therefore independent of BRAFi therapy response?
5. The cellular data and NOD model data look promising. However, UHMK1 is widely expressed in brain, endocrine tissues, liver, and the entire gastrointestinal tract (human protein atlas). Inhibition of adaptive mitochondrial metabolism by targeting UHMK1 may also impair the tolerance of normal

tissues requiring normal mitochondrial metabolism. Serious side effects may be expected. Can authors comment the translational value and approach of targeting UHMK1.

6. Most of the functional tests were carried in siUHMK1 condition, I would like to know whether supplementing UHMK1 in a UHMK1-low, BRAFi-sensitive tumor cells will enhance the cell viability and drug resistance upon VEM treatment.

Minor points

1. The title needs to be more specific. Although UHMK1 was mentioned as a paradigm, it should be included in the title, because the experiments related to UHMK1 occupied over 5/7 of the whole manuscript. Besides, there are almost a dozen regulatory mechanisms post-transcription, and mRNA transport and selective translation cannot represent them all.

2. Please provide significance (if there is any) between each two groups in the column chart of Fig. 5 C-F

3. Can author describe the difference between the two used cell lines WM266.4 and A375 in genetics and cell behaviors, and explain why A375 but not WM266.4 was used in NOD-NSG models.

We thank the reviewers for constructive comments and suggestions that we believe has improved the overall quality of the manuscript. We also apologise for the delay in completing the associated experiments due to a combination of technical difficulties encountered with reagent generation and restrictions associated with COVID-19.

Reviewers' comments:

Reviewer #1 (Remarks to the Author);

In this manuscript, Smith and colleagues show that post-transcriptional regulation by UHMK1 promotes adaptation of melanoma cells to BRAF inhibitors. UHMK1 promotes expression of factors involved in glucose and mitochondrial metabolism under BRAF-inhibition, and depletion of UHMK1 decreases expression of these factors, sensitizes melanoma cells to BRAF/MEK inhibition and promotes survival in vivo. Regarding the mechanisms of UHMK1 action, the authors claim that UHMK1 promotes the transport and translation of transcripts encoding metabolic proteins (UQCRC2, MTCO1, GLUT1). While the role in translation is clear, that in mRNA transport is, in my opinion, not supported by the current data. Altogether, this is a nice piece of work that highlights the role of mRNA-specific translation in metabolic reprogramming and resistance to BRAF inhibitors.

- Main comments:

1) The reproducibility of the high-throughput data seems low, as only 33% of tested hits could be validated using independent assays (Fig S2E). Validation does not seem to improve for genes bypassing more stringent thresholds (z-score). What is the real value, then, of the conclusions reached by global analysis of the screen?

The reported validation rate of 33% refers specifically to the RNA binding, transport and translation gene set identified as enriched in the screen. The overall validation rates for the screen are summarised in the table below, whereby genes with 2 or more individual siRNA duplexes that reproduce the primary screen phenotype are classified as a validated hit. We selected 400 hits from the primary screen for the deconvolution validation screen, based on their drug-enhancer effects, whereby the magnitude of difference between the control and drug values were weighted together with the individual Z-score of a gene for the drug viability or lactate/cell parameter. This was in order to identify genes with drug-specific effects. Notably, the validation rates for the primary outputs of the screen (drug enhancers in the context of viability and glycolysis at 60% and 53.25%, respectively) exceeds trends observed in other siRNA-based genome-scale screening studies which report anywhere from 10-40% validation rates (Brass et al. 2008, Simpson et al. 2008, Smith et al. 2010, Adamson et al. 2012, Falkenberg et al. 2014, Williams et al. 2017; references are appended to this document). We therefore consider that the global analysis of our screening dataset is up to current standards and thus we feel that it is of considerable value. However, we also acknowledge that genome wide screens are hypothesis generating tools, therefore all findings require subsequent validation, which is not dissimilar from other large scale analyses. This is emphasized in the revised version of the text. To this end, we have now discussed our hit selection, screen performance and validation strategy in more detail. We have also published an accompanying manuscript in Scientific Data (as suggested by the

Nat. Comms editor), which describes all technical aspects of the screen in detail and provides indicators of screen performance and comparison with other siRNA-based genome wide screens (Smith et al, Scientific Data, Volume 7, 2020). The screening dataset is deposited on Pubchem (<https://pubchem.ncbi.nlm.nih.gov/bioassay/1508588>) as a resource to allow other researchers to perform their own analyses.

Screen Output Parameters	siRNA DUPLEX VALIDATION					PHENOTYPE CONFIRMED
	0/4	1/4	2/4	3/4	4/4	
Control cell count (T48) bin	83	149	104	52	12	168
Percentage	21%	37%	26%	13%	3%	42%
Drug cell count (τ48) bin	61	133	123	65	18	206
Percentage	15.2%	33.2%	30.8%	16.2%	4.5%	51.5%
Control viability (deltaτ) bin	89	112	81	79	39	199
Percentage	22.2%	28%	20.2%	19.8%	9.8%	49.8%
Drug viability (deltaτ) bin	66	94	85	104	51	240
Percentage	16%	24%	21%	26%	13%	60%
Control lactate/cell bin	137	140	78	30	15	123
Percentage	34.2%	35%	19.5%	7.5%	3.8%	30.8%
Drug lactate/Control lactate per cell ratio < 0.55	183	4	121	71	21	213
Percentage	45%	1%	30.25%	17.75%	5.25%	53.25%

Table reproduced from our Scientific Data manuscript (Smith et al, Sci Data, 2020).

New text (Page 6, line 185):

“The major findings of the screen were confirmed using a secondary de-convolution screen, whereby four individual siRNA duplexes were assessed to determine reproducibility of gene knockdown phenotypes. Confirmed hits were defined as those with ≥ 2 siRNA duplexes reproducing the primary screen phenotype. Overall, validation rates for the screen exceeded those previously reported for comparable RNAi screens¹⁵, whereby 60% of genes were confirmed as “drug enhancers” in the context of viability, and 53.25% of genes were confirmed as “drug enhancers” in the context of glycolysis. Notably, 33% of the RNA transport and translation genes were validated by 2 or more duplexes (Figure S2E).”

2) Page 8 (lanes 243-247) and Fig S3F-G: It is difficult to reconcile inactivation of glycolysis if the validated targets GLUT1 and HK2 decrease at the mRNA level but their translation remains unaffected. I would rather interpret that the cell tries to preserve normal rates of glycolysis upon Vem treatment by using translational buffering. How does this fit with the statement that “concordant inactivation of transcription and selective mRNA translation pathways may achieve more rapid and complete inactivation of glycolysis following BRAF targeted therapy”? Can the authors provide specific genes of the glycolytic pathway whose

translation (measured by polysome association) and protein levels (measured by Western blot) decrease upon Vem treatment?

We agree with the interpretation that the cells attempt to sustain glycolysis upon Vem treatment via translational buffering. Based on this, we propose that attenuating translational buffering in combination with BRAFi will achieve maximal suppression of glycolysis. This tenet is supported by the experiment wherein inactivation of UHMK1 in conjunction with BRAFi results in stronger glycolytic suppression (Figure 3) and reduction of GLUT1 protein synthesis (Figure 5) which is reflective of the disruption of translational buffering. We have modified the text to reflect these findings.

Regarding additional glycolytic genes, we now provide evidence that HIF1alpha, that acts as a central factor in BRAF^{V600}- driven glycolysis (Parmenter et al, 2014) is suppressed at total mRNA and translation level 24-40hr BRAFi, which is reflected in downregulation of HIF1alpha protein (Figure S3E-G). These data strongly suggest that the observed effects on additional glycolytic genes are mediated via HIF1alpha.

New text (Page 9, line 313):

“Analysis of GLUT1 and HK2 revealed decreased total mRNA levels throughout Vem treatment (Figure S3E), however no change in polysome bound mRNA was observed (Figure S3Fi-ii). Analysis of GLUT1 and HK2 protein levels revealed a decrease following Vem treatment (Figure S3G), however this occurred at later timepoints, particularly for GLUT1. Although this does not fit the classical definition of translational buffering (characterized by alterations in mRNA levels that are not accompanied by changes in polysome occupancy nor protein levels), our data suggests that translational mechanisms may blunt rapid transcriptional inactivation of glycolysis pathway components in an attempt to preserve normal rates of glycolysis and facilitate cell survival during the acute response to BRAFi. We also assessed HIF1 α , that acts as a central factor in BRAF^{V600}- driven glycolysis¹, and here we observed congruent downregulation of total mRNA, polysome bound mRNA and protein levels (Figure S3E-G). Together these data raise the hypothesis that inactivation of adaptive reprogramming of mRNA translation may achieve more rapid and complete inactivation of the glycolysis pathway following BRAFi, which is consistent with reduced lactate production in the original RNAi screen when expression of genes encoding regulators of mRNA processing and translation were reduced.”

3) Figure 5 and S5: Assessing lack of cross-contamination after nucleo-cytoplasmic separation using only RNA read-outs is dangerous because RNAs are transcribed from the nucleus and, thus, there is always a fraction present in that compartment. Please, confirm correct nucleo-cytoplasmic fractionation using Western blots against exclusive cytoplasmic (e.g. tubulin) and nuclear (e.g. histone) proteins. This is essential to hold the claims of Figure 5.

We now provide western blot analysis of nuclear (Histone H3) and cytoplasmic (tubulin) proteins to further verify the nuclear-cytoplasmic fractionation (Figure S5Bii).

New text (Page 14, line 551):

“The fractionation was verified by monitoring levels of mRNA known to be enriched within the nucleus (metastasis associated lung adenocarcinoma transcript 1; MALAT1) and cytoplasm (ribosomal protein S14; RPS14)(Figure S5Bi), and western blot analysis of cytoplasmic (tubulin) and nuclear (Histone H3) specific proteins (Figure S5Bii).”

Furthermore, observed changes in mRNA distribution do not always correlate, and some statements in the text do not seem to be supported by the results shown in Figure 5:

- Fig 5D: UQCRC2 total mRNA levels do not change upon siUHMK1, whether or not Vem is applied. For the total levels to remain unchanged, any change in one compartment should be compensated by the opposite change in the other compartment. However, in the DMSO control there is an increase in cytoplasmic levels without concomitant decrease in nuclear levels. A similar situation happens upon Vem treatment: there is a decrease in cytoplasmic levels that is not accompanied by a significant increase in nuclear levels.

It is difficult to directly compare total mRNA with the individual nuclear and cytoplasm mRNA data because there are differences between levels of individual mRNA, such as UQCRC2, in the cytoplasm when compared to the nucleus (please see below). We also note that there is a corresponding increase in the nuclear UQCRC2 compartment, however this was just beyond statistical significance ($p = 0.06$). We have also now modified the presentation of the RNA binding and transport data as per your additional comments below. The individual compartment data is now presented as supplementary information (Figure S5C), and we also now show the nuclear/cytoplasm ratio for each individual transcript as suggested.

Relative UQCRC2 mRNA levels in nucleus compared to cytoplasm

- Fig S5B: The authors claim that increased HK2 nuclear mRNA is observed, specifically in Vem+siUHMK1 treated cells. According to the figure, these changes are not significant. Even if they were, when assessing differences between Vem and Vem+ siUHMK1 cells, there is also an increase in cytoplasmic levels, ruling out a function for UHMK1 in nucleo-cytoplasmic distribution of this mRNA.

We agree and have removed the HK2 analysis for clarity and focussed on the 2 major targets we have identified linked with UHMK1’s role in both mitochondrial oxidative metabolism (UQCRC2) and glycolysis (GLUT1) following BRAFi.

The authors were careful of using the term “transport” rather than “export” in their statements in the main text, but this does not change the fact that only a role of UHMK1 in

promoting export of specific mRNAs upon Vem treatment would fit with the requirement of this factor for increased expression of such transcripts (Fig 4H). Thus, even though some changes in nucleo-cytoplasmic “transport” are observed, the contribution of these to the roles of UHMK1 in supporting resistance to Vem treatment are unclear.

We were careful to use the more general term of transport rather than export, because UHMK1 has been shown to facilitate mRNA localisation throughout distinct regions of the cytoplasm, as well as facilitate export of mRNA from the nucleus to the cytoplasm. This is supported by our observation that UHMK1 translocates from the nucleus to the cytoplasm, and then also associates with sites of active translation (Figure 6D-E). These observations support a broader role for UHMK1 than just mRNA export, however we have modified our language when we discuss the nucleo-cytoplasmic fractionation data.

Regarding the role of the mRNA binding and export/transport function of UHMK1 in the BRAFi response, we now provide additional data showing that the UHM RNA binding domain is essential for interaction with UQCRC2 and GLUT1 mRNA (Figure S5D-E), and is required for UHMK1 mediated effects on BRAFi response, whereby the UHMK1- Δ RBD-V5 protein is not sufficient to rescue the enhanced Vem sensitivity observed in the UHMK1-gRNA cells (Figure 3G). Unfortunately, the method of isolating the UHMK1 protein from the polysome fractions involves protein precipitation using TCA, and therefore precludes analysis of mRNA transcripts that may be delivered to active sites of translation by the UHMK1 protein. It is therefore difficult to further demonstrate a causative role for the mRNA export/transport defects in BRAFi response and we acknowledge this in the manuscript.

Modified text (Page 14, Line 568):

“We were next interested in whether UHMK1 can regulate localization of these transcripts, therefore we assessed nuclear-cytoplasmic export of UQCRC2 and GLUT1 mRNA using RT-qPCR analysis of nuclear and cytoplasmic mRNA pools generated from subcellular fractionation. The fractionation was verified by monitoring levels of mRNA known to be enriched within the nucleus (metastasis associated lung adenocarcinoma transcript 1; MALAT1) and cytoplasm (ribosomal protein S14; RPS14)(Figure S5Bi), and western blot analysis of cytoplasmic (tubulin) and nuclear (Histone H3) specific proteins (Figure S5Bii). Notably, reduced cytoplasmic mRNA (UQCRC2) and increased nuclear mRNA (GLUT1) was observed in the Vem+siUHMK1 treated cells when compared to Vem alone (Figure S5C), culminating in a significant increase in the nuclear/cytoplasm mRNA ratio (Figure 5D). These data indicate UHMK1 depletion modifies localization of GLUT1 and UQCRC2 mRNA following BRAFi. In contrast, analysis of ATP5A transcripts revealed no significant change in mRNA distribution (Figure 5D & S5C), consistent with no evidence of a role for post-transcriptional mechanisms or UHMK1 in ATP5A regulation from previous analyses (Figure 2, 4 & 5C). Together, these observations demonstrate that UHMK1 can selectively associate with GLUT1 and UQCRC2 mRNA in the context of therapeutic adaptation in BRAF^{V600} melanoma cells treated with BRAFi, and this is associated with changes in their nuclear-cytoplasmic localization.

We were next interested in assessing the requirement of the different UHMK1 domains in regulation of these transcripts. To do this, we made use of our UHMK1 gRNA A375 cell line

panel expressing the UHMK1-V5, UHMK1-K54R-V5 (kinase dead), UHMK1- Δ RBD-V5 (lacking UHM RBD) and UHMK1-K54R- Δ RBD-V5 mutant proteins (Figure S4D). We first verified immunoprecipitation of these proteins (Figure S5D), then assessed association of the different proteins with UQCRC2, GLUT1 and ATP5A mRNA using RT-qPCR (Figure S5E). Because of the different levels of the mutant proteins (discussed above, see Figure S4), we normalized mRNA levels to input UHMK1 protein levels. Notably, interaction between UHMK1-V5 protein and UQCRC2 and GLUT1 mRNA following BRAFi was confirmed in these independently generated cells expressing UHMK1-V5, however these interactions were significantly reduced in cells expressing the UHMK1-K54R-V5, UHMK1- Δ RBD-V5 and UHMK1-K54R- Δ RBD-V5 mutant proteins (Figure S5E). Analysis of ATP5A revealed no significant association with any of the UHMK1 proteins. Interestingly, these data establish that UHMK1 requires both the UHM domain and its kinase activity to associate with GLUT1 and UQCRC2 mRNA following BRAFi.”

We also now provide extended analysis of the UHMK1 targets in mitochondrial metabolism and glycolysis, and demonstrate that UQCRC2 depletion phenocopies the effects of UHMK1 depletion on spare respiratory capacity and ATP production (Figure S6C-D), and GLUT1 depletion phenocopies the effects on glycolysis (Figure S6D), therefore supporting the idea they function in a common pathway in the context of BRAFi. Finally, we also now provide data demonstrating that the effects on glycolysis and mitochondrial metabolism underpin UHMK1’s role in BRAFi sensitivity by showing partial rescue of the proliferative, and complete rescue of the cell death, phenotypes when media is supplemented with electron acceptors (that have been shown to rescue proliferation in respiration deficient cells(Sullivan, Gui et al. 2015); Figure 4I-J).

We believe together these data strengthen the idea that association of UHMK1 to mRNA encoding metabolism proteins, and subsequent alterations in their localisation and protein synthesis, is a critical component of its ability to regulate adaptive responses to BRAFi.

4) Figure S4C: Why the signal of V5 in the middle and right panels of the western blot does not correlate with the signal of UHMK1? How does p27 behave after over-expression of wt and kinase-mutant UHMK1 in Cas9 control cells? Right now, this information is difficult to infer as there are separate blots with differing tubulin amounts.

Similarly, for Figure S5C-II, why the signal of V5 does not correspond to that of UHMK1? The levels of UHMK1 decrease upon Vem treatment, but this is not reflected in the V5 western blot.

We speculate the differences observed between the V5 and UHMK1 antibody in this cell line panel was mainly due to the poor quality of the UHMK1 antibody (see reviewer 2 comments for example westerns using the UHMK1 antibody), and due to this technical limitation, we have much higher confidence in the V5 antibody data over the UHMK1 antibody.

Nevertheless, we have now removed these data and replaced it with an extended panel of cell lines that include inactivation of the UHMK1 RNA binding domain to accommodate other reviewer comments (Figure S4D). Western blot analysis now allows comparison between the different cell lines (Figure S4F).

We have performed these experiments in our UHMK1 gRNA cells because expression of the UHMK1 mutant proteins do not function as a dominant negative in our cells, as reflected by no change in p27 levels upon expression of the K54R mutant in the parental A375 cells. The UHMK1 gRNA cells also allow rescue experiments and investigation of the requirements of the different domains of UHMK1 in the BRAFi response (Figure 3G).

- Other comments:

5) Figure 5: If there is no association of UHMK1 to the reported transcripts in the absence of Vem (Fig 5F), changes in the distribution of these mRNAs in the absence of VEM (Fig 5D) are bound to be indirect. Perhaps it would be easier for the reader if the authors would show first the association of UHMK1 to mRNAs, and then the effect in nucleo-cytoplasmic distribution as siUHMK1/siOPT in DMSO and Vem conditions with two graphics: total mRNA, and nucleo-cytoplasmic ratio. Then the authors can show the partition (i.e. current Figure 5C-E) as supplementary data.

We agree and have now shown the RNA-IP data for each transcript prior to the localisation data for clarity, and have also displayed the nuclear/cyto ratio for each transcript for UHMK1 KD cells relative to the siOPT controls in the primary figure, and now provide the individual nuclear and cytoplasmic compartmental analyses as supplementary data (See Figure 5 and Figure S5).

6) Please, define the SMARTpool library and screen design: how many libraries (from the Tables it seems that 3 different libraries were used), how many genes per library, how many different siRNAs per gene, reference n< of the libraries (if not customized); how many siRNAs per cell in the transfections, how many replicates of the controls in each plate, etc.

We have now modified the screen methods to include more details describing the genome wide SMARTpool library used for the primary screen and number of control wells used per plate. As discussed above, we have also published an accompanying manuscript (as suggested by the editor) which provides a comprehensive description of the screen method and associated analyses (Smith et al, Scientific Data, Volume 7, 2020).

New text in supplementary information (Page 4, Line 215):

“The Dharmacon human siGENOME SMARTpool library (RefSeq27; Dharmacon RNAi Technologies, Horizon Discovery) was used for the screen. This library contains 18,120 SMARTpool reagents (4x individual siRNA duplexes targeting each gene per SMARTpool) targeting each gene in the human genome. The library was arrayed across 58x library plates and screened in 384-well format within the Victorian Centre for Functional Genomics (VCFG, Peter MacCallum Cancer Centre, Australia).”

New text in supplementary information (Page 5, Line 242):

“The transfection was performed 24hrs post cell seeding using a Caliper Sciclone ALH3000 liquid handling robot (Perkin Elmer, USA), RNAi MAX transfection lipid (Invitrogen, 0.03µL per well in 37.5µL) and siGENOME SMARTpool siRNA at a final concentration of 40nM. siOPT (D-001810-10-10) was used as the non-targeting control (16x wells per plate), siPLK1 (M-003290-01-0005) was used as a cell viability positive control (8x wells per plate), and siPDK1

(M-005019-00-0005) was used as a lactate assay positive control (8x wells per plate).”

7) Page 6 (lane 148): The authors mention that they find components of the eIF4F translation initiation complex. In Table S3, however, I could only find eIF4A3, which is not a component of eIF4F and is not involved in translation.

We have adjusted the manuscript to correct this error (Page 6, line 178).

8) Fig 2I: Can the authors show a Western blot against VDAC? This is the transcript that changes the most in levels and, although the polysome association does not change upon Vem treatment, it is possible that polysomes are stacked and not translating this mRNA.

We have now provided western blot analysis showing no change in VDAC1 protein levels throughout the Vem treatment time course (Figure 2I), confirming translational buffering of VDAC1.

9) Page 5 (lane 116): “Cell number and viability were determined...”. Should it better read: “Cell number was used as a proxy of viability, and was determined from nuclear DAPI staining...”

Our screen contained 3 independent arms in order to quantify cell viability in its design; a T0 cell number plate (fixed and stained prior to treatment), a T48 control plate (DMSO treated for 48hrs) and T48 drug plate (Vem treated for 48hrs). Viability was calculated by subtracting the T0 count from the T48 count, generating the parameter “deltaT” equating for change in cell number during drug treatment for both the control and drug treated conditions. A negative value indicates cell death. We have edited our text for more clarity, and have also modified our screen schematic in Figure 1A.

Main text (Page 5, line 137):

“Cell number was determined from nuclear DAPI staining using automated image analysis and change in cell number throughout the drug treatment was used as a proxy of viability, whereby negative values indicate cell death (see methods)”.

10) Figure 1C: Please indicate the names of genes rather than function, and explain blue vs red.

Figure 1C is a plot displaying Gene Ontology and KEGG pathway enrichment data as determined using DAVID – not data for individual genes. The individual genes comprising each enriched pathway/annotation are listed in Table S3. We have now modified the figure legend to explain blue (previously associated pathways with BRAFi response/resistance) and red (annotations associated with RNA binding/transport/translation)(see page 22, line 1066).

11) Why Fig 2D is slightly different than Fig S3D, while including the same comparison?

Data presented in Figure 2D is derived from the KEGG oxidative phosphorylation gene set and data presented in Figure S3D is derived from the Hallmark oxidative phosphorylation

gene set. Because these two gene sets do not completely overlap (ie they contain different genes associated with oxidative phosphorylation), we initially included both for robustness. However, in order to incorporate additional data into the supplementary figure as requested by another reviewer, we have now removed the figure showing the enrichment profile for the Hallmark OXPHOS gene set.

12) Figure 3E: Please, indicate the cell lines at the bottom.

We apologise for this error and have now included the cell lines in Figure 3E.

13) Figure 4H: MTOC1, which is mentioned in the main text, is not present in this Western blot. Similarly, in Figure S6, the data on GLUT1 mentioned in the main text is not shown in the figure.

Figure 4H: We have corrected this error when referring to this figure in the text.

Figure S6: We referred the reader to our previous analysis of GLUT1 depletion in BRAFi response (Parmenter et al, 2014), however we have now repeated this analysis for clarity and have included the new data in Figure S6.

14) Figure S5C-I: I don't understand that the protocol includes an RNA-IP for 16 h, and this is not explained in the Materials and Methods or the figure legend. The pull-down (which is the same thing as RNA-IP) is performed 16h after what?

We performed the RNA-IP overnight for 16hr and have adjusted Figure S5Ci for clarity (now FigS5A).

15) Please, mention in the introduction that UHMK1 is KIS.

We have adjusted the text in our manuscript accordingly.

Main text (Page 4, line 110):

“This translational reprogramming requires the RNA binding kinase UHMK1 (also known as Kinase Interacting with Stathmin, KIS) that regulates mitochondrial flexibility to control BRAFi sensitivity, and controls the abundance of metabolic proteins through the export and translation of the mRNA that encode them.”

16) It is RT-qPCR (not qRT-PCR). It is “these data are...” (not this data is...).

We have changed all abbreviations to RT-qPCR.

Regarding “these data”, we have adjusted the text in our manuscript accordingly.

Reviewer #2 (Remarks to the Author);

The manuscript “Adaptive post-transcriptional reprogramming of metabolism limits

response to targeted therapy in BRAF(V600E) melanoma” by Smith et al details the identification and characterization of UHMK1 in modulating the response to BRAF-inhibition.

Briefly, the authors have using a genome-wide siRNA screen set out to identify genes that improve the response of BRAF(V600E)-mutant melanoma cells to a sub-lethal dose of vemurafenib. Guided by the screen hits, organizing network analyses, and interpolating known tenets of response to BRAFi, the authors somehow arrive at UHMK1 as a subject for characterization. Genetic loss of function studies based on siRNA transfection, supported by CRISPR/Cas9 targeted genome-editing, validates UHMK1 as a modulator of BRAF-i sensitivity both in vitro and in vivo. An excursion into the metabolic effects of UHMK1 suppression and response to BRAF-is conducted provide some correlates, but its interaction with certain mRNAs (UQCRC2 and GLUT1) links it’s RNA binding and reducing these RNAs polysome association with a potential metabolic regulatory role.

This is indeed an interesting study with a potentially important hitherto uncovered role of UHMK1 in modulating BRAF-inhibitor responses. However, there are some outstanding concerns that should be addressed before proceeding with this work.

MAJOR CONCERNS:

How was UHMK1 rationally selected for characterization from the 622 hits; Based on L2F/p-value rank? It is not clear from the text or the figures, which leaves the reader wondering.

We employed a systems-biology approach in order to identify lead candidates from our functional screen. Because the strength of phenotype observed in genetic screens can be a direct function of efficacy of reagents, we chose to employ a more comprehensive approach for hit selection than just relying on FC and p-value cutoffs of individual genes, such as pathway enrichment and network analyses.

Given the primary goal of the screen was to identify genes whose depletion enhanced the effect of the drug on viability and glycolysis, we focused on these genes for the deconvolution validation screen. We selected 400 of these genes for validation based on a range of parameters: 300 top ranked drug enhancers for glycolysis and 50 top ranked drug enhancers for viability, where enhancer hits (as defined in our methods) were ranked based on FC and Z-score values for the viability and lactate/cell parameters in the drug arm of the screen; and 50 additional genes based on pathway enrichment analysis and *a priori* knowledge of genes of potential interest to the underlying biology of the screen.

UHMK1 was amongst these top ranked enhancer hits and validated most strongly in the deconvolution screen for both viability and glycolysis, specifically in the presence of Vem (Figure S2). Moreover, based on UHMK1’s unusual characteristics (only known kinase that contains a classical RNA binding domain), incorporation into the mRNA transport and translation hub in our network analysis, previous implications in mediating cellular plasticity phenotypes, and potential clinical actionability given it is a kinase; we were particularly interested in exploring this gene further.

We have now modified the text in the manuscript to more clearly articulate these points.

Main text (Page 10, line 367):

“Our systematic functional and transcriptomic approaches supported a role for selective RNA processing and translation pathways in metabolic response to BRAFi. Among the RNA processing proteins identified in our screen, U2AF homology motif (UHM) kinase 1 (UHMK1, also known as Kinase interacting with Stathmin, KIS) was of most interest given it validated strongly in the deconvolution screen, and was also part of the RNA transport and translation hub connecting both the glycolysis and viability networks. UHMK1 is the only known kinase to contain a classical RNA recognition motif (the UHM domain), raising the hypothesis that it may function as a hub linking cell signaling and RNA processing, and moreover, UHMK1 regulates neuronal plasticity and adaptation via selective RNA transport and translation (Cambray, Pedraza et al. 2009, Pedraza, Ortiz et al. 2014) thus we hypothesized it may facilitate adaptive cellular reprogramming in the context of adaptation following BRAFi.”

We have also published an accompanying manuscript (as suggested by the editor) which provides a comprehensive description of the screen method and associated analyses (Smith et al, Scientific Data, volume 7, 2020).

While the rescue of Cas9/sgUHMK1 edited cells using WT UHMK1 is good, the kinase-dead allele K54A is not. Specifically, the commonly used kinase-dead allele for the lysine residue substitution is arginine, thus the K54R allele should be used. Moreover, it needs to be shown that the allele is indeed expressed to the same extent as the WT rescue to be able to draw conclusions from this experiment. Alternatively, the catalytic aspartate could be mutated to alanine, which may help to maintain the overall structure.

The mutation we introduced into the kinase domain was K54R, not K54A as stated in the manuscript. We apologise for this mistake and thank the reviewer for identifying the error. We have now generated a new cell line panel to accommodate interrogation of the RNA binding domain to address your comment below, and that of other reviewers. We used the MSCV-GFP vector allowing GFP-based sorting of the cells to normalise expression levels. In these cell lines, the wild type UHMK1-V5 protein is expressed at similar levels as the K54R protein (Figure S4E-F), and similar to our previous data obtained from expression of UHMK1 from the pLX304 vector, the K54R protein cannot rescue the UHMK1 KO phenotype (Figure 3G). We also now include the RNA binding domain mutants (Δ RBD-V5 and K54R- Δ RBD-V5) in the rescue experiments and demonstrate the UHM domain is also required for UHMK1 mediated regulation of BRAFi sensitivity (Figure 3G). Unfortunately, there are caveats with this part of the experiment due to higher levels of the Δ RBD-V5 and K54R- Δ RBD-V5 mutant proteins compared to the UHMK1-V5 and K54R-V5 proteins (Figure S4F). Because equivalent mRNA expression of these constructs was achieved (Figure S4E) we conclude these differences in protein levels were due to changes in protein stability induced by modification of the RBD. Indeed, we first deleted the entire UHM domain from the UHMK1 protein and could not obtain any stable expression of this mutant protein, further suggesting modifications to the UHM domain in UHMK1 changes its stability. However, because expression of these proteins even to higher levels than the wildtype protein was not sufficient to rescue the effect of UHMK1 gRNA on BRAFi sensitivity, we conclude that UHMK1 also requires the RNA binding domain for regulation of BRAFi sensitivity.

On the same issue, it would be appropriate to mutate the RNA binding domain and measure association with UQCRC2/GLUT1 RNAs for the V5-IPs (Figure 5F).

As discussed above, we have now generated a panel of UHMK1 mutant proteins, either lacking kinase activity or the UHM domain, alone or in combination (Figure S4). We have performed RNA-IP experiments to assess the UHMK1 domains required for association to UQCRC2 and GLUT1 (Figure S5D-E). Because of the different levels of the mutant proteins, we normalized mRNA levels to input UHMK1 protein levels, and these data demonstrate that UHMK1 requires both the UHM domain and its kinase activity in order to associate with UQCRC2 and GLUT1 mRNA (Figure S5E). Due to the issues encountered with changes in stability of the Δ RBD mutant proteins we have only included these data as supplementary information, however we believe these experiments are still sufficient to assess the role of the UHM domain in the RNA interactions. We also discuss limitation of this assay (i.e. unequal expression levels of mutants) in the revised version of the text.

Main text (Page 14, line 568):

“We were next interested in assessing the requirement of the different UHMK1 domains in regulation of these transcripts. To do this, we made use of our UHMK1 gRNA A375 cell line panel expressing the UHMK1-V5, UHMK1-K54R-V5 (kinase dead), UHMK1- Δ RBD-V5 (lacking UHM RBD) and UHMK1-K54R- Δ RBD-V5 mutant proteins (Figure S4D). We first verified immunoprecipitation of these proteins (Figure S5D), then assessed association of the different proteins with UQCRC2, GLUT1 and ATP5A mRNA using RT-qPCR (Figure S5E). Because of the different levels of the mutant proteins, we normalized mRNA levels to input UHMK1 protein levels. Notably, interaction between UHMK1-V5 protein and UQCRC2 and GLUT1 mRNA following BRAFi was confirmed in these independently generated cells expressing UHMK1-V5, however these interactions were significantly reduced in cells expressing the UHMK1-K54R-V5, UHMK1- Δ RBD-V5 and UHMK1-K54R- Δ RBD-V5 mutant proteins (Figure S5E). Analysis of ATP5A revealed no significant association with any of the UHMK1 proteins. Interestingly, these data establish that UHMK1 requires both the UHM domain and its kinase activity to associate with GLUT1 and UQCRC2 mRNA following BRAFi.”

It would also be interesting to examine the effects of over-expressing UHMK1 on modulating BRAFi effects both in vitro, and possibly in vivo. If there is no effect, its inhibitory effects on translation (Figure 6E) is probably not a key effect, but rather the kinase activity (see above).

We have assessed over-expression of UHMK1-V5 in our melanoma cells and we did not observe major effects to BRAFi sensitivity. We note however that UHMK1 is among the most highly expressed genes in our panel of 71 melanoma cell lines (within the top 90th percentile, as assessed by microarray expression analysis). This suggests that the high levels of endogenous UHMK1 are saturating and mitigate the additional effects of overexpression of exogenous protein. We also now provide additional data demonstrating that UHMK1 requires both its UHM RNA binding domain and its kinase activity to mediate sensitivity to BRAFi (see above comments).

For the in vivo experiments, it would be important to demonstrate the effects of sgUHMK1

potentiating the effects of DABRA+TRAM in an additional melanoma cell line.

We agree it is of interest to assess the generalizability of our *in vivo* studies by assessing an additional orthotopic model. Prior to COVID19 related complications, we had generated stably expressing sgUHMK1 WM266.4 cells (our initial attempts at single cell cloning the knockout cells were unsuccessful therefore we settled on a stable knockout, pooled cell population approach in this cell line). We experienced a prolonged and complete lab lockdown at the start of the pandemic in Victoria, Australia, after which, we had reduced access to the laboratory at 30-50% capacity throughout Melbourne's subsequent long lockdown during our second wave of the pandemic that extended for over 100 days. Due to these circumstances, there were restrictions on long term *in vivo* experiments. Unfortunately, because the growth kinetics of WM266.4 tumours in NSG is slower than A375 tumours, these experiments require a minimum of ~5 months to achieve a meaningful endpoint and we have therefore been unable to proceed with these experiments. We however believe that this does not significantly detract from the overall quality or interpretation of the data presented in the manuscript.

Finally, why would the effects be limited to BRAF(V600E)-mutant melanomas; Is the expression of shUHMK1 limited to melanomas? To this end, what happens with response to MEK-i after sgUHMK1 genome editing in a NRAS-mutant melanoma cell line. Alternatively would a non-melanoma BRAF(V600E)-mutant cell line, i.e colon, NSCLC, or papillary thyroid cancer cell line show alternate response to BRAF-i?

In our opinion, one of the most exciting possibilities emerging from our work is that activation of an adaptive UHMK1-mediated mRNA translation mechanism is not limited to inhibition of oncogenic BRAF in melanoma cells. We have now tested UHMK1 depletion in combination with trametinib in NRAS mutant melanoma, and observed that UHMK1 depletion enhances anti-proliferative responses in two NRAS mutant melanoma cell lines (D04M1 and IPC298). These data are now included in Figure 7G.

Although of high interest, we believe assessment of UHMK1 activity in additional cancer models is beyond the scope of the current article.

New main text (Page 17, line 743):

"Finally, we assessed the effectiveness of UHMK1 depletion in combination with the MEK inhibitor trametinib (tram) in the setting of NRAS mutant melanoma. The siUHMK1+Tram combination resulted in more robust growth inhibition in multiple NRAS mutant melanoma cell lines (Figure 7G) providing evidence that UHMK1 depletion can also play a role in MAPK targeted therapy response in the setting of a different oncogenic driver."

Minor comments:

The concept of minimal residual disease (MRD) is derived based on outcome studies of patients with hematological cancers, but whether it indeed correlates (inversely) with treatment responses in solid cancers, and melanoma in particular, is largely unknown.

The idea of residual disease posing a clinical challenge in a broad range of solid and hematological cancers following treatment with a range of therapies, including targeted therapies, has been recently reviewed (Marine et al, Nature Reviews Cancer, 2020). We also refer the reviewer to Rambow et al (Cell, 2018) where they show a very nice description of this concept in preclinical and clinical models of melanoma using single cell RNA sequencing which clearly demonstrates residual melanoma cells following MAPK pathway targeted therapy. The major point we wanted to convey with this statement was that in many cancers, relapse occurs because available therapeutics often leave behind residual cancer cells that can then go on to drive therapy resistance and relapse (whether by genetic or non-genetic mechanisms). We are interested in understanding the processes that allow these residual cells to survive so that we may target them and prevent the relapse, and this is a central concept underlying the research presented in this manuscript.

The sentence (page 9, lines 268-270) need to be rewritten to say that “there are no commercially available antibodies raised against UHMK1 that can detect the endogenous protein”. This is if this indeed correct because there are perhaps a few; see SCBT sc-393605 and from other vendors.

Although multiple UHMK1 antibodies are commercially available, we found that neither Santa Cruz sc-393605 nor Protein Tech 11624-1-AP-2 specifically detected a 47kDa band that would correspond to endogenous human UHMK1 in our melanoma cells (see below). Santa Cruz sc-393605 was however sufficient to detect exogenous expression of UHMK1 in our cells, however we have now removed all data using this antibody due to discrepancies with the V5 antibody which is a much more reliable and robust reagent.

We have adjusted the text in the manuscript to more accurately reflect this (Page 10, line 403):

“Because the available UHMK1 antibodies do not specifically detect the endogenous human protein in our melanoma cells, we also confirmed increased levels of its key target p27, which is degraded following phosphorylation by UHMK1 ³⁰.”

MALME is not the name of a commonly known melanoma cell line...! Please, indicate whether MALME3M or MALME3 cells are used (figure 3D/F).

We used the MALME3 cell line and have now corrected this error, and thank the reviewer for identifying this mistake.

Reviewer #3 (Remarks to the Author);

In this article Smith et al, employed a genome wide RNAi screen to identify gene expression and metabolic adaptations of BRAF-mutated melanoma to targeted therapy. This revealed that post-transcriptional regulation of expression of metabolic genes may play a major role in adaptation of BRAF-mutant melanoma cells to BRAFi. Specifically, the authors show that UHMK1 acts as a major regulator of nuclear export and translation of mRNAs encoding pivotal metabolic factors which appear to underpin development of resistance to BRAFi. Finally, Smith et al provide evidence that disrupting UHMK1 post-transcriptional network dramatically potentiates the effects of BRAFi. Collectively, I found that this is a strong study wherein a large body of data strongly support major conclusions of the manuscript. Moreover, considering the heightened interest in understanding the mechanisms of the development of drug resistance, I thought that this study is of a sufficient interest to the broad scientific audience. Notwithstanding the above mentioned strengths, some weaknesses were also noted, which if addressed, at least in my opinion, would further strengthen this already excellent study. My specific comments and concerns are provided below:

Major concerns:

-“ genome wide RNAi glycolysis screen” is somewhat of misnomer as it may suggest that the only genes that were screened are those involved in glycolysis. The authors are encouraged to consider rephrasing in this part of the manuscript. Moreover, the authors should provide a better rationale why glycolysis (lactate production) was used as a functional readout.

We agree, and have adjusted the text in our manuscript accordingly (Page 4, line 124).

“To identify regulators of metabolic response following treatment with oncogene targeted therapy, we performed a genome wide RNAi screen using BRAF^{V600} melanoma cells treated with the BRAF inhibitor (BRAFi) vemurafenib (Vem) as a paradigm (Figure 1A)¹⁶. We assessed glycolysis in our primary screen based on the observation that glycolytic response confers BRAFi sensitivity in pre-clinical⁹ and clinical studies¹⁰.”

-In figures 2G, H and I it is hard to appreciate the modes of translational regulation as polysome profiling and total mRNA experiments were done at 24 and 40h whereas Western blots are done after 48h and 72h. Consolidation of time points appears to be warranted.

We agree and have now provided western blots to allow consolidation of time points – please see modified panel in Figure 2I. These new data reveal that OXPHOS proteins NDUFB8, SDHB and UQCRC2 start to increase following 40h treatment with Vem. VDAC1 and ATP5A proteins remain constant throughout the extended Vem treatment time course.

-In figure 2I VDAC1 Western blot is missing.

We now provide western blots for VDAC1 in Figure 2I. These data show no change in VDAC1 protein levels throughout the Vem treatment time course. Together with analysis of total and polysome-bound mRNA levels, these data confirm translational buffering of VDAC1 following BRAFi.

New main text (Page 9, line 295):

“VDAC1 (voltage dependent anion channel 1) was translationally buffered, whereby no change in polysome bound mRNA or protein levels were observed, despite a significant reduction in total mRNA levels (Figure 2G-I).”

-The authors should perform experiments to exclude the contribution of changes in protein stability on the ETC component levels.

We agree and thank the reviewer for this suggestion. To directly address the role of mRNA translation in accumulation of OXPHOS proteins following BRAFi, we have assessed OXPHOS protein levels throughout a Vem treatment time course, +/- treatment with the mRNA translation inhibitor cycloheximide (new data panel Figure 2J). These data demonstrate that the increase in UQCRC2, SDHB and NDUFB8 OXPHOS proteins following BRAFi is almost completely dependent on mRNA translation, thus ruling out a major role for protein stability in regulation of these specific OXPHOS proteins. Notably we saw no effect on ATP5A protein levels following CHX treatment indicating there is a dominant role for stability in regulation of this OXPHOS complex V protein during the acute BRAFi response.

New main text (Page 9, line 302):

“Notably, treatment with the mRNA translation inhibitor cycloheximide (CHX) obliterated the BRAFi-induced increase in UQCRC2, SDHB and NDUFB8 OXPHOS proteins (Figure 2J), directly confirming a role for mRNA translation in OXPHOS protein accumulation following BRAFi. In contrast, CHX did not affect ATP5A protein levels thus suggesting that ATP5A is regulated at the level of protein stability during the acute response to BRAFi in melanoma cells.”

-Data in sup. figures 3F and G should be supported by monitoring the levels of GLUT1 and HK2 proteins.

We now provide western blot analysis of both GLUT1 and HK2 proteins (Figure S3G). These data show a decrease in GLUT1 and HK2 proteins, however this occurs later in the drug treatment particularly for GLUT1. We have also clarified our discussion of these data to acknowledge that this mRNA and protein expression profile does not fit the classical definition of translational buffering, but rather focus on these data potentially suggesting that adaptive mRNA translation mechanisms may blunt the transcriptional inactivation of glycolysis observed following BRAFi. This is consistent with our functional screening data, and subsequent analysis that revealed enhanced suppression of glycolysis (ECAR and lactate) and GLUT1 protein synthesis following inhibition of mRNA processing pathways mediated by UHMK1.

New text (Page 9, line 313):

“Analysis of GLUT1 and HK2 revealed decreased total mRNA levels throughout Vem treatment (Figure S3E), however no change in polysome bound mRNA was observed (Figure S3Fi-ii). Analysis of GLUT1 and HK2 protein levels revealed a decrease following Vem treatment (Figure S3G), however this occurred at later timepoints, particularly for GLUT1. Although this does not fit the classical definition of translational buffering (characterized by alterations in mRNA levels that are not accompanied by changes in polysome occupancy nor protein levels), our data suggests that translational mechanisms may blunt rapid transcriptional inactivation of glycolysis pathway components in an attempt to preserve normal rates of glycolysis and facilitate cell survival during the acute response to BRAFi. We also assessed HIF1 α , that acts as a central factor in BRAF^{V600}- driven glycolysis¹, and here we observed congruent downregulation of total mRNA, polysome bound mRNA and protein levels (Figure S3E-G). Together these data raise the hypothesis that inactivation of adaptive reprogramming of mRNA translation may achieve more rapid and complete inactivation of the glycolysis pathway following BRAFi, which is consistent with reduced lactate production in the original RNAi screen when expression of genes encoding regulators of mRNA processing and translation were reduced.”

-In figure 4H the authors should comment on different dynamics of induction of UQCRC2 and SDHB vs. NDUFB8 protein induction by BRAFi. Are these differential dynamics reflected in polysome profiles?

We do not see a consistent difference in the dynamics of induction of NDUFB8 (CI), SDHB (CII) and UQCRC2 (CIII) across our experiments (Figure 2I+J and Figure 4H), and there are no obvious differences in translational dynamics in the polysome profiles.

-Loading control used throughout the manuscript should also be added to figure 6B.

Because ATP5A protein levels do not change following BRAF inhibition (Fig 2I), we used ATP5A as loading control for this experiment. This is because loading controls used for other experiments in the manuscript run at the same size as proteins assessed using the OXPHOS antibody cocktail (tubulin=50kDa; actin=45kDa; ATP5A=55kDa; and UQCRC2=48kDa) and avoids stripping and re-probing for loading control.

-Sup. figure S6C is missing control Western blots to confirm depletion of indicated proteins. In addition, how do the effects of UQCRC2 and ATP5A depletion on mitochondrial functions compare?

We now provide western blots to confirm knockdown of UQCRC2 and ATP5A at the protein level (Figure S6A). We also now include new analyses of GLUT1 knockdown to avoid the complicated reference to our previous manuscript (Figure S6). We have also extended analysis of these UHMK1 mRNA targets using Seahorse analysis. We observed that depletion of UQCRC2 significantly reduces spare respiratory capacity and ATP production in combination with BRAFi (Figure S6), mimicking UHMK1 knockdown (Figure 4A, D, E), however ATP5A does not (Figure S6). We do note that we did not achieve strong knockdown of ATP5A which may be due to the large role of protein stability in regulation of this protein

(Figure 2J). No effect of UQCRC2 or ATP5A depletion was observed on ECAR (glycolysis), however we saw a significant reduction in siGLUT1+Vem treated cells compared to Vem alone, mimicking the effects of UHMK1 on glycolysis.

New main text (Page 15, line 638):

“Linking these observations to UHMK1’s role in cellular responses to BRAFi, depletion of UQCRC2 and GLUT1 phenocopies UHMK1 knockdown whereby enhanced sensitivity to BRAFi was observed in cell proliferation assays (Figure S6A-B). However, in contrast, no effect on Vem sensitivity was observed in the context of Vem+siATP5A treated cells (Figure S6A-B). We do note that we did not achieve strong knockdown of ATP5A which may be due to the large role of protein stability in regulation of this protein in our cells (Figure 2J). Notably, a significant decrease in both SRC (Figure S6C) and ATP production (Figure S6D) were also observed in Vem+siUQCRC2 treated cells, but not in ATP5A depleted cells. With regard to glycolysis, whilst depletion of UQCRC2 or ATP5A had no significant effect on ECAR either alone or in combination with Vem, depletion of GLUT1 significantly enhanced the effects of Vem on glycolysis (Figure S6E). Together, these data support a model whereby UHMK1 regulates glycolysis and mitochondrial metabolism following BRAFi via translational regulation of key pathway components including UQCRC2 and GLUT1.”

We also now provide additional new data that more directly links the metabolic defects observed in siUHMK1+BRAFi treated cells with the enhanced anti-proliferative effects and increased cell death. We show that supplementation of growth media with the electron acceptors pyruvate and α -ketobutyrate, that have been shown to rescue proliferation in respiration deficient cells (Sullivan et al, Cell, 2015) partially rescue the anti-proliferative, effects of UHMK1 knockdown in BRAFi treated cells, and completely rescue increased induction of cell death (Figure 4I-J). These observations therefore demonstrate that UHMK1 regulates adaptive reprogramming of metabolism to limit response to BRAFi.

New main text (Page 12, line 468):

“In order to establish whether these metabolic defects underpin the enhanced anti-proliferative and cell death responses to BRAFi in UHMK1 depleted cells, we supplemented growth media with the electron acceptors pyruvate and α -ketobutyrate, which have been shown to rescue proliferation in respiration deficient cells³². Although pyruvate and α -ketobutyrate only partially rescue the anti-proliferative effects of the siUHMK1+BRAFi combination (Figure 4I), a near complete rescue of cell death was observed (Figure 4J), demonstrating that defects in metabolism in siUHMK1+Vem treated cells underpin enhanced BRAFi sensitivity. Together these data suggest that UHMK1 is required for adaptive reprogramming of oxidative metabolism following BRAFi, and this reprogramming limits response to BRAFi.”

-Quantification appears to be warranted for the data presented in figures 6D-E.

Quantification of western blot analysis is provided in Figure 6E, expressed as a ratio of each indicated protein in sub-polysome versus heavy polysome fractions, as determined by densitometry analysis – see source data file for raw images used for these analyses.

-In figure 7D it seems that there is still some residual expression of UHMK1. Can authors comment on this?

mRNA expression measured using qRT-PCR is not the best biomarker for CRISPR genome editing, however we included these data as additional evidence of UHMK1 depletion because we have issues with commercial antibodies for the endogenous human UHMK1 protein in our melanoma cells (see reviewer 2 comments for more details). We speculate that residual UHMK1 mRNA expression may result from genome editing which can cause unpredictable truncations in the gene and thereby allow residual expression of truncated mRNA transcripts that can be detected using qRT-PCR. However, given we only see very low levels of residual mRNA, and increased levels of p27 protein in these cells, which is a reliable measure of UHMK1 protein levels and activity, we suggest that we have significantly reduced UHMK1 protein in these cells. We have now adjusted our language when discussing these data and refer to these cells as UHMK1 gRNA cells, as opposed to UHMK1 knockout cells.

-In many instances, controls which are normalized to e.g. 1 are missing SD or SEM values. These should be included.

In most of our experiments we calculate the fold change for individual experiments, normalised to endogenous experimental controls, then average the mean for each group across 3x biological replicates. Because we calculate the SEM in biological experiments, there is no variation in the controls, which are all 1. We provide all raw data in the associated source data files for all primary and supplementary figures therefore readers can assess the technical variation within each experiment, and the variation between control values of each independent biological replicate.

Minor comments:

-The authors should indicate which mRNAs change translational efficiency, are congruently regulated or buffered when referring to figure 2G in the text.

We agree and have modified our discussion of Figure 2G-I to more specifically describe the different modes of regulation reflected by these data.

Modified main text (Page 8, line 269):

“We next assessed individual components of the OXPHOS gene sets using RT-qPCR analysis of independently generated samples (Figure 2G). Analysis of polysome bound UQCRC2 (OXPHOS complex III) and SDHB (OXPHOS complex II) mRNA revealed an initial decrease in translation efficiency 24hr post Vem treatment, followed by a pronounced redistribution of these mRNA to heavy polysome fractions after 40hr treatment (Figure 2G), indicating an increase in mRNA translation efficiency following 40hr BRAFi. Total UQCRC2 mRNA remained unchanged, while a decrease in SDHB was observed (Figure 2H). Consistent with elevated translation efficiency, UQCRC2 and SDHB protein levels increased after 40hr Vem treatment and continued to increase throughout a 72hr treatment time course (Figure 2I). VDAC1 (voltage dependent anion channel 1) was translationally buffered, whereby no

change in polysome bound mRNA or protein levels were observed, despite a significant reduction in total mRNA levels (Figure 2G-I). These data therefore indicate multiple modes of post-transcriptional regulation for OXPHOS associated proteins in response to BRAFi. Demonstrating specificity in the analysis and regulation of pathway components, analysis of ATP5A (OXPHOS complex V) revealed no significant change in translation efficiency, total mRNA levels or protein levels.”

-In the discussion session, the authors should consider speculating regarding potential mechanisms of translational buffering in the context of adaptation to BRAFi and its functional consequences.

We have now extensively re-written our discussion to incorporate new conclusions arising from new data, and now also include a more detailed discussion on our analysis of translation, including buffering.

New main text (Page 18, line 788):

“mRNA translation has been implicated in responses to MAPK pathway inhibition and development of resistance in melanoma^{19, 40}, and a growing body of evidence now supports translational reprogramming as a mechanism that mediates adaptation to metabolic stress^{15, 41, 42}. Here, our systematic functional genomic analysis of metabolic response to BRAFi identified mRNA binding, transport and translation pathways as key regulators of the adaptive BRAFi response, and our analysis of the global translome directly supports these observations. Despite global suppression of translation during the early drug response phase, extensive reprogramming of specific pathways, including OXPHOS, occurs via changes in mRNA translation efficiency and translational buffering, revealing an underappreciated and prominent role for translational regulation of selective transcripts in the BRAFi response. The extensive translational buffering we identified throughout the BRAFi response is particularly intriguing and may represent an adaptive response to preserve activity of critical pathways. Interestingly, analysis of the translome following ER α inactivation in prostate cancer cells also revealed extensive translational buffering that appeared to sustain an adaptive proteome⁴³. Of note, a recent study described a mechanism whereby mRNA bound to polysomes are protected from degradation following exposure to stress, such as glucose deprivation⁴⁴. It is tempting to speculate that this mechanism may protect specific transcripts to allow rapid protein production during the adaptive stress response, and it is possible this phenomenon may contribute to the buffering phenotype identified in our polysome profiling analysis. Further investigation of BRAFi induced translational buffering is warranted to more completely understand the post-transcriptional mechanisms that underpin the BRAFi response.

Our analysis of individual OXPHOS related transcripts and proteins revealed regulation at the level of elevated translational efficiency (UQCRC2, SDHB, NDUF8), translational buffering (VDAC1) and protein stability (ATP5A). Analysis of *de novo* protein synthesis directly confirmed elevated translation of OXPHOS transcripts following BRAFi, and importantly, this was dependent on the RNA binding kinase UHMK1. Translational buffering of glycolysis genes (GLUT1 and HK2) also emerged from our polysome profiling analysis, however although these genes do not fit the classical definition of buffering due to a reduction in protein levels, these data support a model whereby translational mechanisms

may blunt rapid transcriptional inactivation of glycolysis pathway components in an attempt to preserve normal rates of glycolysis and facilitate cell survival. Supporting this model, *de novo* protein synthesis assays revealed GLUT1 translation was maximally suppressed following UHMK1 depletion in combination with BRAFi, reflective of disrupted translational buffering, and stronger glycolytic suppression was observed in the siUHMK1+BRAFi cells. Although GLUT1 is a key transcriptional target of MYC and HIF1 α , recent studies have also shown regulation of GLUT1 translation by RBPs during adaptive responses to hypoxia⁴⁵, and codon-specific translational reprogramming of glycolytic metabolism occurs in melanoma, in this case mediated by translational regulation of HIF1 α by uridine 34 (U₃₄) tRNA enzymes⁴⁰. Interestingly, these tRNA enzymes have been linked with translational buffering or “offsetting” in prostate cancer cells depleted of ER α ⁴³. Mechanistically, the reduction in metabolic protein synthesis in BRAFi+siUHMK1 treated cells likely reduces the capacity of these cells to cope with glucose deprivation associated with BRAFi, a model supported by a reduction in spare respiratory and glycolytic capacity, and the ability of the electron acceptors pyruvate and AKB to rescue cell death in the siUHMK1+BRAFi treated cells. Therefore, we suggest these translational mechanisms contribute to the metabolic plasticity observed in melanoma cells following BRAFi in order to facilitate survival. Notably, upregulation of OXPHOS proteins occurs in melanoma patients progressing on BRAF and MEK targeted therapy¹³, and patient response to BRAFi correlates with glycolytic response as assessed by FDG-PET imaging¹⁰, suggesting that inactivation of adaptive translational reprogramming may mitigate therapy induced metabolic plasticity and improve targeted therapy response in melanoma patients. Indeed, we observe a significant delay in resistance to MAPK targeted therapy in our preclinical mouse model implanted with melanoma cells depleted of UHMK1. Interestingly, UHMK1 has recently been reported to promote gastric cancer progression by promoting *de novo* purine synthesis⁴⁶, revealing a potentially broader role for this kinase in metabolic reprogramming in non-oncogene driven cancers, however in this case, it was UHMK1’s kinase activity that mediated this effect. Because UHMK1 knock out mice remain viable with no severe defects⁴⁷, and both the kinase activity and RNA binding domain are required for UHMK1-mediated regulation of BRAFi sensitivity, this makes UHMK1 an attractive therapeutic target and development of specific inhibitors is a priority.”

- “This data suggests that UHMK1 depletion may cooperate with BRAFi to elicit a double-hit on the glycolysis pathway, whereby both GLUT1 mRNA transcription and translation is concurrently switched off” This statement should perhaps be clarified, since the effects of UHMK1 have broad effects on metabolism (including OXPHOS).

We agree and have re-worded our statements describing these observations.

Modified main text (Page 15, line 629):

“Strikingly, we also observed that although GLUT1 protein synthesis was decreased following Vem treatment, this reduction was significantly more pronounced following UHMK1 knockdown (Figure 6B-C). Notably, these data are consistent with polysome profiling analysis of GLUT1 mRNA (Figure S3D) which indicated that cells may attempt to preserve critical components of the glycolysis pathway via a translational mechanism, and suggest that UHMK1 depletion can overcome this process and thereby achieve more rapid and complete inhibition of GLUT1 protein synthesis. These observations are consistent with

enhanced suppression of glycolytic function observed in our siUHMK1+Vem treated cells (Figure 3).”

-“Data” are plural. Consider changing “this data” to “these data”.

We have adjusted the text in our manuscript accordingly.

Reviewer #4 (Remarks to the Author):

The manuscript by Smith et al., described a potential mechanism of non-mutational adaptive reprogramming responsible for drug resistance in melanoma. The manuscript started from a genome-wide RNAi screen of potential target genes that may improve the efficiency of targeted therapy against BRAF in melanoma cells. The screening and gene expression profiling revealed that the mRNA of metabolic proteins, including glucose transporters and oxyphosphatases enzymes, were selectively transported and translated. Among those post-transcriptional regulators, UHMK1, a RNA binding kinase required for mitochondrial flexibility, was proposed to be responsible for selective mRNA translocation and translation, as well as metabolic remodeling in cell adaption. In addition, inactivation of UHMK1 improves the sensitivity of targeted therapy against BRAF, and delays drug resistance and disease recurrence. The authors have done tremendous amount of work including high-throughput screenings. Some parts in the manuscript are novel and worthy of publication. Here are some of my concerns.

Major points:

1. It is not clear how BRAF and/or MEK inhibition upregulates mRNA transport and translation genes' expression, including UHMK1's. Can authors find out the molecular mechanism?

It remains unclear how BRAF/MEKi modifies expression of mRNA transport and translation gene expression, as demonstrated by our analysis of the patient dataset (Figure 1). We included these data to establish relevance of the RNA transport and translation gene set identified in our functional screen to patients treated with BRAF/MAPKi. Although of interest, we believe the molecular mechanisms of this aspect of our observations would constitute a new study and is beyond the scope of the current manuscript.

However, our analysis of poly(A)⁺-mRNA export confirms a prominent defect in mRNA export following BRAFi, which is a phenotype consistent with our polysome profiling data whereby we observe potent suppression of global mRNA translation (Figure 2). We believe further investigation of how this occurs is of considerable interest to understand the BRAFi response, but we believe this represents a completely new line of investigation. Moreover, our functional screen and polysome profiling analysis indicated a key role for selective mRNA translation pathways as mediators of the BRAFi response, and it was this aspect of the biology that we were interested in focusing on with the current study.

Our data also indicates that BRAF inhibition induces post-translational regulation of UHMK1, whereby we observe nuclear to cytoplasm translocation of the protein (see Figure 6F), therefore indicating that the molecular mechanism underlying regulation of the RNA

binding, transport and translation gene set may not be merely transcriptional.

2. To conclude that depletion of UHMK1 sensitizes BRAFi treatment. The combination treatment need to be further evaluated, in order to determine the exact effect (synergistic, additive or antagonistic). Otherwise, the effects of UHMK1 inhibition may be independent of BRAFi treatment. Actually, some of the data in Figure 3 seem to support the additive effect. Figure 7B has the same problem.

We respectively disagree with this comment. In the majority of our data we do not see statistically significant effects induced by UHMK1 depletion in the absence of BRAF inhibition. Indeed, we selected UHMK1 from our screen based on selective activity in BRAFi cells, and the vast majority of our data support this initial finding, as indicated by a lack of significant effects in cells treated with DMSO+siUHMK1. Interestingly, we do see more prominent effects resulting from inactivation of the kinase *in vivo* (Figure 7C-F), however in this instance, only one gRNA produces a statistically significant result therefore we think it is inaccurate to conclude that the effects we observe upon UHMK1 depletion are independent of BRAFi.

3. One of the most significant findings in this paper is that UHMK1 selectively regulates the mRNAs of certain metabolic or mitochondrial genes upon BRAFi, but it is unclear how UHMK1 obtains such preferences of mRNAs. Which motifs/domains of UHMK1 are responsible for the binding/interaction? Can authors comment?

We have now generated a panel of UHMK1 mutant proteins, either lacking kinase activity or the UHM RNA binding domain, alone or in combination (Figure S4). We have performed RNA-IP experiments to assess the UHMK1 domains required for association to UQCRC2 and GLUT1 (Figure S5D). Because of the different levels of the mutant proteins (caused by changes in endogenous protein stability, rather than expression, see Figure S4), we normalized mRNA levels to input UHMK1 protein levels. These data demonstrate that UHMK1 requires both the UHM domain and its kinase activity in order to associate with UQCRC2 and GLUT1 mRNA (Figure S5E). Due to the issues encountered with changes in stability of the \otimes RBD mutant proteins we have only included these data as supplementary information, however we believe these experiments are still sufficient to assess the role of the UHM domain in the RNA interactions. We also note that we achieved equal expression of the wildtype and K54R kinase dead mutant proteins, and therefore clearly define a role for UHMK1 kinase activity in the association with UQCRC2 and GLUT1 mRNA.

New main text (Page 14, line 568):

“We were next interested in assessing the requirement of the different UHMK1 domains in regulation of these transcripts. To do this, we made use of our UHMK1 gRNA A375 cell line panel expressing the UHMK1-V5, UHMK1-K54R-V5 (kinase dead), UHMK1- Δ RBD-V5 (lacking UHM RBD) and UHMK1-K54R- Δ RBD-V5 mutant proteins (Figure S4D). We first verified immunoprecipitation of these proteins (Figure S5D), then assessed association of the different proteins with UQCRC2, GLUT1 and ATP5A mRNA using RT-qPCR (Figure S5E). Because of the different levels of the mutant proteins (due to changes in stability of the Δ RBD-V5 proteins, see Figure S4), we normalized mRNA levels to input UHMK1 protein levels. Notably, interaction between UHMK1-V5 protein and UQCRC2 and GLUT1 mRNA

following BRAFi was confirmed in these independently generated cells expressing UHMK1-V5, however these interactions were significantly reduced in cells expressing the UHMK1-K54R-V5, UHMK1- Δ RB-D-V5 and UHMK1-K54R- Δ RB-D-V5 mutant proteins (Figure S5E). Analysis of ATP5A revealed no significant association with any of the UHMK1 proteins. Interestingly, these data establish that UHMK1 requires both the UHM domain and its kinase activity to associate with GLUT1 and UQCRC2 mRNA following BRAFi.”

4. It should be explained why there was no further change in VEM-treated melanoma after UHMK1 inhibition (Fig. 5AB). Does this mean the selective role for UHMK1 in mRNA transport does not play a role in BRAFi, and is therefore independent of BRAFi therapy response?

Figure 5A-B assesses global mRNA transport using a poly-A probe which recognises all mRNA. In Figure 5C-E we are looking at specific transcripts which reveals a selective role for UHMK1 in mRNA transport, specifically in the context of Vem treated cells. These data suggest that UHMK1 can function both as a regulator of global nuclear-cytoplasmic mRNA transport in melanoma cells and as a selective regulator of specific transcripts, dependent on cellular context (with or without BRAFi). Given that UHMK1 selectively binds to specific mRNA only in the presence of BRAFi and defects in mRNA transport and translation correlate with this observation, we suggest the selective role of UHMK1 in mRNA transport is important to the BRAF inhibitor response. Indeed, deletion of the UHM RNA binding domain prevented rescue of the UHMK1-gRNA effects on drug sensitivity. We now more clearly make this distinction in the main text.

Modified text (Page 13, line 503):

“Notably, nuclear accumulation of poly(A)⁺ mRNA was also observed in UHMK1 depleted cells, confirming a role for UHMK1 in mRNA export in the context of melanoma cells. BRAFi also gave rise to a significant increase in the poly(A)⁺ nuclear to cytoplasm ratio (Figure 5B), however no further change was observed in the siUHMK1+Vem and siNXF1+Vem treated cells. These data identify UHMK1 as a regulator of global mRNA export in melanoma cells, however this role is unlikely to contribute to the effects of UHMK1 depletion in the context of BRAFi. These data also establish a prominent role for mRNA export in the BRAFi response, consistent with the findings of our genome wide screen.

The more modest phenotype of UHMK1 compared to NXF1 depletion indicated a selective role for UHMK1 in mRNA export. UHMK1 directly regulates localization and translation of specific mRNA transcripts by binding to mRNA^{28, 29}. Therefore to extend our observations, we next assessed individual mRNA transcripts encoding GLUT1 and UQCRC2 that showed evidence of post-transcriptional regulation from our polysome profiling analysis and are critical components of the glycolysis and oxidative metabolism pathways, respectively.”

5. The cellular data and NOD model data look promising. However, UHMK1 is widely expressed in brain, endocrine tissues, liver, and the entire gastrointestinal tract (human protein atlas). Inhibition of adaptive mitochondrial metabolism by targeting UHMK1 may also impair the tolerance of normal tissues requiring normal mitochondrial metabolism. Serious side effects may be expected. Can authors comment the translational value and approach of targeting UHMK1.

One of the most promising features of UHMK1 inactivation with regard to clinical efficacy and tolerability, is that UHMK1 inactivation is synthetic lethal with BRAF inhibition, whereby the majority of phenotypes are highly selective to BRAF-inhibitor treated cells, with only mild effects observed after UHMK1 depletion in DMSO treated melanoma cells. This suggests that the effects of UHMK1 inhibition will occur primarily in cells expressing the BRAF (or NRAS) oncogene. Moreover, UHMK1 knock-out mice are viable and no difference in total weight or brain weight could be detected when compared to control mice (Manceau V et al, 2012). UHMK1 knock-out mice also perform similarly to control mice in most behavioral tests; the only differences observed were spontaneous activity and contextual fear conditioning indicating a mild defect in fear responses. Finally, UHMK1 is a Ser/Thr kinase with little sequence similarity to other human kinases, and this uniqueness suggests the possibility of generating inhibitors with high specificity which will further reduce the likelihood of unwanted off-target side effects. Together, these data suggest that inhibitors of UHMK1 may represent a valuable tool in the clinic when used in combination with oncogene targeted therapies. We have now included some of these points in our discussion.

New main text (Page 20, line 869):

“Because UHMK1 knock out mice remain viable with no severe defects⁴⁷, and both the kinase activity and RNA binding domain are required for UHMK1-mediated regulation of BRAFi sensitivity, this makes UHMK1 an attractive therapeutic target and development of specific inhibitors is a priority.”

6. Most of the functional tests were carried in siUHMK1 condition, I would like to know whether supplementing UHMK1 in a UHMK1-low, BRAFi-sensitive tumor cells will enhance the cell viability and drug resistance upon VEM treatment.

UHMK1 is expressed at high levels in all melanoma cell lines we have tested (within the 90th percentile of top expressed genes in a panel of 71 melanoma cell lines, as assessed by microarray expression analysis). When the kinase was overexpressed in wild-type melanoma cells we did not see any overt effects on cellular responses to Vem within the early drug response phase. This suggests that the high levels of endogenous UHMK1 are saturating and mitigate the additional effects of overexpression of exogenous protein.

Minor points

1. The title needs to be more specific. Although UHMK1 was mentioned as a paradigm, it should be included in the title, because the experiments related to UHMK1 occupied over 5/7 of the whole manuscript. Besides, there are almost a dozen regulatory mechanisms post-transcription, and mRNA transport and selective translation cannot represent them all.

We agree and have now modified the title of our manuscript.

“Adaptive translational reprogramming of metabolism by the RNA binding kinase UHMK1 limits response to targeted therapy in BRAF^{V600} melanoma.”

2. Please provide significance (if there is any) between each two groups in the column chart of Fig. 5 C-F

We have now modified Figure 5 to accommodate other reviewer's comments, and we now include all significance indicators for all relevant comparisons.

3. Can author describe the difference between the two used cell lines WM266.4 and A375 in genetics and cell behaviors, and explain why A375 but not WM266.4 was used in NOD-NSG models.

The A375 and WM266.4 cell lines are both BRAF^{V600} mutant melanoma cell lines, however the A375 cell line harbours the V600E mutation and the WM266.4 cell line harbours the V600K mutation. The cell lines behave similarly, however the WM266.4 cell line is less sensitive to BRAFi than the A375 cell line. The WM266.4 cell line was chosen for the initial screen based on the moderate sensitivity of this cell line to BRAF inhibitors (Parmenter et al, 2014), therefore we reasoned it was a good model for a drug enhancement screen.

We chose the A375 cell line for subsequent CRISPR genome editing and *in vivo* experiments because we required a cell line that was capable of single cell cloning. This allowed employment of transient synthetic gRNA to edit UHMK1, and single cell sorting allowed expansion of a genetically homogeneous cell population that could be verified by re-expressing the UHMK1-V5 construct (stable expression of UHMK1 gRNA would preclude re-expression of the UHMK1 cDNA). These verified cell lines were then subsequently utilised for the NSG *in vivo* model. Unfortunately, our attempts at single cell cloning the WM266.4 cell line after CRISPR genome editing failed, precluding their use in rescue experiments and ultimately in our *in vivo* model.

Adamson, B., A. Smogorzewska, F. D. Sigoillot, R. W. King and S. J. Elledge (2012). "A genome-wide homologous recombination screen identifies the RNA-binding protein RBMX as a component of the DNA-damage response." *Nat Cell Biol* **14**(3): 318-328.

Boehm, M., T. Yoshimoto, M. F. Crook, S. Nallamshetty, A. True, G. J. Nabel and E. G. Nabel (2002). "A growth factor-dependent nuclear kinase phosphorylates p27(Kip1) and regulates cell cycle progression." *Embo j* **21**(13): 3390-3401.

Brass, A. L., D. M. Dykxhoorn, Y. Benita, N. Yan, A. Engelman, R. J. Xavier, J. Lieberman and S. J. Elledge (2008). "Identification of host proteins required for HIV infection through a functional genomic screen." *Science* **319**(5865): 921-926.

Cambray, S., N. Pedraza, M. Rafel, E. Gari, M. Aldea and C. Gallego (2009). "Protein kinase KIS localizes to RNA granules and enhances local translation." *Mol Cell Biol* **29**(3): 726-735.

Falkenberg, K. J., C. M. Gould, R. W. Johnstone and K. J. Simpson (2014). "Genome-wide functional genomic and transcriptomic analyses for genes regulating sensitivity to vorinostat." *Sci Data* **1**: 140017.

McArthur, G. A., I. Puzanov, R. Amaravadi, A. Ribas, P. Chapman, K. B. Kim, J. A.

Sosman, R. J. Lee, K. Nolop, K. T. Flaherty, J. Callahan and R. J. Hicks (2012). "Marked, homogeneous, and early [18F]fluorodeoxyglucose-positron emission tomography responses to vemurafenib in BRAF-mutant advanced melanoma." *J Clin Oncol* **30**(14): 1628-1634.

Parmenter, T. J., M. Kleinschmidt, K. M. Kinross, S. T. Bond, J. Li, M. R. Kaadige, A. Rao, K. E. Sheppard, W. Hugo, G. M. Pupo, R. B. Pearson, S. L. McGee, G. V. Long, R. A. Scolyer, H. Rizos, R. S. Lo, C. Cullinane, D. E. Ayer, A. Ribas, R. W. Johnstone, R. J. Hicks and G. A. McArthur (2014). "Response of BRAF-mutant melanoma to BRAF inhibition is mediated by a network of transcriptional regulators of glycolysis." Cancer Discov **4**(4): 423-433.

Pedraza, N., R. Ortiz, A. Cornado, A. Llobet, M. Aldea and C. Gallego (2014). "KIS, a kinase associated with microtubule regulators, enhances translation of AMPA receptors and stimulates dendritic spine remodeling." J Neurosci **34**(42): 13988-13997.

Simpson, K. J., L. M. Selfors, J. Bui, A. Reynolds, D. Leake, A. Khvorova and J. S. Brugge (2008). "Identification of genes that regulate epithelial cell migration using an siRNA screening approach." Nat Cell Biol **10**(9): 1027-1038.

Smith, J. A., E. A. White, M. E. Sowa, M. L. Powell, M. Ottinger, J. W. Harper and P. M. Howley (2010). "Genome-wide siRNA screen identifies SMCX, EP400, and Brd4 as E2-dependent regulators of human papillomavirus oncogene expression." Proc Natl Acad Sci U S A **107**(8): 3752-3757.

Sullivan, L. B., D. Y. Gui, A. M. Hosios, L. N. Bush, E. Freinkman and M. G. Vander Heiden (2015). "Supporting Aspartate Biosynthesis Is an Essential Function of Respiration in Proliferating Cells." Cell **162**(3): 552-563.

Williams, S. P., C. M. Gould, C. J. Nowell, T. Karnezis, M. G. Achen, K. J. Simpson and S. A. Stacker (2017). "Systematic high-content genome-wide RNAi screens of endothelial cell migration and morphology." Sci Data **4**: 170009.

Reviewers' comments:

Reviewer #1 (Remarks to the Author):

The manuscript has been largely improved with the new experiments and text adjustments. I have only two minor comments:

Figure 2J.- Contrary to the statement that CHX did not affect total ATP5A protein levels, I do see a reduction at 48h and 72h in the + CHX versus the -CHX lanes, which will be probably more evident in a less exposed gel. Furthermore, if no change in the total mRNA levels, polysome association or protein levels were indeed observed in a time-course of Vem treatment, there is no reason -in principle- to believe that there is regulation of this protein at any level. To this reviewer, ATP5A appears to be a highly stable protein, but this does not mean that stability is 'regulated'. Thus, it would be sufficient to simply state that the protein is highly stable, here and elsewhere this is mentioned in the text (e.g. page 16, lanes 679-680; page 19, lane 832).

Page 14 lane 552.- Substitute 'mRNA' by 'RNAs', as MALAT1 is a non-coding RNA.

Reviewer #2 (Remarks to the Author):

The revised and re-titled manuscript "Adaptive translational reprogramming of metabolism by the RNA binding kinase UHMK1 limits response to targeted therapy in BRAFV600 melanoma" by Smith et al., has been improved by responding to the initial critique.

Importantly, however, the in vivo analyses (Fig 7E/F) are still limited to a single cell line which makes it impossible to judge whether the observed effects are generalizable.

Moreover, the attempt to conduct structure-function analyses domains required did not succeed to tease out any mechanistic insight (Fig 3G). Neither the K54R or deltaRBD, nor double K54R/deltaRBD, altered the sensitivity of A375-sgUHMK1 cells to vemurafenib compared to UHMK1(wt) rescue. But the deltaRBD is a pretty substantial disruption of the entire protein, and should have used single aa level mutations instead. In addition, the A375-sgUHMK1 cells does not seem to be as affected by vemurafenib as A375-siUHMK1 are (Fig 3D), which perhaps may help to explain why the rescue is not seen. To this end, there is no data indicating that A375-sUHMK1 cells had bi-allelic targeted UHMK1 resulting in complete loss of protein.

These two outstanding issues should be resolved before proceeding with this manuscript.

Reviewer #3 (Remarks to the Author):

I thought that the authors have responded to my comments and questions in a satisfactory manner. To this end, I have no further concerns and find that the revised manuscript merits publication.

Reviewer #4 (Remarks to the Author):

Authors answered all reviews' concerns. After revision, data are more solid and convincing. Current version of the manuscript is able to be accepted.

Reviewer #2 (Remarks to the Author):

The revised and re-titled manuscript "Adaptive translational reprogramming of metabolism by the RNA binding kinase UHMK1 limits response to targeted therapy in BRAFV600 melanoma" by Smith et al., has been improved by responding to the initial critique.

Importantly, however, the *in vivo* analyses (Fig 7E/F) are still limited to a single cell line which makes it impossible to judge whether the observed effects are generalizable.

We have now performed an additional *in vivo* experiment to address the concern of generalizability. For this experiment, we used WM266.4-CAS9 cells stably expressing two independent pools of UHMK1 gRNA, using the transEDIT system. Importantly, we observed a significant increase in overall survival in mice implanted with both independent WM266.4 sgUHMK1 cells when treated with BRAFi+MEKi compared to mice implanted with the control cell line. These new data clearly demonstrate our observations are not selective to one *in vivo* melanoma model and are displayed in Figure S8. We also note that we have now demonstrated significant effects *in vivo* using 4 independent UHMK1 knockdown reagents which further strengthens these observations.

New text in manuscript (line 634):

Importantly, we also observed a significant increase in overall survival in mice implanted with WM266.4 sgUHMK1 cells treated with BRAFi+MEKi compared to mice implanted with the control cell line (Figure S8), indicating these observations are not selective to one *in vivo* melanoma model.

Moreover, the attempt to conduct structure-function analyses domains required did not succeed to tease out any mechanistic insight (Fig 3G). Neither the K54R or deltaRBD, nor double K54R/deltaRBD, altered the sensitivity of A375-sgUHMK1 cells to vemurafenib compared to UHMK1(wt) rescue. But the deltaRBD is a pretty substantial disruption of the entire protein, and should have used single aa level mutations instead.

We have now generated multiple point mutations in 2 conserved motifs found in the UHMK1 UHM domain. UHM domains contain consensus RNA recognition motifs RNP1/RNP2, as well as a conserved RXF motif that has been shown to mediate interactions with UHM ligand motif (ULM) containing proteins¹. The UHMK1 RNP1 motif shows conserved residues with the consensus RNP1 sequence at position 1,2 and 5 (see Figure S7A)², and structural modelling using AlphaFold (<https://alphafold.ebi.ac.uk/>)³ supports an important function for these residues based on their predicted involvement in hydrogen bond formation and presence in the juxtaposed RNP2/RNP1 core (see Figure S7A). We also note that mutation of the RXF motif to AAA in the UHM domain containing protein SPF45 is sufficient to disrupt interactions with RNA processing proteins and inactivate its RNA processing function⁴. Based on these features of UHM domains and observations in other UHM domain containing proteins, we introduced point

mutations in conserved residues of the UHMK1 RNP1 (R369A-G370A-Q-V-F372A) and RXF (R392A-M393A-F394A) motifs (see Figure S7B). We achieved equivalent expression of wildtype UHMK1, the K54R kinase dead mutant, and the 2 UHM mutant proteins (Figure S7C). Analysis of p27 levels identified the expected increase in the K54R mutant cells. The RXF mutant also increased p27 levels, however no change was observed in the UHM-RNP1 mutant cells. Importantly, the RNP1 but not RXF motif in the UHMK1 UHM domain was critical for binding β -actin RNA (see Figure S7D). Because the UHM-RNP1 mutations disrupted RNA associations with no detectable change in kinase activity (using p27 as readout), we used this UHM mutant to specifically assess the requirement of UHMK1's RNA processing function in the BRAFi response. First, we established the UHM-RNP1 mutant does not efficiently bind to UQCRC2 or GLUT1 mRNA (see Figure 7A). We then reperformed our drug sensitivity assays which showed increased sensitivity in UHMK1-gRNA cells was rescued by expression of UHMK1-V5, but not the kinase dead K54R-V5 nor the UHM-RNP1-V5 mutant proteins (see Figure 7B). These new data therefore demonstrate that UHMK1 regulates response to BRAFi via both its kinase and UHM domain. They also confirm an essential role for the RNA processing function of UHMK1 in mediating adaptive responses to BRAFi.

New text (line 576):

UHMK1 requires a functional kinase and UHM domain to regulate the BRAFi response.

We were next interested in establishing the role of UHMK1's kinase activity and RNA processing function mediated via the UHM domain in the response to BRAFi. The kinase domain of UHMK1 shows limited homology to known kinases, however a K54R mutation in the putative active site extinguishes kinase activity⁵. The UHM domain of UHMK1 has not been extensively characterised, however there are multiple features conserved across UHM domain containing proteins¹. The UHM domain is classified as an RNA binding domain based on the presence of ribonucleoprotein (RNP) 1 and RNP2 RNA recognition motifs, however these motifs are atypical, which is consistent with the previously documented ability of UHM domains to interact with RNA processing proteins¹. The UHMK1 RNP1 motif shows conserved residues with the consensus RNP1 sequence at position 1,2 and 5 (Figure S7A)², and structural modelling using AlphaFold (<https://alphafold.ebi.ac.uk/>)³ supports an important function for these residues based on their predicted involvement in hydrogen bond formation and presence in the juxtaposed RNP2/RNP1 core (Figure S7A). The UHM domain also contains a conserved RXF motif that is required for interaction with UHM ligand motif (ULM) containing proteins¹. Mutation of the RXF motif to AAA in the UHM domain containing protein SPF45 is sufficient to disrupt interactions with RNA processing proteins and inactivate its RNA processing function⁴. Based on these observations, we introduced the K54R mutation in the kinase domain to assess the role of UHMK1 kinase activity in BRAFi responses, and point mutations in conserved residues of the UHMK1 RNP1 (R369A-G370A-Q-V-F372A) and RXF (R392A-M393A-F394A) motifs (Figure S7B). Inactivation of the kinase domain in the K54R mutant was verified by increased accumulation of p27 protein levels, an established biomarker of UHMK1 kinase activity (Figure S7C)⁵. Notably, we also observed increased p27 levels in the RXF, but not the RNP1, mutant expressing cells. RNA association was examined by analysing a previously established mRNA target of UHMK1, β -actin⁶, in UHMK1 RNA-immunoprecipitation (RNA-IP) experiments (Figure S7C-D). The RNP1 but not RXF motif in the UHMK1 UHM domain is critical

for complexing with β -actin RNA, whilst the K54R mutant also showed reduced association with β -actin mRNA (Figure S7D). Because the UHM-RNP1 mutant protein did not associate with mRNA or alter UHMK1 kinase activity, we used this mutant protein to specifically assess the requirement of UHMK1's RNA processing function in the response to BRAFi. First, we established the UHM-RNP1 mutant does not efficiently bind to UQCRC2 or GLUT1 mRNA (Figure 7A). To assess the contribution of these domains in BRAFi responses, we first genetically inactivated UHMK1 using CRISPR-Cas9 (Figure S7E) and confirmed increased sensitivity of A375 cells to BRAFi (Figure 7B). Notably the increased sensitivity in UHMK1-gRNA cells was rescued by expression of UHMK1-V5, but not the kinase dead K54R-V5 nor the UHM-RNP1-V5 mutant proteins (Figure 7B). Together, these data demonstrate that UHMK1 regulates response to BRAFi via both its kinase and UHM domain, and thus confirm an essential role for both the kinase and RNA processing function of UHMK1 in mediating adaptive responses to BRAFi.

Additional data

We also provide additional new data which further strengthens the role of UHMK1-mediated RNA transport in the response to BRAFi. We applied a modified RNA-IP protocol to allow UHMK1 immunoprecipitation from polysome fractions. After verifying UHMK1 protein was successfully isolated from polysomes using the V5 tag, we analysed its RNA cargo. Significantly, we could detect UQCRC2 mRNA in association with UHMK1 protein at actively translating polysomes specifically in the context of cells treated with BRAFi (see Figure 6F-G), strongly supporting a role for UHMK1 in delivery of selective transcripts to polysomes to facilitate their translation.

In addition, the A375-sgUHMK1 cells does not seem to be as affected by vemurafenib as A375-siUHMK1 are (Fig 3D), which perhaps may help to explain why the rescue is not seen. To this end, there is no data indicating that A375-sUHMK1 cells had bi-allelic targeted UHMK1 resulting in complete loss of protein.

We now provide Sanger sequencing data verifying the bi-allelic CRISPR editing of UHMK1 in the A375 knockout clone. Alignment of the sequence obtained from the UHMK1 gRNA cells to the UHMK1 reference sequence identified a 10bp deletion resulting in a frameshift mutation in the kinase domain of UHMK1 and non-sense sequence for the remainder of the gene (Figure 1). The sanger sequencing profile indicates a homozygous, bi-allelic mutation due to the absence of multiple sequencing reads generated from this clonal cell line (Figure 2). We also suggest that the differences observed between the UHMK1-gRNA cells and the siUHMK1 cells are due to the differences associated with a stable, chronic knockdown of a protein versus a transient knockdown of a protein. We prefer the transient knockdown approach as it more faithfully recapitulates the kinetics of acute protein inhibition that occurs with a drug and avoids the adaptive reprogramming that can occur in stable knockdown cells that may lead to changes in cellular responses.

Figure 1. UHMK1 gRNA-clone-4.12 alignment to UHMK1 gDNA ref seq

```

1 TGTGGCAGGTACAGAGCCGCTGGGTAGCGGCTCCTCCGCTCGGTGTATCGGGTTCGCTGCTGGGCAACCCGGCTCGCCCCGGCGCCCTCAAGCA 100
1 -----GGTTCGCTGCTGGGCAACCCGGCTCGCCCCGGCGCCCTCAAGCA 48
101 GTTCTTGCCGCCAGGAACCCGGGGCTGCGGCCCTCGCCCGGAGTATGTTTCGCAAAGAGAGGGCGGCTGGAACAGTTGCAGGGTCACAGAAAC 200
49 GTTCTTGCCG-----CCGGGGCTGCGGCCCTCGCCCGGAGTATGTTTCGCAAAGAGAGGGCGGCTGGAACAGTTGCAGGGTCACAGAAAC 138
201 ATCGGTAATGCGCGTGTCTCCTTCTCTTTCGCCAGGTCACAGTCCGAGCACACTTCTCCTCGCTGCTGGCGTTCATCTTCCTCCCTTCTGC 300
139 ATCGGTAATGCGCGTGTCTCCTTCTCTTTCGCCAGGTCACAGTCCGAGCACACTTCTCCTCGCTGCTGGCGTTCATCTTCCTCCCTTCTGC 238

```

Figure 2. UHMK1 gRNA-clone-4.12 Sanger sequencing profile

1. Kielkopf, C.L., Lücke, S. & Green, M.R. U2AF homology motifs: protein recognition in the RRM world. *Genes & development* **18**, 1513-1526 (2004).
2. Maris, C., Dominguez, C. & Allain, F.H. The RNA recognition motif, a plastic RNA-binding platform to regulate post-transcriptional gene expression. *The FEBS journal* **272**, 2118-2131 (2005).
3. Jumper, J. *et al.* Highly accurate protein structure prediction with AlphaFold. *Nature* (2021).
4. Corsini, L. *et al.* U2AF-homology motif interactions are required for alternative splicing regulation by SPF45. *Nat Struct Mol Biol* **14**, 620-629 (2007).
5. Manceau, V., Kielkopf, C.L., Sobel, A. & Maucuer, A. Different requirements of the kinase and UHM domains of KIS for its nuclear localization and binding to splicing factors. *Journal of molecular biology* **381**, 748-762 (2008).
6. Cambray, S. *et al.* Protein kinase KIS localizes to RNA granules and enhances local translation. *Molecular and cellular biology* **29**, 726-735 (2009).

Reviewers' comments:

Reviewer #2 (Remarks to the Author):

The re-revised manuscript "Adaptive translational reprogramming of metabolism by the RNA processing kinase UHMK1 limits the response to targeted therapy in BRAF^{V600} melanoma" by Smith et al, has been substantially improved in this revision as it a) now includes a second melanoma cell line for the in vivo BRAF-inhibitor treatments (Fig S8), which suggests robustness, and more detailed structure-function analyzes that indicates that each of the kinase and RNA-binding domains are required (Fig 7B, S7A-D).

With only minor concern, however, the genetic examination as to whether A375-sgUHMK1 have edited both alleles is not clear. The authors should have sequenced individual cloned PCR products, which should then reveal whether the large and short are contained within the same DNA fragment (cis/mono-allelic, or separate (trans/bi-allelic). This exercise would only help to answer whether UHMK1 is required for overall growth, and haploinsufficiency retards proliferation, which it might be as WM266.4-sgUHMK1 cells grow moderately slower. To this end, it is noteworthy that each of A375 and WM266.4 are only moderately sensitive to BRAF-inhibition as tumors growing in vivo.

Reviewer #2 (Remarks to the Author):

The re-revised manuscript "Adaptive translational reprogramming of metabolism by the RNA processing kinase UHMK1 limits the response to targeted therapy in BRAF^{V600} melanoma" by Smith et al, has been substantially improved in this revision as it a) now includes a second melanoma cell line for the *in vivo* BRAF-inhibitor treatments (Fig S8), which suggests robustness, and more detailed structure-function analyzes that indicates that each of the kinase and RNA-binding domains are required (Fig 7B, S7A-D).

With only minor concern, however, the genetic examination as to whether A375-sgUHMK1 have edited both alleles is not clear. The authors should have sequenced individual cloned PCR products, which should then reveal whether the large and short are contained within the same DNA fragment (*cis*/mono-allelic, or separate (*trans*/bi-allelic). This exercise would only help to answer whether UHMK1 is required for overall growth, and haploinsufficiency retards proliferation, which it might be as WM266.4-sgUHMK1 cells grow moderately slower. To this end, it is noteworthy that each of A375 and WM266.4 are only moderately sensitive to BRAF-inhibition as tumors growing *in vivo*.

We thank the reviewer for their helpful comments and agree that the manuscript is substantially improved.

Given the additional validation of the knockout cells is only minor concern, we do not provide any new data validating UHMK1 inactivation in our CRISPR cell lines.

With regard to the low sensitivity of A375 and WM266.4 cells to BRAF/MEKi *in vivo*, interestingly, we consistently observe reduced efficacy of BRAF/MEKi across melanoma models in fully immune compromised models. This likely reflects the role of the immune system in mediating response to these inhibitors, which has been extensively published. The advantage of this model is that it allows to more cleanly assess cell intrinsic mediators of BRAF/MEKi response and resistance, and we believe using a moderately sensitive model is powerful for identifying new combinations that can improve response.